# Calibration and Consistency of Adversarial Surrogate Losses

**Pranjal Awasthi**
Google Research
New York, NY 10011
pranjalawasthi@google.com

**Natalie S. Frank**
Courant Institute
New York, NY 10012
nf1066@nyu.edu

**Anqi Mao**
Courant Institute
New York, NY 10012
aqmao@cims.nyu.edu

**Mehryar Mohri**
Google Research & Courant Institute
New York, NY 10011
mohri@google.com

**Yutao Zhong**
Courant Institute
New York, NY 10012
yutao@cims.nyu.edu

## Abstract

Adversarial robustness is an increasingly critical property of classifiers in applications. The design of robust algorithms relies on surrogate losses since the optimization of the adversarial loss with most hypothesis sets is NP-hard. But, which surrogate losses should be used and when do they benefit from theoretical guarantees? We present an extensive study of this question, including a detailed analysis of the $\mathcal{H}$-*calibration* and $\mathcal{H}$-*consistency* of adversarial surrogate losses. We show that convex loss functions, or the supremum-based convex losses often used in applications, are not $\mathcal{H}$-*calibrated* for common hypothesis sets used in machine learning. We then give a characterization of $\mathcal{H}$-*calibration* and prove that some surrogate losses are indeed $\mathcal{H}$-*calibrated* for the adversarial zero-one loss, with common hypothesis sets. In particular, we fix some calibration results presented in prior work for a family of linear models and significantly generalize the results to the nonlinear hypothesis sets. Next, we show that $\mathcal{H}$-*calibration* is not sufficient to guarantee consistency and prove that, in the absence of any distributional assumption, no continuous surrogate loss is consistent in the adversarial setting. This, in particular, proves that a claim made in prior work is inaccurate. Next, we identify natural conditions under which some surrogate losses that we describe in detail are $\mathcal{H}$-*consistent*. We also report a series of empirical results which show that many $\mathcal{H}$-*calibrated* surrogate losses are indeed not $\mathcal{H}$-*consistent*, and validate our theoretical assumptions. Our adversarial $\mathcal{H}$-consistency results are novel, even for the case where $\mathcal{H}$ is the family of all measurable functions.

## 1   Introduction

Complex multi-layer neural networks trained on large datasets have achieved a remarkable performance in several applications in recent years, in particular in speech and visual recognition tasks (Sutskever et al., 2014; Krizhevsky et al., 2012). However, these rich models are susceptible to imperceptible perturbations (Szegedy et al., 2013). A complex neural network may, for example, misclassify a traffic sign, as a result of a minor variation, which may be the presence of a small advertisement sticker on the sign. Such misclassifications can have dramatic consequences in practice, for example with self-driving cars. These concerns have motivated the study of *adversarial robustness*, that is the design of classifiers that are robust to small $\ell_p$ norm input perturbations (Goodfellow et al., 2014; Madry et al., 2017; Tsipras et al., 2018; Carlini and Wagner, 2017). The standard 0/1 loss is

35th Conference on Neural Information Processing Systems (NeurIPS 2021).

then replaced with a more stringent *adversarial loss*, which requires a predictor to correctly classify an input point $\mathbf{x}$ and also to maintain the same classification for all points at a small $\ell_p$ distance of $\mathbf{x}$. But, can we devise efficient learning algorithms with theoretical guarantees for the adversarial loss?

Designing such robust algorithms requires resorting to appropriate surrogate losses since optimizing the adversarial loss is NP-hard for most hypothesis sets. A key property for surrogate adversarial losses is their consistency, that is, that exact or near optimal minimizers of the surrogate loss be also exact or near optimal minimizers of the original adversarial loss. The notion of consistency has been extensively studied in the case of the standard $0/1$ loss or the multi-class setting (Zhang, 2004; Bartlett et al., 2006; Tewari and Bartlett, 2007; Steinwart, 2007). However, those results or proof techniques cannot be used to establish or characterize consistency in adversarial settings. This is because the adversarial loss of a predictor $f$ at point $\mathbf{x}$ is inherently not just a function of $f(\mathbf{x})$ but also of its values around a neighborhood of $\mathbf{x}$. As we shall see, the study of consistency is significantly more complex in the adversarial setting, with subtleties that have in fact led to some inaccurate claims made in prior work that we discuss later.

Consistency requires a property of the surrogate and the original losses to hold true for the family of all measurable functions. As argued by Long and Servedio (2013), the notion of $\mathcal{H}$-*consistency*, which requires a similar property for the surrogate and original losses, but with the near or optimal minimizers considered on the restricted hypothesis set $\mathcal{H}$, is a more relevant and desirable property for learning. Long and Servedio (2013) gave examples of surrogate losses that are not $\mathcal{H}$-*consistent* when $\mathcal{H}$ is the class of all measurable functions but that satisfy a *realizable $\mathcal{H}$-consistency* condition when $\mathcal{H}$ is the class of linear functions. More recently, Zhang and Agarwal (2020) studied the notion of *improper realizable $\mathcal{H}$-consistency* of linear classes where the surrogate $\phi$ can be optimized over a larger class, such as that of piecewise linear functions. Note that these studies only deal with the standard $0/1$ classification loss. This motivates our main objective: an extensive study of the $\mathcal{H}$-*consistency* of adversarial surrogate losses, which is critical to the design of robust algorithms with guarantees in this setting.

A more convenient notion in the study of $\mathcal{H}$-*consistency* is that of $\mathcal{H}$-*calibration*, which is a related notion that involves conditioning on the input point. $\mathcal{H}$-*calibration* often is a sufficient condition for $\mathcal{H}$-*consistency* in the standard classification settings (Steinwart, 2007). However, the adversarial loss presents new challenges and requires carefully distinguishing among these notions to avoid drawing false conclusions. As an example, the recent COLT 2020 paper of Bao et al. (2020) presents a study of $\mathcal{H}$-*calibration* for the adversarial loss in the special case where $\mathcal{H}$ is the class of linear functions. However, several comments are due regarding that work. See a detailed discussion in Appendix B.

**Our Contributions.** We present a more systematic study of the $\mathcal{H}$-calibration and $\mathcal{H}$-consistency including for the case where $\mathcal{H} = \mathcal{H}_{\text{all}}$ of adversarial surrogate losses. In Section 4, we give a detailed analysis of the $\mathcal{H}$-*calibration* properties of several natural surrogate losses. We present a series of new negative results showing that, under some general assumptions, convex loss functions and *supremum-based convex losses*, that are loss functions defined as the supremum over a ball of a convex function, which are those commonly used in applications, are not $\mathcal{H}$-*calibrated* for common hypothesis sets used in machine learning. Next, we give a characterization of calibration and prove that a family of proposed surrogates are $\mathcal{H}$-*calibrated*, with common hypothesis sets. These fix previous calibration results presented for the family of linear models in (Bao et al., 2020) and significantly generalize the results to the nonlinear hypothesis sets. In Section 5, we study the $\mathcal{H}$-*consistency* of surrogate loss functions. We prove that, in the absence of distributional assumptions, many surrogate losses shown to be $\mathcal{H}$-*calibrated* in Section 4 are in fact not $\mathcal{H}$-*consistent*. This, in particular, proves that a claim presented in a COLT 2020 publication is inaccurate. Next, in contrast, we show that when the minimum of the surrogate loss is achieved within $\mathcal{H}$, under some general conditions, the $\rho$-margin ramp loss (see, for example, (Mohri et al., 2018)) is $\mathcal{H}$-*consistent* for $\mathcal{H}$ being the linear hypothesis set, or any non-decreasing and continuous $g$-based hypothesis set, including the ReLU-based hypothesis set. We then give similar $\mathcal{H}$-*consistency* guarantees for supremum-based surrogate losses based on a non-increasing auxiliary function, including the calibrated supremum-based $\rho$-margin ramp loss when $\mathcal{H}$ is any symmetric hypothesis set, e.g., the multi-layer neural networks. In Section 6, we further report a series of empirical results on simulated data, which show that many $\mathcal{H}$-*calibrated* surrogate losses are indeed not $\mathcal{H}$-*consistent*, and justify our conditions for consistency. Overall, our results imply that the loss functions commonly used in practice for optimizing the adversarial loss are not $\mathcal{H}$-consistent and that minimizing such losses may not lead to a more favorable adversarial loss. This could be in fact the reason why the empirical results reported in the literature have not been

favorable. Instead, we suggest alternative surrogate losses that we prove are $\mathcal{H}$-consistent and that can be useful to the design of effective algorithms.

We give a detailed discussion of related work in Appendix A. We start with basic concepts of calibration and consistency (Section 2) and an introduction of robust classification (Section 3).

## 2 Preliminaries

We will denote vectors as lowercase bold letters (e.g. $\mathbf{x}$). The $d$-dimensional $l_2$-ball with radius $r$ is denoted by $B_2^d(r) := \{\mathbf{z} \in \mathbb{R}^d \mid \|\mathbf{z}\|_2 \leq r\}$. We denote by $\mathcal{X}$ the set of all possible examples. $\mathcal{X}$ is also sometimes referred to as the input space. The set of all possible labels is denoted by $\mathcal{Y}$. We will limit ourselves to the case of binary classification where $\mathcal{Y} = \{-1, +1\}$. Let $\mathcal{H}$ be a family of functions from $\mathbb{R}^d$ to $\mathbb{R}$. Given a fixed but unknown distribution $\mathcal{P}$ over $\mathcal{X} \times \mathcal{Y}$, the binary classification learning problem is then formulated as follows. The learner seeks to select a predictor $f \in \mathcal{H}$ with small *generalization error* with respect to the distribution $\mathcal{P}$. The *generalization error* of a classifier $f \in \mathcal{H}$ is defined by $\mathcal{R}_{\ell_0}(f) = \mathbb{E}_{(\mathbf{x},y) \sim \mathcal{P}}[\ell_0(f, \mathbf{x}, y)]$, where $\ell_0(f, \mathbf{x}, y) = \mathbb{1}_{yf(\mathbf{x}) \leq 0}$ is the standard 0/1 loss. More generally, the *$\ell$-risk* of a classifier $f$ for a surrogate loss $\ell(f, \mathbf{x}, y)$ is defined by

$$\mathcal{R}_\ell(f) = \mathop{\mathbb{E}}_{(\mathbf{x},y) \sim \mathcal{P}}[\ell(f, \mathbf{x}, y)]. \tag{1}$$

Moreover, the *minimal $(\ell, \mathcal{H})$-risk*, which is also called the *Bayes $(\ell, \mathcal{H})$-risk*, is defined by $\mathcal{R}_{\ell,\mathcal{H}}^* = \inf_{f \in \mathcal{H}} \mathcal{R}_\ell(f)$. In the standard classification setting, the goal of a consistency analysis is to determine whether the minimization of a surrogate loss $\ell$ can lead to that of the binary loss generalization error. Similarly, in adversarially robust classification, the goal of a consistency analysis is to determine if the minimization of a surrogate loss $\ell$ yields that of the *adversarial generalization error* defined by $\mathcal{R}_{\ell_\gamma}(f) = \mathbb{E}_{(\mathbf{x},y) \sim \mathcal{P}}[\ell_\gamma(f, \mathbf{x}, y)]$, where

$$\ell_\gamma(f, \mathbf{x}, y) := \sup_{\mathbf{x}':\|\mathbf{x} - \mathbf{x}'\| \leq \gamma} \mathbb{1}_{yf(\mathbf{x}') \leq 0} \tag{2}$$

is the *adversarial 0/1 loss*. This motivates the definition of $\mathcal{H}$-*consistency*.

**Definition 1** ($\mathcal{H}$-Consistency). *Given a hypothesis set $\mathcal{H}$, we say that a loss function $\ell_1$ is $\mathcal{H}$-consistent with respect to a loss function $\ell_2$, if the following holds:*

$$\mathcal{R}_{\ell_1}(f_n) - \mathcal{R}_{\ell_1,\mathcal{H}}^* \xrightarrow{n \to +\infty} 0 \implies \mathcal{R}_{\ell_2}(f_n) - \mathcal{R}_{\ell_2,\mathcal{H}}^* \xrightarrow{n \to +\infty} 0, \tag{3}$$

*for all probability distributions and sequences of $\{f_n\}_{n \in \mathbb{N}} \subset \mathcal{H}$.*

For a distribution $\mathcal{P}$ over $\mathcal{X} \times \mathcal{Y}$ with random variables $X$ and $Y$, let $\eta_\mathcal{P}: \mathcal{X} \to [0, 1]$ be a measurable function such that, for any $\mathbf{x} \in \mathcal{X}$, $\eta_\mathcal{P}(\mathbf{x}) = \mathcal{P}(Y = 1 \mid X = \mathbf{x})$. By the property of conditional expectation, we can rewrite (1) as $\mathcal{R}_\ell(f) = \mathbb{E}_X[\mathcal{C}_\ell(f, \mathbf{x}, \eta_\mathcal{P}(\mathbf{x}))]$, where $\mathcal{C}_\ell(f, \mathbf{x}, \eta)$ is the *inner $\ell$-risk* defined as followed:

$$\forall \mathbf{x} \in \mathcal{X}, \forall \eta \in [0, 1], \quad \mathcal{C}_\ell(f, \mathbf{x}, \eta) := \eta \ell(f, \mathbf{x}, +1) + (1 - \eta)\ell(f, \mathbf{x}, -1). \tag{4}$$

Moreover, the *minimal inner $\ell$-risk* on $\mathcal{H}$ is denoted by $\mathcal{C}_{\ell,\mathcal{H}}^*(\mathbf{x}, \eta) := \inf_{f \in \mathcal{H}} \mathcal{C}_\ell(f, \mathbf{x}, \eta)$. For a margin-based loss $\phi$, the *generic conditional $\phi$-risk* is $\bar{\mathcal{C}}_\phi(t, \eta) := \eta \phi(t) + (1 - \eta)\phi(-t)$ for any $\eta \in [0, 1]$ and $t \in \mathbb{R}$ (Bartlett et al., 2006). The notion of *calibration* for the inner risk is often a powerful tool for the analysis of $\mathcal{H}$-consistency (Steinwart, 2007).

**Definition 2** ($\mathcal{H}$-Calibration). *[Definition 2.7 in (Steinwart, 2007)] Given a hypothesis set $\mathcal{H}$, we say that a loss function $\ell_1$ is $\mathcal{H}$-calibrated with respect to a loss function $\ell_2$ if, for any $\epsilon > 0$, $\eta \in [0, 1]$, and $\mathbf{x} \in \mathcal{X}$, there exists $\delta > 0$ such that for all $f \in \mathcal{H}$ we have*

$$\mathcal{C}_{\ell_1}(f, \mathbf{x}, \eta) < \mathcal{C}_{\ell_1,\mathcal{H}}^*(\mathbf{x}, \eta) + \delta \implies \mathcal{C}_{\ell_2}(f, \mathbf{x}, \eta) < \mathcal{C}_{\ell_2,\mathcal{H}}^*(\mathbf{x}, \eta) + \epsilon. \tag{5}$$

Steinwart (2007) points out that if $\ell_1$ is $\mathcal{H}$-calibrated wrt $\ell_2$, then $\mathcal{H}$-consistency, that is condition (3), holds for any probability distribution verifying the additional condition of *minimizability* (Steinwart, 2007, Definition 2.4). Next, we introduce the notions of *calibration function* from (Steinwart, 2007).

**Definition 3** (Calibration function). *Given a hypothesis set $\mathcal{H}$, we define the* calibration function *$\delta_{\max}$ for a pair of losses $(\ell_1, \ell_2)$ as follows: for all $\mathbf{x} \in \mathcal{X}$, $\eta \in [0, 1]$ and $\epsilon > 0$,*

$$\delta_{\max}(\epsilon, \mathbf{x}, \eta) = \inf_{f \in \mathcal{H}} \left\{ \mathcal{C}_{\ell_1}(f, \mathbf{x}, \eta) - \mathcal{C}_{\ell_1,\mathcal{H}}^*(\mathbf{x}, \eta) \mid \mathcal{C}_{\ell_2}(f, \mathbf{x}, \eta) - \mathcal{C}_{\ell_2,\mathcal{H}}^*(\mathbf{x}, \eta) \geq \epsilon \right\}. \tag{6}$$

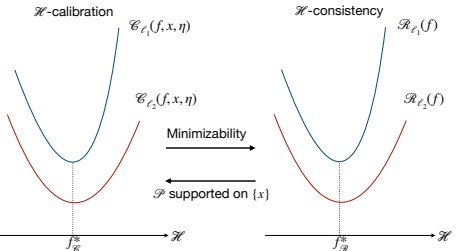

Figure 1: Illustration of $\mathcal{H}$-calibration and $\mathcal{H}$-consistency. Left: $\mathcal{H}$-calibration, for any $\mathbf{x} \in \mathcal{X}$, minimization of $\mathcal{C}_{\ell_1}(f, \mathbf{x})$ can lead to that of $\mathcal{C}_{\ell_2}(f, \mathbf{x})$. Right: $\mathcal{H}$-consistency, minimization of $\mathcal{R}_{\ell_1}(f)$ can lead to that of $\mathcal{R}_{\ell_2}(f)$. $\mathcal{H}$-consistency reduces to $\mathcal{H}$-calibration when the support of underlying distribution $\mathcal{P}$ is the single point set $\{\mathbf{x}\} \subset \mathcal{X}$; Under the minimizability condition, $\mathcal{H}$-calibration would imply $\mathcal{H}$-consistency.

For any $\mathbf{x} \in \mathcal{X}$, $\eta \in [0, 1]$ and $\epsilon > 0$, the calibration function gives the maximal $\delta$ satisfying the calibration condition (5). The following proposition is an important result from (Steinwart, 2007).

**Proposition 4** (Lemma 2.9 in (Steinwart, 2007)). *Given a hypothesis set $\mathcal{H}$, loss $\ell_1$ is $\mathcal{H}$-calibrated with respect to $\ell_2$ if and only if its calibration function $\delta_{\max}$ satisfies $\delta_{\max}(\epsilon, \mathbf{x}, \eta) > 0$ for all $\mathbf{x} \in \mathcal{X}$, $\eta \in [0, 1]$ and $\epsilon > 0$.*

Since the concepts of calibration and consistency may not be familiar to readers without an extensive background in this area, we further comment on these notions before presenting our main results. Informally, a loss function is $\mathcal{H}$-consistent if minimizing it results in a classifier whose generalization error is close to the minimal generalization error within $\mathcal{H}$. Similarly, a loss function is $\mathcal{H}$-calibrated if minimizing it results in a classifier whose inner $\ell_2$-risk is close to the minimal inner $\ell_2$-risk within $\mathcal{H}$ for each $\mathbf{x} \in \mathcal{X}$. $\mathcal{H}$-calibration (5) is a necessary condition for $\mathcal{H}$-consistency (3), but is not always sufficient. As an example, we show in Section 5.1 that $\mathcal{H}$-calibrated surrogate losses proposed in (Bao et al., 2020) are not $\mathcal{H}$-consistent. For this reason, $\mathcal{H}$-consistency, that is consistency for a particular hypothesis set $\mathcal{H}$, is a difficult problem even in standard non-adversarial scenarios. When $\mathcal{H}$ is the family of all measurable functions, the notions of calibration and consistency with respect to the $0/1$ loss have been widely studied in the literature to analyze the properties of margin-based losses (Zhang, 2004; Bartlett et al., 2006). In this special case, calibration implies consistency. Steinwart (2007) further establishes a sufficient condition called minimizability under which $\mathcal{H}$-calibration (5) implies $\mathcal{H}$-consistency (3). Note that the minimizability condition holds in (Zhang, 2004; Bartlett et al., 2006). However, it does not hold in general in the adversarial scenario and thus analyzing $\mathcal{H}$-consistency becomes much harder. To the best of our knowledge, our work is the first to prove $\mathcal{H}$-consistency results for general hypothesis sets $\mathcal{H}$, including for the case where $\mathcal{H} = \mathcal{H}_{\text{all}}$, in the context of adversarial classification. We conclude this section with an illustration of the connection between the notations of calibration and consistency in Figure 1.

# 3 Adversarially Robust Classification

In adversarially robust classification, the loss at $(\mathbf{x}, y)$ is measured in terms of the worst loss incurred over an adversarial perturbation of $\mathbf{x}$ within a ball of a certain radius in a norm. For simplicity, we will consider perturbations in the $l_2$ norm $\|\cdot\|$.[1] We will denote by $\gamma$ the maximum magnitude of the allowed perturbations. Given $\gamma > 0$, a data point $(\mathbf{x}, y)$, a function $f \in \mathcal{H}$, and a margin-based loss $\phi : \mathbb{R} \to \mathbb{R}_+$, we define the *adversarial loss* of $f$ at $(\mathbf{x}, y)$ as

$$\tilde{\phi}(f, \mathbf{x}, y) = \sup_{\mathbf{x}': \|\mathbf{x} - \mathbf{x}'\| \leq \gamma} \phi(y f(\mathbf{x}')). \tag{7}$$

The above naturally motivates *supremum-based* surrogate losses that are commonly used to optimize the adversarial $0/1$ loss (Goodfellow et al., 2014; Madry et al., 2017; Shafahi et al., 2019; Wong et al.,

---

[1]Our analysis in the paper can be extended directly to other perturbations such as the $l_1$ ball or $l_\infty$ ball, and in fact for any $l_p$ norm for $p \in [1, \infty]$. In particular, the proofs of our calibration and consistency results for general hypothesis sets (e.g., Theorem 6, Theorem 7, Theorem 10, Theorem 16, Theorem 20, Theorem 23, Theorem 24) do not require the norm being $l_2$ and work for other norms too.

2020). We say that a surrogate loss $\tilde{\phi}(f, \mathbf{x}, y)$ is *supremum-based* if it is of the form defined in (7). We say that the supremum-based surrogate is convex if the function $\phi$ in (7) is convex. When $\phi$ is non-increasing, the following equality holds (Yin et al., 2019):

$$\sup_{\mathbf{x}':\|\mathbf{x}-\mathbf{x}'\|\leq\gamma} \phi(yf(\mathbf{x}')) = \phi\left(\inf_{\mathbf{x}':\|\mathbf{x}-\mathbf{x}'\|\leq\gamma} yf(\mathbf{x}')\right). \tag{8}$$

The adversarial $0/1$ loss defined in (2) is a special case of (7), where $\phi$ is the $0/1$ loss, that is, $\phi(yf(\mathbf{x})) = \ell_0(f, \mathbf{x}, y) = \mathbb{1}_{yf(\mathbf{x})\leq 0}$. Therefore, the adversarial $0/1$ loss has the equivalent form

$$\ell_\gamma(f, \mathbf{x}, y) = \sup_{\mathbf{x}':\|\mathbf{x}-\mathbf{x}'\|\leq\gamma} \mathbb{1}_{yf(\mathbf{x}')\leq 0} = \mathbb{1}_{\inf_{\mathbf{x}':\|\mathbf{x}-\mathbf{x}'\|\leq\gamma} yf(\mathbf{x}')\leq 0}. \tag{9}$$

This alternative equivalent form of adversarial $0/1$ loss is more advantageous to analyze than (2) and would be adopted in our proofs. Without loss of generality, let $\mathcal{X} = B_2^d(1)$ and $\gamma \in (0, 1)$. In this paper, we aim to characterize surrogate losses $\ell_1$ satisfying $\mathcal{H}$-consistency (3) and $\mathcal{H}$-calibration (5) with $\ell_2 = \ell_\gamma$ and for the hypothesis sets $\mathcal{H}$ which are *regular for adversarial calibration*.

**Definition 5** (Regularity for Adversarial Calibration). *We say that a hypothesis set $\mathcal{H}$ is* regular for adversarial calibration *if there exists a* distinguishing $\mathbf{x}$ *in* $\mathcal{X}$, *that is if there exist $f, g \in \mathcal{H}$ such that* $\inf_{\|\mathbf{x}'-\mathbf{x}\|\leq\gamma} f(\mathbf{x}') > 0$ *and* $\sup_{\|\mathbf{x}'-\mathbf{x}\|\leq\gamma} g(\mathbf{x}') < 0$.

When studying $\mathcal{H}$-calibration of surrogate losses, it suffices to study sets $\mathcal{H}$ that are regular for adversarial calibration not only because all common hypothesis sets admit the property, but also because of the following result. (See Appendix E.1 for the proof.) We say that a hypothesis set $\mathcal{H}$ is *symmetric*, if for any $f \in \mathcal{H}$, $-f$ is also in $\mathcal{H}$.

**Theorem 6.** *Let $\mathcal{H}$ be a symmetric hypothesis set. If $\mathcal{H}$ is not regular for adversarial calibration, then any surrogate loss $\ell$ is $\mathcal{H}$-calibrated with respect to $\ell_\gamma$.*

Moreover, we specifically study the following hypothesis sets that are regular for adversarial calibration: *linear models*: $\mathcal{H}_{\mathrm{lin}} = \{\mathbf{x} \to \mathbf{w} \cdot \mathbf{x} \mid \|\mathbf{w}\| = 1\}$, as in (Bao et al., 2020); *generalized linear models*: $\mathcal{H}_g = \{\mathbf{x} \to g(\mathbf{w} \cdot \mathbf{x}) + b \mid \|\mathbf{w}\| = 1, |b| \leq G\}$ where $g$ is a non-decreasing function; *the family of all measurable functions*: $\mathcal{H}_{\mathrm{all}}$; and *multi-layer neural networks*: $\mathcal{H}_{\mathrm{NN}} = \{\mathbf{x} \to \mathbf{u} \cdot \rho_n(\mathbf{W}_n(\cdots\rho_2(\mathbf{W}_2\rho_1(\mathbf{W}_1\mathbf{x} + \mathbf{b}_1) + \mathbf{b}_2)\cdots) + b_n) \mid \|\mathbf{u}\|_1 \leq \Lambda, \|\mathbf{W}_j\| \leq W, \|\mathbf{b}_j\|_1 \leq B\}$, where $\rho_j$ is an activation function; In the special case of $g = (\cdot)_+ = \max(\cdot, 0)$, we denote the corresponding *ReLU-based hypothesis set* by $\mathcal{H}_{\mathrm{relu}} = \{\mathbf{x} \to (\mathbf{w} \cdot \mathbf{x})_+ + b \mid \|\mathbf{w}\| = 1, |b| \leq G\}$.

# 4   $\mathcal{H}$-Calibration

Calibration is a condition that often guarantees consistency and is a first step in analyzing surrogate losses. Thus, in this section, we first present a detailed study of the calibration properties of several loss functions. We first give a series of negative results showing that, under general assumptions, convex losses and supremum-based convex losses, which are typically used in practice for adversarial robustness, are not calibrated. We then complement these results with positive ones by identifying a family of losses that are indeed calibrated under certain general conditions.

## 4.1   Negative results: convex losses

We first study convex losses, which are often used for standard binary classification problems.

**Theorem 7.** *Assume $\mathcal{H}$ is such that there exists a distinguishing $\mathbf{x}_0 \in \mathcal{X}$ and $f_0 \in \mathcal{H}$ such that $f_0(\mathbf{x}_0) = 0$. If a margin-based loss $\phi\colon\mathbb{R} \to \mathbb{R}_+$ is convex, then it is not $\mathcal{H}$-calibrated with respect to $\ell_\gamma$.*

In particular, the assumption holds when $\mathcal{H}$ is regular for adversarial calibration and contains 0. By Theorem 7, we obtain the following corollary, which fixes the main negative result of Bao et al. (2020) and generalizes the result to nonlinear hypothesis sets. Note $\mathcal{H}_{\mathrm{lin}}$, $\mathcal{H}_{\mathrm{NN}}$ and $\mathcal{H}_{\mathrm{all}}$ all satisfy there exists a distinguishing $\mathbf{x}_0 \in \mathcal{X}$ and $f_0 \in \mathcal{H}$ such that $f_0(\mathbf{x}_0) = 0$. When $g(-\gamma) + G > 0$ and $g(\gamma) - G < 0$, $\mathcal{H}_g$ also satisfies this assumption. Verifying this condition on $\mathcal{H}_g$ is straightforward for $G$ sufficiently large.

**Corollary 8.** *If a margin-based loss $\phi \colon \mathbb{R} \to \mathbb{R}_+$ is convex, then $\phi$ is not $\mathcal{H}$-calibrated with respect to $\ell_\gamma$, for $\mathcal{H} = \mathcal{H}_{\text{lin}}$, $\mathcal{H}_g$ with a non-decreasing and continuous function $g$ such that $g(-\gamma) + G > 0$ and $g(\gamma) - G < 0$, $\mathcal{H}_{\text{relu}}$ with $G > \gamma$, $\mathcal{H}_{\text{NN}}$, and $\mathcal{H}_{\text{all}}$.*

While convex surrogates are natural for the $0/1$ loss, the current practice in designing practical algorithms for the adversarial loss involves using convex supremum-based surrogates (Madry et al., 2017; Wong et al., 2020; Shafahi et al., 2019). We next investigate such losses.

## 4.2 Negative results: supremum-based convex losses

We study losses of the type $\tilde{\phi}(f, \mathbf{x}, y) = \sup_{\mathbf{x}' \colon \|\mathbf{x} - \mathbf{x}'\| \leq \gamma} \phi(y f(\mathbf{x}'))$, with $\phi$ convex, which are often used in practice as surrogates for the adversarial $0/1$ loss. The following theorems presents negative results for supremum-based convex surrogate losses for the common hypothesis sets $\mathcal{H}$.

**Theorem 9.** *Let $\phi$ be a convex and non-increasing margin-based loss. Consider the surrogate loss defined by $\tilde{\phi}(f, \mathbf{x}, y) = \sup_{\mathbf{x}' \colon \|\mathbf{x} - \mathbf{x}'\| \leq \gamma} \phi(y f(\mathbf{x}'))$. Then $\tilde{\phi}$ is not $\mathcal{H}$-calibrated with respect to $\ell_\gamma$, for $\mathcal{H} = \mathcal{H}_{\text{lin}}$, $\mathcal{H}_g$ with a non-decreasing and continuous function $g$ such that $g(-\gamma) + G > 0$ and $g(\gamma) - G < 0$, and $\mathcal{H}_{\text{relu}}$ with $G > \gamma$.*

**Theorem 10.** *Let $\mathcal{H}$ be a hypothesis set containing $0$ that is regular for adversarial calibration. If a margin-based loss $\phi$ is convex and non-increasing, then the surrogate loss defined by $\tilde{\phi}(f, \mathbf{x}, y) = \sup_{\mathbf{x}' \colon \|\mathbf{x} - \mathbf{x}'\| \leq \gamma} \phi(y f(\mathbf{x}'))$ is not $\mathcal{H}$-calibrated with respect to $\ell_\gamma$.*

The theorems above provides evidence that the current practice of making networks adversarially robust via minimizing convex supremum-based surrogates may have serious deficiencies. This may also explain why in practice the adversarial accuracies that are achievable are much lower than the corresponding natural accuracies of the model (Madry et al., 2017). In general, optimizing non-calibrated or non-consistent surrogates could lead to undesirable solutions even under strong assumptions (such as the Bayes risk being zero). See Section 6, where we empirically demonstrate this in a variety of settings. By Theorem 10 and the fact that $\mathcal{H}_{\text{NN}}$ and $\mathcal{H}_{\text{all}}$ both contain $0$ and are regular for adversarial calibration, we can derive the following corollary.

**Corollary 11.** *Let $\phi$ be a convex and non-increasing margin-based loss. Consider the surrogate loss defined by $\tilde{\phi}(f, \mathbf{x}, y) = \sup_{\mathbf{x}' \colon \|\mathbf{x} - \mathbf{x}'\| \leq \gamma} \phi(y f(\mathbf{x}'))$. Then $\tilde{\phi}$ is not $\mathcal{H}$-calibrated with respect to $\ell_\gamma$, for $\mathcal{H} = \mathcal{H}_{\text{NN}}$, and $\mathcal{H} = \mathcal{H}_{\text{all}}$.*

The proofs of Theorem 7, Theorem 9 and Theorem 10 are included in Appendix E.2. The key in proving the above theorems is to analyze the calibration function $\delta_{\max}(\epsilon, \mathbf{x}, \eta)$ as defined in (6) of losses $(\ell, \ell_\gamma)$ at $\eta = \frac{1}{2}$, $\epsilon = \frac{1}{2}$ and distinguishing $\mathbf{x}_0 \in \mathcal{X}$. Naturally, this requires us to understand the inner risk $\mathcal{C}_\ell(f, \mathbf{x}, \eta)$ that in turn depends on the worst case perturbation of a given data point according to $\ell$. Our key insight (Lemma 25) is that $\delta_{\max}(\epsilon, \mathbf{x}_0, \eta)$ can be characterized by two quantities $\underline{M}(f, \mathbf{x}_0, \gamma) = \inf_{\mathbf{x}' \colon \|\mathbf{x}_0 - \mathbf{x}'\| \leq \gamma} f(\mathbf{x}')$, $\overline{M}(f, \mathbf{x}_0, \gamma) = \sup_{\mathbf{x}' \colon \|\mathbf{x}_0 - \mathbf{x}'\| \leq \gamma} f(\mathbf{x}')$. Requiring $\delta_{\max}\left(\frac{1}{2}, \mathbf{x}_0, \frac{1}{2}\right) > 0$ corresponds to an appropriate convex function not achieving a minimum in a set that has global optimum, thereby reaching a contradiction.

## 4.3 Positive results

In this section, we aim to provide alternative losses which could be calibrated with respect to $\ell_\gamma$.

### 4.3.1 Characterization

In light of the above negative results, we need to consider non-convex surrogates. One possible candidate is the family of losses introduced by Bao et al. (2020) that satisfy the property that the generic conditional $\phi$-risk $\bar{\mathcal{C}}_\phi(t, \eta)$ is quasi-concave in $t \in \mathbb{R}$ for all $\eta \in [0, 1]$. Theorem 12 below is a correction to the main positive result, Theorem 11 in (Bao et al., 2020), where we prove the theorem under the correct calibration definition.

**Theorem 12.** *Let a margin-based loss $\phi$ be bounded, continuous, non-increasing, and satisfy the property that $\bar{\mathcal{C}}_\phi(t, \eta)$ is quasi-concave in $t \in \mathbb{R}$ for all $\eta \in [0, 1]$. Assume that $\phi(-t) > \phi(t)$ for any $\gamma < t \leq 1$. Then $\phi$ is $\mathcal{H}_{\text{lin}}$-calibrated with respect to $\ell_\gamma$ if and only if for any $\gamma < t \leq 1$,*

$$\phi(\gamma) + \phi(-\gamma) > \phi(t) + \phi(-t). \tag{10}$$

The proof of Theorem 12 is included in Appendix E.4, where we make use of Lemma 27 and Lemma 28, which are powerful since they apply to any symmetric hypothesis sets. These lemmas would be used for proving more general positive results, as we will show later. Note Theorem 11 in (Bao et al., 2020) does not hold any more under the correct calibration Definition 2, since their condition $\phi(\gamma) + \phi(-\gamma) > \phi(1) + \phi(-1)$ is much weaker than (10).

The following theorem extends the above to show that under certain conditions, such surrogate losses are $\mathcal{H}$-calibrated for the class of generalized linear models with respect to the adversarial $0/1$ loss.

**Theorem 13.** *Let $g$ be a non-decreasing and continuous function such that $g(1 + \gamma) < G$ and $g(-1 - \gamma) > -G$ for some $G \geq 0$. Let a margin-based loss $\phi$ be bounded, continuous, non-increasing, and satisfy the property that $\bar{\mathcal{C}}_\phi(t, \eta)$ is quasi-concave in $t \in \mathbb{R}$ for all $\eta \in [0, 1]$. Assume that $\phi(g(-t) - G) > \phi(G - g(-t))$ and $g(-t) + g(t) \geq 0$ for any $0 \leq t \leq 1$. Then $\phi$ is $\mathcal{H}_g$-calibrated with respect to $\ell_\gamma$ if and only if for any $0 \leq t \leq 1$,*

$$\phi(G - g(-t)) + \phi(g(-t) - G) = \phi(g(t) + G) + \phi(-g(t) - G)$$

*and* $\quad \min\{\phi(\overline{A}(t)) + \phi(-\overline{A}(t)), \phi(\underline{A}(t)) + \phi(-\underline{A}(t))\} > \phi(G - g(-t)) + \phi(g(-t) - G),$

*where $\overline{A}(t) = \max_{s \in [-t,t]} g(s) - g(s - \gamma)$ and $\underline{A}(t) = \min_{s \in [-t,t]} g(s) - g(s + \gamma)$.*

See Appendix E.5 for the proof. The conditions in the theorem above are necessary and sufficient and thus characterize calibration for such surrogate losses. To interpret the conditions better, consider ReLU functions. In that case, the assumptions can be further simplified to get the following corollary.

**Corollary 14.** *Assume that $G > 1 + \gamma$. Let a margin-based loss $\phi$ be bounded, continuous, non-increasing, and satisfy the property that $\bar{\mathcal{C}}_\phi(t, \eta)$ is quasi-concave in $t \in \mathbb{R}$ for all $\eta \in [0, 1]$. Assume that $\phi(-G) > \phi(G)$. Then $\phi$ is $\mathcal{H}_{\mathrm{relu}}$-calibrated with respect to $\ell_\gamma$ if and only if for any $0 \leq t \leq 1$,*

$$\phi(G) + \phi(-G) = \phi(t + G) + \phi(-t - G) \quad \text{and} \quad \phi(\gamma) + \phi(-\gamma) > \phi(G) + \phi(-G).$$

### 4.3.2 Calibration

To demonstrate the applicability of Theorem 13, we consider a specific surrogate loss namely the $\rho$-*margin loss* $\phi_\rho(t) := \min\{1, \max\{0, 1 - \frac{t}{\rho}\}\}$, $\rho > 0$, which is a generalization of the ramp loss (see, (Mohri et al., 2018)). We also define its supremum-based counterpart as $\tilde{\phi}_\rho(f, \mathbf{x}, y) := \sup_{\mathbf{x}': \|\mathbf{x} - \mathbf{x}'\| \leq \gamma} \phi_\rho(y f(\mathbf{x}'))$. Using Theorem 12, Theorem 13 and Corollary 14 in Section 4.3.1, we can conclude that the $\rho$-margin loss is calibrated under reasonable conditions for linear hypothesis sets and non-decreasing $g$-based hypothesis sets, since $\phi_\rho(t)$ is bounded, continuous, non-increasing, and satisfies $\bar{\mathcal{C}}_{\phi_\rho}(t, \eta)$ is quasi-concave in $t \in \mathbb{R}$ for all $\eta \in [0, 1]$. This is stated formally below.

**Theorem 15.** *The surrogate $\phi_\rho$ is $\mathcal{H}_{\mathrm{lin}}$-calibrated with respect to $\ell_\gamma$ if and only if $\rho > \gamma$. Given a non-decreasing and continuous function $g$ such that $g(1+\gamma) < G$ and $g(-1-\gamma) > -G$ for some $G \geq 0$, assume that $g(-t)+g(t) \geq 0$ for any $0 \leq t \leq 1$, then $\phi_\rho$ is $\mathcal{H}_g$-calibrated with respect to $\ell_\gamma$ if and only if for any $0 \leq t \leq 1$, $\phi_\rho(G - g(-t)) = \phi_\rho(g(t)+G)$ and $\min\{\phi_\rho(\overline{A}(t)), \phi_\rho(-\underline{A}(t))\} > \phi_\rho(G - g(-t))$, where $\overline{A}(t) = \max_{s \in [-t,t]} g(s) - g(s - \gamma)$ and $\underline{A}(t) = \min_{s \in [-t,t]} g(s) - g(s + \gamma)$. Assume that $G > 1 + \gamma$, then $\phi_\rho$ is $\mathcal{H}_{\mathrm{relu}}$-calibrated with respect to $\ell_\gamma$ if and only if $G \geq \rho > \gamma$.*

Recall that in Theorem 10 we ruled out the possibility of finding $\mathcal{H}$-calibrated supremum-based convex surrogate losses with respect to the adversarial $0/1$ loss. However, we show that the supremum-based $\rho$-margin loss is indeed $\mathcal{H}$-calibrated, where $\mathcal{H}$ is any symmetric hypothesis set.

**Theorem 16.** *Let $\mathcal{H}$ be a symmetric hypothesis set, then $\tilde{\phi}_\rho$ is $\mathcal{H}$-calibrated with respect to $\ell_\gamma$.*

The proof of Theorem 16 is included in Appendix E.4. By Theorem 16 and the fact that $\mathcal{H}_{\mathrm{lin}}$, $\mathcal{H}_{\mathrm{NN}}$ and $\mathcal{H}_{\mathrm{all}}$ are all symmetric, we derive the following.

**Corollary 17.** *$\tilde{\phi}_\rho$ is $\mathcal{H}$-calibrated with respect to $\ell_\gamma$, for $\mathcal{H} = \mathcal{H}_{\mathrm{lin}}$, $\mathcal{H}_{\mathrm{NN}}$, and $\mathcal{H}_{\mathrm{all}}$.*

The results of this section suggest that the $\rho$-margin loss and supremum-based $\rho$-margin loss may be good surrogates for the adversarial $0/1$ loss. However, calibration, in general, is not equivalent to consistency, our eventual goal. In the next section, we study conditions under which we can expect these surrogates losses to be $\mathcal{H}$-consistent as well.

# 5 H-Consistency

In this section, we study the $\mathcal{H}$-consistency of surrogate loss functions. The results of the previous section suggest that convex losses or supremum-based convex losses would not be $\mathcal{H}$-consistent. However, $\mathcal{H}$-calibrated losses, such as the $\rho$-margin loss and supremum-based $\rho$-margin loss present an intriguing possibility. Bao et al. (2020) made a claim that since the losses they proposed are $\mathcal{H}_{\mathrm{lin}}$-calibrated they are also $\mathcal{H}_{\mathrm{lin}}$-consistent. We first present a result that implies this claim is incorrect. In fact, our result stated below shows that without assumptions on the data distribution, no continuous margin based loss or a supremum-based continuous surrogate could be $\mathcal{H}_{\mathrm{lin}}$-consistent.

## 5.1 Negative results

**Theorem 18.** *No continuous margin-based loss function $\phi$ is $\mathcal{H}_{\mathrm{lin}}$-consistent with respect to $\ell_\gamma$. Furthermore, for any continuous and non-increasing margin-based loss $\phi$, surrogates of the form*

$$\tilde{\phi}(f, \mathbf{x}, y) = \sup_{\mathbf{x}': \|\mathbf{x} - \mathbf{x}'\| \le \gamma} \phi(yf(\mathbf{x}'))$$

*are not $\mathcal{H}_{\mathrm{lin}}$-consistent with respect to $\ell_\gamma$.*

The above theorem is proven in Appendix E.6. In particular, Theorem 18 contradicts the $\mathcal{H}$-consistency claim of Bao et al. (2020) for their proposed losses when $\mathcal{H}$ is the family of linear functions. Furthermore, the theorem rules out $\mathcal{H}$-consistency of supremum-based surrogates.

## 5.2 Positive results

In this section, we investigate the nature of the assumptions on the data distributions that may lead to $\mathcal{H}$-consistency of surrogate losses. We take inspiration from the work of Long and Servedio (2013) and Zhang and Agarwal (2020) who study $\mathcal{H}$-consistency for the standard $0/1$ loss. These studies establish consistency under a realizability assumption on the data distribution stated below that requires the Bayes $(\ell_0, \mathcal{H})$-risk to be zero.

**Definition 19** ($\mathcal{H}$-realizability). *A distribution $\mathcal{P}$ over $\mathcal{X} \times \mathcal{Y}$ is $\mathcal{H}$-realizable if it labels points according to a deterministic model in $\mathcal{H}$, i.e., if $\exists f \in \mathcal{H}$ such that $\mathbb{P}_{(\mathbf{x}, y) \sim \mathcal{P}}(\mathrm{sgn}(f(\mathbf{x})) = y) = 1$.*

As with $\mathcal{H}$-realizability, we will assume that, under the data distribution, the Bayes $(\ell_\gamma, \mathcal{H})$-risk is zero. We show that the $\mathcal{H}$-calibrated losses studied in previous sections are $\mathcal{H}$-consistent under natural conditions along with the realizability assumption.

### 5.2.1 Non-supremum-based surrogates

**Theorem 20.** *Let $\mathcal{P}$ be a distribution over $\mathcal{X} \times \mathcal{Y}$ and $\mathcal{H}$ a hypothesis set for which $\mathcal{R}^*_{\ell_\gamma, \mathcal{H}} = 0$. Let $\phi$ be a margin-based loss. If for $\eta \ge 0$, there exists $f^* \in \mathcal{H} \subset \mathcal{H}_{\mathrm{all}}$ such that $\mathcal{R}_\phi(f^*) \le \mathcal{R}^*_{\phi, \mathcal{H}_{\mathrm{all}}} + \eta < +\infty$ and $\phi$ is $\mathcal{H}$-calibrated with respect to $\ell_\gamma$, then for all $\epsilon > 0$ there exists $\delta > 0$ such that for all $f \in \mathcal{H}$,*

$$\mathcal{R}_\phi(f) + \eta < \mathcal{R}^*_{\phi, \mathcal{H}} + \delta \implies \mathcal{R}_{\ell_\gamma}(f) < \mathcal{R}^*_{\ell_\gamma, \mathcal{H}} + \epsilon.$$

For the family of linear models, some convex losses may also be $\mathcal{H}_{\mathrm{lin}}$-consistent verifying the conditions ($\eta = 0$) in Theorem 20. However, $\mathcal{H}_{\mathrm{lin}}$-calibrated losses can be $\mathcal{H}_{\mathrm{lin}}$-consistent under more benign assumptions, where the realizability condition $\mathcal{R}^*_{\ell_\gamma, \mathcal{H}_{\mathrm{lin}}} = 0$ can be further relaxed.

**Theorem 21.** *Let $\mathcal{P}$ be a distribution over $\mathcal{X} \times \mathcal{Y}$. Assume that there exists $g^* \in \mathcal{H}_{\mathrm{lin}}$ such that $\mathcal{R}_{\ell_\gamma}(g^*) = \mathcal{R}^*_{\ell_\gamma, \mathcal{H}_{\mathrm{all}}}$. Let $\phi$ be a margin-based loss. If for $\eta \ge 0$, there exists $f^* \in \mathcal{H}_{\mathrm{lin}} \subset \mathcal{H}_{\mathrm{all}}$ such that $\mathcal{R}_\phi(f^*) \le \mathcal{R}^*_{\phi, \mathcal{H}_{\mathrm{all}}} + \eta < +\infty$ and $\phi$ is $\mathcal{H}_{\mathrm{lin}}$-calibrated with respect to $\ell_\gamma$, then for all $\epsilon > 0$ there exists $\delta > 0$ such that for all $f \in \mathcal{H}_{\mathrm{lin}}$ we have*

$$\mathcal{R}_\phi(f) + \eta < \mathcal{R}^*_{\phi, \mathcal{H}_{\mathrm{lin}}} + \delta \implies \mathcal{R}_{\ell_\gamma}(f) < \mathcal{R}^*_{\ell_\gamma, \mathcal{H}_{\mathrm{lin}}} + \epsilon.$$

The proofs of Theorem 20 and Theorem 21 are presented in Appendix E.7. Using Theorem 15 in Section 4.3.2 and theorems above, we immediately conclude that the calibrated $\rho$-margin loss is consistent with respect to $\ell_\gamma$ for all distributions that satisfy our realizability assumptions.

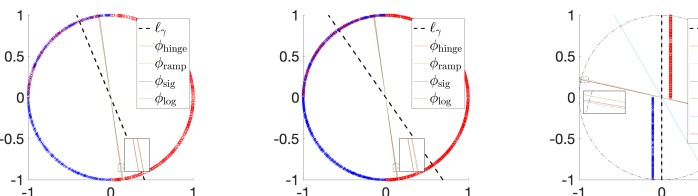

Figure 2: Left: Unit Circle with 1,000 and 2,000 samples. Right: Segment with 5,00 samples.

**Theorem 22.** *If $\rho > \gamma$, then $\phi_\rho$ is $\mathcal{H}_{\mathrm{lin}}$-consistent wrt $\ell_\gamma$ for all distributions such that there exists $g^* \in \mathcal{H}_{\mathrm{lin}}$ with $\mathcal{R}_{\ell_\gamma}(g^*) = \mathcal{R}^*_{\ell_\gamma,\mathcal{H}_{\mathrm{all}}}$ and there exists $f^* \in \mathcal{H}_{\mathrm{lin}}$ such that $\mathcal{R}_{\phi_\rho}(f^*) = \mathcal{R}^*_{\phi_\rho,\mathcal{H}_{\mathrm{all}}}$. If $g$ verifies the calibration condition in Theorem 15, then $\phi_\rho$ is $\mathcal{H}_g$-consistent wrt $\ell_\gamma$ for all distribution $\mathcal{P}$ over $\mathcal{X} \times \mathcal{Y}$ that satisfies $\mathcal{R}^*_{\ell_\gamma,\mathcal{H}_g} = 0$ and there exists $f^* \in \mathcal{H}_g$ such that $\mathcal{R}_{\phi_\rho}(f^*) = \mathcal{R}^*_{\phi_\rho,\mathcal{H}_{\mathrm{all}}}$. If $G > 1 + \gamma$ and $G \geq \rho > \gamma$, then $\phi_\rho$ is $\mathcal{H}_{\mathrm{relu}}$-consistent wrt $\ell_\gamma$ for all distribution $\mathcal{P}$ over $\mathcal{X} \times \mathcal{Y}$ that satisfies $\mathcal{R}^*_{\ell_\gamma,\mathcal{H}_{\mathrm{relu}}} = 0$ and there exists $f^* \in \mathcal{H}_{\mathrm{relu}}$ such that $\mathcal{R}_{\phi_\rho}(f^*) = \mathcal{R}^*_{\phi_\rho,\mathcal{H}_{\mathrm{all}}}$.*

### 5.2.2 Supremum-based surrogates

We can also extend the above to obtain $\mathcal{H}$-consistency of supremum-based convex surrogates. However we need the stronger condition that $\mathcal{R}_\phi$ is minimized exactly inside $\mathcal{H}$.

**Theorem 23.** *Given a distribution $\mathcal{P}$ over $\mathcal{X} \times \mathcal{Y}$ and a hypothesis set $\mathcal{H}$ such that $\mathcal{R}^*_{\ell_\gamma,\mathcal{H}} = 0$. Let $\phi$ be a non-increasing margin-based loss. If there exists $f^* \in \mathcal{H} \subset \mathcal{H}_{\mathrm{all}}$ such that $\mathcal{R}_\phi(f^*) = \mathcal{R}^*_{\phi,\mathcal{H}_{\mathrm{all}}} < +\infty$ and $\tilde{\phi}(f, \mathbf{x}, y) = \sup_{\mathbf{x}':\|\mathbf{x}-\mathbf{x}'\|\leq\gamma} \phi(yf(\mathbf{x}'))$ is $\mathcal{H}$-calibrated with respect to $\ell_\gamma$, then for all $\epsilon > 0$ there exists $\delta > 0$ such that for all $f \in \mathcal{H}$ we have*

$$\mathcal{R}_{\tilde{\phi}}(f) < \mathcal{R}^*_{\tilde{\phi},\mathcal{H}} + \delta \implies \mathcal{R}_{\ell_\gamma}(f) < \mathcal{R}^*_{\ell_\gamma,\mathcal{H}} + \epsilon.$$

The proof of Theorem 23 is presented in Appendix E.7. Again, when combined with Theorem 16 in Section 4.3.2 we conclude that the $\mathcal{H}$-calibrated supremum-based $\rho$-margin loss is also $\mathcal{H}$-consistent with respect to $\ell_\gamma$ for all distributions that satisfy our realizability assumptions.

**Theorem 24.** *Let $\mathcal{H}$ be a symmetric hypothesis set, then $\tilde{\phi}_\rho$ is $\mathcal{H}$-consistent with respect to $\ell_\gamma$ for all distributions $\mathcal{P}$ over $\mathcal{X} \times \mathcal{Y}$ that satisfy: $\mathcal{R}^*_{\ell_\gamma,\mathcal{H}} = 0$ and there exists $f^* \in \mathcal{H}$ such that $\mathcal{R}_{\phi_\rho}(f^*) = \mathcal{R}^*_{\phi_\rho,\mathcal{H}_{\mathrm{all}}} < +\infty$.*

## 6 Experiments

Here, we present experiments on simulated data to support our theoretical findings. The goal is two-fold. First, we empirically demonstrate that indeed $\mathcal{H}$-calibrated surrogates in (Bao et al., 2020) may not be $\mathcal{H}$-consistent unless assumptions on the data distribution are made, even when $\mathcal{H}$ is the class of linear functions. This is consistent with our negative result in Theorem 18 and provides an empirical counterexample to the claim made in (Bao et al., 2020). Second, we study the necessity of the realizability assumptions we adopted in Section 5.2 to establish $\mathcal{H}$-consistency of surrogates satisfying the conditions in Theorem 12.

We generate data points $\mathbf{x} \in \mathbb{R}^2$ on the unit circle and consider $\mathcal{H}$ to be linear models $\mathcal{H}_{\mathrm{lin}}$. We denote $f(\mathbf{x}) = \mathbf{w} \cdot \mathbf{x}$, $\mathbf{w} = (\cos(t), \sin(t))^\top$, $t \in [0, 2\pi)$, $f \in \mathcal{H}_{\mathrm{lin}}$. All risks are approximated by their empirical counterparts computed over $10^7$ i.i.d. samples. To demonstrate the need for some assumptions for $\mathcal{H}$-consistency, we construct a scenario we call the **Unit Circle** case. We consider four surrogates: $\phi_{\mathrm{hinge}}$, $\phi_{\mathrm{ramp}}$, $\phi_{\mathrm{sig}}$ and $\phi_{\mathrm{log}}$ defined in Appendix C.1. In general, we refer all of these surrogates as $\phi_{\mathrm{sur}}$. We generate data points $\mathbf{x}$ from the uniform distribution on the unit circle. Define $\mathbf{x}$ as $\mathbf{x} = (\cos(\theta), \sin(\theta))^\top$, $\theta \in [0, 2\pi)$. Set the label of a point $\mathbf{x}$ as follows: if $\theta \in \left(\frac{\pi}{2}, \pi\right)$, then $y = -1$ with probability $\frac{3}{4}$ and $y = 1$ with probability $\frac{1}{4}$; if $\theta \in \left(0, \frac{\pi}{2}\right)$ or $\left(\frac{3\pi}{2}, 2\pi\right)$, then $y = 1$; if $\theta \in \left(\pi, \frac{3\pi}{2}\right)$, then $y = -1$. Set $\gamma = \frac{\sqrt{2}}{2}$. In this case, the Bayes $(\ell_\gamma, \mathcal{H}_{\mathrm{lin}})$-risk is $\mathcal{R}^*_{\ell_\gamma,\mathcal{H}_{\mathrm{lin}}} \approx 0.5000 \neq 0$ and is achieved by $w_{\ell_\gamma} = (\cos(\theta), \sin(\theta))^\top$ with $\theta \approx 0.7855$. The results obtained by optimizing the different surrogate losses are reported in Table 1(a) and the plots for

Table 1: (a) Unit Circle; (b) Segments.

| $\phi_{\text{sur}}$ | $\mathcal{R}_{\ell_\gamma}(f^*)$ | $\theta_{\phi_{\text{sur}}}$ | $\mathcal{H}_{\text{lin}}$-cal. | $\mathcal{H}_{\text{lin}}$-cons. |
|---|---|---|---|---|
| $\phi_{\text{hinge}}$ | 0.5257 | 0.1420 | ✗ | ✗ |
| $\phi_{\text{ramp}}$ | 0.5263 | 0.1288 | ✓ | ✗ |
| $\phi_{\text{sig}}$ | 0.5261 | 0.1320 | ✓ | ✗ |
| $\phi_{\text{log}}$ | 0.5258 | 0.1414 | ✗ | ✗ |

(a)

| $\phi_{\text{sur}}$ | $\mathcal{R}_{\ell_\gamma}(f^*)$ | $\mathcal{R}_{\phi_{\text{sur}}}(f^*)$ | $\theta_{\phi_{\text{sur}}}$ | $\mathcal{H}_{\text{lin}}$-cal. | $\mathcal{H}_{\text{lin}}$-cons. |
|---|---|---|---|---|---|
| $\phi_{\text{hinge}}$ | 0.0781 | 0.6907 | 1.3548 | ✗ | ✗ |
| $\phi_{\text{ramp}}$ | 0.0781 | 0.3454 | 1.3548 | ✓ | ✗ |
| $\phi_{\text{sig}}$ | 0.0777 | 0.4247 | 1.3498 | ✓ | ✗ |
| $\phi_{\text{log}}$ | 0.0763 | 0.8078 | 1.3341 | ✗ | ✗ |
| $\phi_1$ | 0.0111 | 0 | $\frac{\pi}{6}$ | ✗ | ✗ |
| $\phi_2$ | 0 | 0 | 0 | ✓ | ✓ |

(b)

1,000 samples and 2,000 samples are shown in Figure 2. Table 1(a) shows that neither calibrated nor non-calibrated (convex) surrogates are $\mathcal{H}_{\text{lin}}$-consistent with respect to $\ell_\gamma$ for this distribution. Figure 2 shows that the classifiers obtained by optimizing the four surrogates are almost the same but deviate a lot from the optimal Bayes classifier for $\ell_\gamma$. This shows that indeed calibrated surrogates may not be consistent and contradicts Figure 12 of (Bao et al., 2020). The discrepancy results from an incorrect calculation of the adversarial Bayes risk in (Bao et al., 2020).

Next, we justify the realizability assumptions made in Section 5.2 for obtaining $\mathcal{H}$-consistency of surrogate losses. To do so, we design a scenario that we call the **Segments** case. Here, we consider six surrogates, the four studied above and two more surrogates $\phi_1$ and $\phi_2$ defined in Appendix C.1. The loss $\phi_1$ is a convex loss and $\phi_2$ is the $\rho$-margin ramp loss for some $\rho > \gamma$. In general, we refer to all of these surrogates as $\phi_{\text{sur}}$. We show in Appendix C.2 that $\phi_{\text{hinge}}$, $\phi_{\text{log}}$ and $\phi_1$ are not $\mathcal{H}_{\text{lin}}$-calibrated while $\phi_{\text{ramp}}$, $\phi_{\text{sig}}$ and $\phi_2$ are $\mathcal{H}_{\text{lin}}$-calibrated with respect to $\ell_\gamma$.

Let $I_{\hat\gamma} = \sqrt{1 - \hat\gamma^2}$ and consider: $\mathbb{P}(Y = 1) = \mathbb{P}(Y = -1) = \frac{1}{2}$, and $X \mid Y = 1$ is the uniform distribution on the line segment $\{(\hat\gamma, z) \mid z \in [0, I_{\hat\gamma}]\}$ and $X \mid Y = -1$ is the uniform distribution on the line segment $\{(-\hat\gamma, z) \mid z \in [-I_{\hat\gamma}, 0]\}$ where $\hat\gamma = \gamma + \frac{1-\gamma}{100} = \frac{1+99\gamma}{100}$, $\gamma \in (0, 1)$. We choose $\gamma = 0.1$ and set $\mathbf{w}^* = (1, 0)^\top$. It is easy to check that $\mathbf{w}^*$ achieves the Bayes $(\ell_\gamma, \mathcal{H}_{\text{lin}})$-risk $\mathcal{R}^*_{\ell_\gamma, \mathcal{H}_{\text{lin}}} = 0$. The results for the six different surrogate losses are indicated in Table 1(b) and the plot for 5,00 samples are shown in Figure 2. For $\phi_{\text{hinge}}$, $\phi_{\text{ramp}}$, $\phi_{\text{sig}}$ and $\phi_{\text{log}}$, the Bayes $(\phi_{\text{sur}}, \mathcal{H}_{\text{lin}})$-risk $\mathcal{R}^*_{\phi_{\text{sur}}, \mathcal{H}_{\text{lin}}} \neq 0$. Table 1(b) shows that they are not $\mathcal{H}_{\text{lin}}$-consistent with respect to $\ell_\gamma$.

For $\phi_1$ and $\phi_2$, the Bayes $(\phi_{\text{sur}}, \mathcal{H}_{\text{lin}})$-risk $\mathcal{R}^*_{\phi_{\text{sur}}, \mathcal{H}_{\text{lin}}} = 0$. Table 1(b) shows that $\phi_1$ is not $\mathcal{H}_{\text{lin}}$-consistent (recall that $\phi_1$ is not calibrated) but $\phi_2$ is $\mathcal{H}_{\text{lin}}$-consistent for this distribution. Hence, even when $\mathcal{R}^*_{\ell_\gamma, \mathcal{H}_{\text{lin}}} = 0$, unless a condition is also imposed on $\mathcal{R}^*_{\phi_{\text{sur}}, \mathcal{H}_{\text{lin}}}$, one cannot expect consistency, thereby justifying our realizability assumption. Note that $\mathcal{R}^*_{\phi_{\text{sur}}, \mathcal{H}_{\text{lin}}} = \mathcal{R}^*_{\ell_\gamma, \mathcal{H}_{\text{lin}}} = 0$ is a special case verifying the conditions of Theorem 20 for $\eta = 0$. For this distribution, $\phi_{\text{ramp}}$ is not $\mathcal{H}_{\text{lin}}$-consistent while $\phi_2$ is $\mathcal{H}_{\text{lin}}$-consistent, although both are $\mathcal{H}_{\text{lin}}$-calibrated. We compare them in Figure 3, showing that minimizing $\mathcal{H}_{\text{lin}}$-consistent surrogate $\phi_2$ minimizes the adversarial generalization error for large sample sizes but the same does not hold for non $\mathcal{H}_{\text{lin}}$-consistent surrogate $\phi_{\text{ramp}}$.

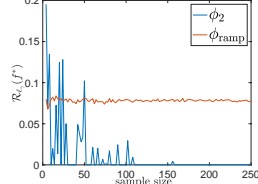

Figure 3: Adv. true risk of consistent and calibrated inconsistent losses vs. sample size.

# 7 Conclusion

We presented a detailed study of calibration and consistency for adversarial robustness. These results can help guide the design of algorithms for learning robust predictors, an increasingly important problem in applications. Our theoretical results show in particular that many of the surrogate losses typically used in practice do not benefit from any guarantee. Our empirical results further illustrate that in the context of a general example. Our results also show that some of the calibration results presented in previous work do not bear any significance, since we prove that in fact they do not guarantee consistency. Instead, we give a series of positive calibration and consistency results for several families of surrogate functions, under some realizability assumptions.

## Acknowledgements

This work was partly funded by NSF CCF-1535987 and NSF IIS-1618662.

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
