# Contents of Appendix

# A  Related Work

The notions of calibration and consistency with respect to the $0/1$ loss have been widely studied in the statistical learning theory literature to analyze the properties of surrogate losses (Zhang, 2004; Bartlett et al., 2006). Bartlett et al. (2006) showed that margin-based convex surrogates, that is mappings of the form $(f, \mathbf{x}, y) \mapsto \phi(yf(\mathbf{x}))$, where $f$ is a real-valued predictor and $\phi\colon \mathbb{R} \to \mathbb{R}_+$ a function differentiable at $0$ with $\phi'(0) < 0$, are calibrated with respect to the class of all measurable functions. Extensions of calibration and consistency to multi-class settings have also been studied (Tewari and Bartlett, 2007). In the special case of the $0/1$ loss and margin-based convex surrogates, calibration immediately implies consistency for the class of all measurable functions. One can then even derive quantitative bounds relating the excess $\phi$-risk to the excess $0/1$ loss of any function $f$ (Zhang, 2004; Bartlett et al., 2006).

The case of adversarial loss is more complex. This is because, in particular, the loss of a predictor $f$ at point $\mathbf{x}$ does not just depend on its value $f(\mathbf{x})$ at that point but also on its values in a neighborhood of $\mathbf{x}$. Steinwart (2007) proposed a general framework to study and characterize calibration and consistency, in particular via a *calibration function*. He also defined a *minimizability* condition under which calibration implies consistency. But, while *minimizability* holds for the $0/1$ loss and margin-based convex surrogates over the class of all measurable functions, the condition does not hold in general for the adversarial loss. Our work borrows tools from the work of Steinwart (2007). However, to establish $\mathcal{H}$-*consistency* in the context of the adversarial loss, additional insights are needed and often stronger assumptions on the data distribution are required. These assumptions are captured in the notion of *realizable $\mathcal{H}$-consistency* that requires that the optimal risk of both the $0/1$ loss and the surrogate loss being achieved inside the class $\mathcal{H}$. Our positive results for $\mathcal{H}$-*consistency* rely on similar but weaker assumptions. Long and Servedio (2013) gave examples of surrogate losses that are not $\mathcal{H}$-*consistent* when $\mathcal{H}$ is the class of all measurable functions but satisfy *realizable $\mathcal{H}$-consistency* when $\mathcal{H}$ is the class of linear functions. Zhang and Agarwal (2020) studied the notion of *improper realizable $\mathcal{H}$-consistency* of linear classes where the surrogate $\phi$ can be optimized over a larger class such as that of piecewise linear functions. The relation between calibration and its implication for consistency has also been explored in ranking problems (Uematsu and Lee, 2011; Gao and Zhou, 2015). In particular, these works show that calibrated surrogate losses for classification problems are not consistent for optimizing ranking losses such as AUC. Hence in these works calibration not implying consistency stems from the mismatch between using a calibrated surrogate for a different loss (classification loss) and applying it for a different purpose. However, in the context of our work the situation is more subtle. Even a calibrated surrogate (with respect to the adversarial 0-1 loss) may not be consistent in general.

These notions of calibration and consistency are relatively unexplored for the adversarial $0/1$ loss. Bao et al. (2020) recently initiated the study of these notions for the adversarial loss. We give a more detailed discussion regarding that work in Appendix B.

There has also been recent works on theoretically understanding different aspects of adversarial robustness. Tsipras et al. (2018) give constructions under which every classifier with small $0/1$ loss has a large adversarial $0/1$ loss thereby pointing to a tension between the two criteria. This tradeoff has been explored in subsequent work (Zhang et al., 2019; Carmon et al., 2019). Bubeck et al. (2018b), Bubeck et al. (2018a) and Awasthi et al. (2019) quantify computational bottlenecks in learning classifiers with small adversarial loss. The recent work of Bartlett et al. (2021) shows that for randomly innitialized neural networks, low perturbation magnitude adversarial examples exist, with high probability, nearby every data point. There has also been a line of work analyzing the sample complexity of optimizing adversarial surrogate losses using notions of VC-dimension and Rademacher complexity appropriately extended to the adversarial case (Yin et al., 2019; Khim and Loh, 2018; Awasthi et al., 2020; Montasser et al., 2019; Cullina et al., 2018). Another recent line of concerns constructing computationally efficient adversarially robust classifiers for linear classifiers (Diakonikolas et al., 2020) and exploring the connections between adversarial learning and agnostic PAC learning (Montasser et al., 2020). Finally, an alternative adversarial setting has been theoretically studied in (Feige et al., 2015, 2018; Attias et al., 2018), where the adversary has at his disposal a finite set of perturbations for each input.

# B  Further comments on (Bao et al., 2020)

Our study is somewhat inspired by and benefits from the prior work of Bao et al. (2020). However, there are some issues worth pointing out.

Bao et al. (2020) analyzed $\mathcal{H}$-*calibration* for adversarially robust classification in the special case where $\mathcal{H}$ is the family of linear models. In particular, the authors studied the $\gamma$-margin loss defined by $\phi_\gamma(f, \mathbf{x}, y) = \mathbb{1}_{yf(\mathbf{x}) \leq \gamma}$, which only coincides with the *adversarial* $0/1$ *loss* in the case of linear hypotheses. The authors showed that, when $\mathcal{H}$ is linear, convex margin-based losses are not $\mathcal{H}$-calibrated and proposed a class of $\mathcal{H}$-calibrated surrogates modulo subtle definition differences.

However, several clarifications are needed. First, the definition of calibration adopted by the authors does not coincide with the standard definition (Steinwart, 2007) in the case of the linear models they study, although it does match that definition in the case of the family of all measurable functions (Steinwart, 2007, Section 3.2): the minimal inner risk in the definition should be defined for a fixed $\mathbf{x}$ and the infimum should be over $f$, instead of an infimum over both $f$ and $\mathbf{x}$. Second, and this is crucial, $\mathcal{H}$-*calibration*, in general, does not imply $\mathcal{H}$-*consistency*, unless a property such as *minimizability* holds (Steinwart, 2007, Theorem 2.8). *Minimizability* holds for standard binary classification and the family of all measurable functions (Steinwart, 2007, Theorem 3.2). However, it does not hold, in general, for adversarially robust classification and a specific hypothesis set $\mathcal{H}$. As a result, the claim made by the authors that the calibrated surrogates they propose are $\mathcal{H}$-*consistent* is proved to be incorrect as a by-product of our results, which further suggests that the adversarial setting is more complex and requires a more delicate analysis. Third, the authors analyzed $\mathcal{H}$-*calibration* with respect to the $\gamma$-margin loss $\phi_\gamma: \mathbf{x} \mapsto \mathbb{1}_{yf(\mathbf{x}) \leq \gamma}$ in the case where $\mathcal{H} \supset [-1, 1]$ is the general family of functions. However, as already mentioned, $\phi_\gamma$ coincides with the *adversarial* $0/1$ *loss* $\ell_\gamma$, introduced in (2), only in the special case where $\mathcal{H}$ is the family of linear models and adversarial perturbations measured in $\ell_2$ norm are considered (Bao et al., 2020, Proposition 1). Fourth, for the negative results, the authors presented a calibration analysis of convex margin-based losses, which is natural for standard $0/1$ loss, but the current practice in designing algorithms for the *adversarial* $0/1$ *loss* typically consists of using convex *supremum-based* surrogates (Madry et al., 2017; Wong et al., 2020; Shafahi et al., 2019). Finally, the experiments in (Bao et al., 2020) are problematic. Equation (12) in Appendix D.1., which they used to compute the Bayes risks, is wrong since $\mathcal{R}_\ell(f^*) = \mathcal{R}^*_{\ell, \mathcal{H}_{\mathrm{lin}}}$ cannot imply $\mathcal{C}_\ell(f^*, \mathbf{x}, \eta) = \mathcal{C}^*_{\ell, \mathcal{H}_{\mathrm{lin}}}(\mathbf{x}, \eta)$ in general. Let us point out that some of the issues just brought up have been discussed in a corrigendum following comments and questions we addressed to the authors (Bao et al., 2021), but some others do not seem to have been fully addressed there.

In contrast with that prior work, instead of studying the $\gamma$-margin loss and the specific family of linear models though, we directly study the adversarial $0/1$ loss and general hypothesis sets. In particular, our calibration results fix the results presented in (Bao et al., 2020) for linear hypothesis sets by using the correct definition, and significantly generalize them to the nonlinear hypothesis sets. For any non-decreasing and continuous $g$-based hypothesis set, including the ReLU-based hypothesis set, we also study the type of margin-based surrogates losses whose *generic conditional risk* has quasi-concave property and further establish several useful properties of such losses building on the work of Bao et al. (2020). Moreover, our results imply that the convex *supremum-based* surrogates commonly used in practice for optimizing the adversarial loss are not $\mathcal{H}$-*consistent* and that minimizing such losses may not lead to a more favorable adversarial loss. Instead, we suggest alternative surrogate losses that we prove are $\mathcal{H}$-*consistent* for any symmetric hypothesis set including the multi-layer neural networks, the supremum-based $\rho$-*margin loss*, which can be useful to the design of effective algorithms. To show the claim in (Bao et al., 2020) that the $\mathcal{H}$-*calibrated* surrogates they propose are $\mathcal{H}$-*consistent* is inaccurate, we carefully design a distribution on the unit disk, where any continuous surrogate can be led astray to a classifier that is far from the optimal classifier of the *adversarial* $0/1$ *loss*. This counterexample in fact rules out the $\mathcal{H}$-*consistency* of a larger class of surrogates, unless assumptions on the data distribution are imposed. In contrast, we give natural $\mathcal{H}$-*consistency* guarantees taking inspiration from the work of Long and Servedio (2013) and Zhang and Agarwal (2020). With the correct approximation of Bayes risks, our experiments further empirically demonstrate that indeed the $\mathcal{H}$-*calibrated* losses proposed in (Bao et al., 2020) are not $\mathcal{H}$-*consistent* and justify our proposed conditions for $\mathcal{H}$-*consistency*.

# C Details of Experiments

As shown by Bao et al. (2020), the adversarial 0/1 loss $\ell_\gamma = \mathbb{1}_{yf(\mathbf{x})\le\gamma}$ when $f \in \mathcal{H}_{\text{lin}}$. In this experiment, we approximate $\mathcal{R}^*_{\ell_\gamma,\mathcal{H}_{\text{lin}}}$ over a grid. For surrogate losses, we approximate $f^* = \operatorname{argmin}_{f\in\mathcal{H}_{\text{lin}}} \mathcal{R}_{\phi_{\text{sur}}}(f)$ over the same grid. The experiments were run on a standard laptop with a 2.4 GHz Quad-Core Intel Core i5 Processor.

## C.1 Definition of Surrogates

- Shifted Hinge loss: $\phi_{\text{hinge}}(t) = \max\{0, 1 - t + 0.2\}$;
- Shifted Ramp loss: $\phi_{\text{ramp}}(t) = \min\{1, \max\{0, \frac{1-t+0.92}{2}\}\}$;
- Shifted Sigmoid loss: $\phi_{\text{sig}}(t) = \frac{1}{1+e^{t-0.2}}$;
- Shifted Logistic loss: $\phi_{\text{log}}(t) = \log_2(1 + e^{-t+0.2})$;
- One convex loss: $\phi_1(t) = \max\{0, \frac{\gamma}{2} - t\}$; and
- $\rho$-margin loss: $\phi_2(t) = \min\{1, \max\{0, 1 - \frac{t}{\hat\gamma}\}\}$ for $\hat\gamma > \gamma$.

## C.2 Theoretical Analysis of Surrogates

$\phi_{\text{hinge}}$, $\phi_{\text{log}}$, and $\phi_1$ are convex surrogates and thus are not $\mathcal{H}_{\text{lin}}$-calibrated with respect to $\ell_\gamma$ by Corollary 8. However, $\phi_{\text{ramp}}$, $\phi_{\text{sig}}$ and $\phi_2$ are $\mathcal{H}_{\text{lin}}$-calibrated with respect to $\ell_\gamma$ since they verify the conditions in Theorem 12.

Note that $\mathbb{E}_{(X,Y)}[\phi_2(Y\mathbf{w}\cdot X)] = 0$ if and only if $w = (1,0)^\top$. Therefore, $\phi_2$ is $\mathcal{H}_{\text{lin}}$-consistent for the distribution **Segments**. However, for $w = (1,0)^\top$ or $w = (\cos(\theta), \sin(\theta))^\top$ where $\theta = \frac{\pi}{6}$, we have $\mathbb{E}_{(X,Y)}[\phi_1(Y\mathbf{w}\cdot X)] = 0$. Note when $\mathbf{w} = (\cos(\theta), \sin(\theta))^\top$ where $\theta = \frac{\pi}{6}$, we have $\mathbb{E}_{(X,Y)}[\ell_\gamma(Y\mathbf{w}\cdot X)] \ne 0$. Therefore, $\phi_1$ is not $\mathcal{H}_{\text{lin}}$-consistent for the distribution **Segments**.

# D Future work

While our calibration and consistency results are very general and apply to several widely used hypothesis sets, other hypothesis sets might require a further study. Nevertheless, we believe that our proof techniques should provide a sufficient tool for the analysis of such other cases.

# E    Deferred Proofs

For convenience, let $\Delta\mathcal{C}_{\ell,\mathcal{H}}(f,\mathbf{x},\eta) := \mathcal{C}_\ell(f,\mathbf{x},\eta) - \mathcal{C}^*_{\ell,\mathcal{H}}(\mathbf{x},\eta)$, $\underline{M}(f,\mathbf{x},\gamma) := \inf_{\mathbf{x}':\|\mathbf{x}-\mathbf{x}'\|\leq\gamma} f(\mathbf{x}')$ and $\overline{M}(f,\mathbf{x},\gamma) := -\inf_{\mathbf{x}':\|\mathbf{x}-\mathbf{x}'\|\leq\gamma} -f(\mathbf{x}') = \sup_{\mathbf{x}':\|\mathbf{x}-\mathbf{x}'\|\leq\gamma} f(\mathbf{x}')$.

## E.1    Proof of Theorem 6

**Theorem 6.** *Let $\mathcal{H}$ be a symmetric hypothesis set. If $\mathcal{H}$ is not regular for adversarial calibration, then any surrogate loss $\ell$ is $\mathcal{H}$-calibrated with respect to $\ell_\gamma$.*

*Proof.* Since $\mathcal{H}$ is symmetric, for any $\mathbf{x} \in \mathcal{X}$, $f \in \mathcal{H}$, $\inf_{\|\mathbf{x}'-\mathbf{x}\|\leq\gamma} f(\mathbf{x}') \leq 0 \leq \sup_{\|\mathbf{x}'-\mathbf{x}\|\leq\gamma} f(\mathbf{x}')$. Thus by the definition of inner risk (4) and adversarial 0-1 loss $\ell_\gamma$ (9), for any $\mathbf{x} \in \mathcal{X}$, $f \in \mathcal{H}$,

$$\mathcal{C}_{\ell_\gamma,\mathcal{H}}(f,\mathbf{x},\eta) = \eta\mathbb{1}_{\inf_{\mathbf{x}':\|\mathbf{x}-\mathbf{x}'\|\leq\gamma} f(\mathbf{x}')\leq 0} + (1-\eta)\mathbb{1}_{\sup_{\mathbf{x}':\|\mathbf{x}-\mathbf{x}'\|\leq\gamma} f(\mathbf{x}')\geq 0} = 1 = \mathcal{C}^*_{\ell_\gamma,\mathcal{H}}(\mathbf{x},\eta),$$

which implies any surrogate loss $\ell$ is $\mathcal{H}$-calibrated with respect to $\ell_\gamma$ by (5).    $\square$

## E.2    Proof of Theorem 7, Theorem 9 and Theorem 10

We first characterize the calibration function $\delta_{\max}(\epsilon,\mathbf{x},\eta)$ of losses $(\ell,\ell_\gamma)$ at $\eta = \frac{1}{2}$, $\epsilon = \frac{1}{2}$ and distinguishing $\mathbf{x}_0 \in \mathcal{X}$ given a hypothesis set $\mathcal{H}$ which is regular for adversarial calibration.

**Lemma 25.** *Let $\mathcal{H}$ be a hypothesis set that is regular for adversarial calibration. For distinguishing $\mathbf{x}_0 \in \mathcal{X}$, the calibration function $\delta_{\max}(\epsilon,\mathbf{x},\eta)$ of losses $(\ell,\ell_\gamma)$ satisfies*

$$\delta_{\max}\left(\frac{1}{2},\mathbf{x}_0,\frac{1}{2}\right) = \inf_{f\in\mathcal{H}:\ \underline{M}(f,\mathbf{x}_0,\gamma)\leq 0\leq\overline{M}(f,\mathbf{x}_0,\gamma)} \Delta\mathcal{C}_{\ell,\mathcal{H}}(f,\mathbf{x}_0,\frac{1}{2}).$$

*Proof.* By the definition of inner risk (4) and adversarial 0-1 loss $\ell_\gamma$ (9), the inner $\ell_\gamma$-risk is

$$\mathcal{C}_{\ell_\gamma}(f,\mathbf{x},\eta) = \eta\mathbb{1}_{\{\underline{M}(f,\mathbf{x},\gamma)\leq 0\}} + (1-\eta)\mathbb{1}_{\{\overline{M}(f,\mathbf{x},\gamma)\geq 0\}}$$

$$= \begin{cases} 1 & \text{if } \underline{M}(f,\mathbf{x},\gamma) \leq 0 \leq \overline{M}(f,\mathbf{x},\gamma), \\ \eta & \text{if } \overline{M}(f,\mathbf{x},\gamma) < 0, \\ 1-\eta & \text{if } \underline{M}(f,\mathbf{x},\gamma) > 0. \end{cases}$$

For distinguishing $\mathbf{x}_0$ and $\eta \in [0,1]$, $\{f \in \mathcal{H} : \overline{M}(f,\mathbf{x}_0,\gamma)\} < 0$ and $\{f \in \mathcal{H} : \underline{M}(f,\mathbf{x}_0,\gamma) > 0\}$ are not empty sets. Thus

$$\mathcal{C}^*_{\ell_\gamma,\mathcal{H}}(\mathbf{x}_0,\eta) = \inf_{f\in\mathcal{H}} \mathcal{C}_{\ell_\gamma}(f,\mathbf{x}_0,\eta) = \min\{\eta,1-\eta\}.$$

Note for $f \in \{f \in \mathcal{H} : \underline{M}(f,\mathbf{x}_0,\gamma) \leq 0 \leq \overline{M}(f,\mathbf{x}_0,\gamma)\}$, $\Delta\mathcal{C}_{\ell_\gamma,\mathcal{H}}(f,\mathbf{x}_0,\eta) = \max\{\eta,1-\eta\}$; for $f \in \{f \in \mathcal{H} : \overline{M}(f,\mathbf{x}_0,\gamma)\} < 0$, $\Delta\mathcal{C}_{\ell_\gamma,\mathcal{H}}(f,\mathbf{x}_0,\eta) = \eta - \min\{\eta,1-\eta\} = \max\{0,2\eta-1\} = |2\eta-1|\mathbb{1}_{(2\eta-1)(\underline{M}(f,\mathbf{x}_0,\gamma))\leq 0}$ since $\underline{M}(f,\mathbf{x}_0,\gamma) \leq \overline{M}(f,\mathbf{x}_0,\gamma) < 0$; for $f \in \{f \in \mathcal{H} : \underline{M}(f,\mathbf{x}_0,\gamma) > 0\}$, $\Delta\mathcal{C}_{\ell_\gamma,\mathcal{H}}(f,\mathbf{x}_0,\eta) = (1-\eta) - \min\{\eta,1-\eta\} = \max\{0,1-2\eta\} = |2\eta-1|\mathbb{1}_{(2\eta-1)(\underline{M}(f,\mathbf{x}_0,\gamma))\leq 0}$. Therefore,

$$\Delta\mathcal{C}_{\ell_\gamma,\mathcal{H}}(f,\mathbf{x}_0,\eta) = \begin{cases} \max\{\eta,1-\eta\} & \text{if } \underline{M}(f,\mathbf{x}_0,\gamma) \leq 0 \leq \overline{M}(f,\mathbf{x}_0,\gamma), \\ |2\eta-1|\mathbb{1}_{(2\eta-1)(\underline{M}(f,\mathbf{x}_0,\gamma))\leq 0} & \text{if } \underline{M}(f,\mathbf{x}_0,\gamma) > 0 \text{ or } \overline{M}(f,\mathbf{x}_0,\gamma) < 0. \end{cases}$$

By (6), for a fixed $\eta \in [0,1]$ and $\mathbf{x} \in \mathcal{X}$, the calibration function of losses $(\ell,\ell_\gamma)$ is

$$\delta_{\max}(\epsilon,\mathbf{x},\eta) = \inf_{f\in\mathcal{H}} \left\{ \Delta\mathcal{C}_{\ell,\mathcal{H}}(f,\mathbf{x},\eta) \mid \Delta\mathcal{C}_{\ell_\gamma,\mathcal{H}}(f,\mathbf{x},\eta) \geq \epsilon \right\}.$$

Observe that for all $\eta \in [0,1]$,

$$\max\{\eta,1-\eta\} = \frac{1}{2}\left[(1-\eta)+\eta+|(1-\eta)-\eta|\right] = \frac{1}{2}\left[1+|2\eta-1|\right] \geq |2\eta-1|. \tag{11}$$

For distinguishing $\mathbf{x}_0$, $\eta = \frac{1}{2}$ and $\epsilon = \frac{1}{2}$, $\Delta\mathcal{C}_{\ell_\gamma,\mathcal{H}}(f,\mathbf{x}_0,\frac{1}{2}) \geq \frac{1}{2}$ if and only if $\underline{M}(f,\mathbf{x}_0,\gamma) \leq 0 \leq \overline{M}(f,\mathbf{x}_0,\gamma)$ since $|2\eta - 1| < \epsilon \leq \max\{\eta, 1-\eta\}$. Therefore,

$$\delta_{\max}\left(\frac{1}{2},\mathbf{x}_0,\frac{1}{2}\right) = \inf_{f\in\mathcal{H}:\ \underline{M}(f,\mathbf{x}_0,\gamma)\leq 0\leq\overline{M}(f,\mathbf{x}_0,\gamma)} \Delta\mathcal{C}_{\ell,\mathcal{H}}(f,\mathbf{x}_0,\frac{1}{2}).$$

$\square$

**Theorem 7.** *Assume $\mathcal{H}$ is such that there exists a distinguishing $\mathbf{x}_0 \in \mathcal{X}$ and $f_0 \in \mathcal{H}$ such that $f_0(\mathbf{x}_0) = 0$. If a margin-based loss $\phi\colon\mathbb{R} \to \mathbb{R}_+$ is convex, then it is not $\mathcal{H}$-calibrated with respect to $\ell_\gamma$.*

*Proof.* By Lemma 25, for distinguishing $\mathbf{x}_0 \in \mathcal{X}$, the calibration function $\delta_{\max}(\epsilon,\mathbf{x},\eta)$ of losses $(\phi,\ell_\gamma)$ satisfies

$$\delta_{\max}\left(\frac{1}{2},\mathbf{x}_0,\frac{1}{2}\right) = \inf_{f\in\mathcal{H}:\ \underline{M}(f,\mathbf{x}_0,\gamma)\leq 0\leq\overline{M}(f,\mathbf{x}_0,\gamma)} \Delta\mathcal{C}_{\phi,\mathcal{H}}(f,\mathbf{x}_0,\frac{1}{2}).$$

Suppose that $\phi$ is $\mathcal{H}$-calibrated with respect to $\ell_\gamma$. By Proposition 4, $\phi$ is $\mathcal{H}$-calibrated with respect to $\ell_\gamma$ if and only if its calibration function $\delta_{\max}$ satisfies $\delta_{\max}(\epsilon,\mathbf{x},\eta) > 0$ for all $\mathbf{x} \in \mathcal{X}$, $\eta \in [0,1]$ and $\epsilon > 0$. In particular, the condition requires $\delta_{\max}\left(\frac{1}{2},\mathbf{x}_0,\frac{1}{2}\right) > 0$, that is,

$$\inf_{f\in\mathcal{H}:\ \underline{M}(f,\mathbf{x}_0,\gamma)\leq 0\leq\overline{M}(f,\mathbf{x}_0,\gamma)} \Delta\mathcal{C}_{\phi,\mathcal{H}}(f,\mathbf{x}_0,\frac{1}{2}) > 0,$$

which is equivalent to

$$\inf_{f\in\mathcal{H}:\ \underline{M}(f,\mathbf{x}_0,\gamma)\leq 0\leq\overline{M}(f,\mathbf{x}_0,\gamma)} \mathcal{C}_\phi(f,\mathbf{x}_0,\frac{1}{2}) > \inf_{f\in\mathcal{H}} \mathcal{C}_\phi(f,\mathbf{x}_0,\frac{1}{2}), \tag{12}$$

By the definition of inner risk (4),

$$\mathcal{C}_\phi(f,\mathbf{x}_0,\frac{1}{2}) = \frac{1}{2}\left(\phi(f(\mathbf{x}_0)) + \phi(-f(\mathbf{x}_0))\right). \tag{13}$$

Since $\phi$ is convex, by Jensen's inequality, for any $f \in \mathcal{H}$, the following holds:

$$\mathcal{C}_\phi(f,\mathbf{x}_0,\frac{1}{2}) \geq \phi\left(\frac{1}{2}f(\mathbf{x}_0) - \frac{1}{2}f(\mathbf{x}_0)\right) = \phi(0).$$

For $f = f_0$, we have $f_0(\mathbf{x}_0) = 0$ and by (13),

$$\mathcal{C}_\phi(f_0,\mathbf{x}_0,\frac{1}{2}) = \frac{1}{2}(\phi(0) + \phi(0)) = \phi(0).$$

Moreover, when $f = f_0$, $\underline{M}(f_0,\mathbf{x}_0,\gamma) \leq f_0(\mathbf{x}_0) = 0 \leq \overline{M}(f_0,\mathbf{x}_0,\gamma)$. Thus

$$\inf_{f\in\mathcal{H}:\ \underline{M}(f,\mathbf{x}_0,\gamma)\leq 0\leq\overline{M}(f,\mathbf{x}_0,\gamma)} \mathcal{C}_\phi(f,\mathbf{x}_0,\frac{1}{2}) = \inf_{f\in\mathcal{H}} \mathcal{C}_\phi(f,\mathbf{x}_0,\frac{1}{2}) = \phi(0),$$

where the minimum can be achieved by $f = f_0$, contradicting (12). Therefore, $\phi$ is not $\mathcal{H}$-calibrated with respect to $\ell_\gamma$. $\square$

**Theorem 9.** *Let $\phi$ be a convex and non-increasing margin-based loss. Consider the surrogate loss defined by $\tilde{\phi}(f,\mathbf{x},y) = \sup_{\mathbf{x}':\|\mathbf{x}-\mathbf{x}'\|\leq\gamma} \phi(yf(\mathbf{x}'))$. Then $\tilde{\phi}$ is not $\mathcal{H}$-calibrated with respect to $\ell_\gamma$, for $\mathcal{H} = \mathcal{H}_{\mathrm{lin}}$, $\mathcal{H}_g$ with a non-decreasing and continuous function $g$ such that $g(-\gamma) + G > 0$ and $g(\gamma) - G < 0$, and $\mathcal{H}_{\mathrm{relu}}$ with $G > \gamma$.*

*Proof.* By Lemma 25, for distinguishing $\mathbf{x}_0 \in \mathcal{X}$, the calibration function $\delta_{\max}(\epsilon,\mathbf{x},\eta)$ of losses $(\tilde{\phi},\ell_\gamma)$ satisfies

$$\delta_{\max}\left(\frac{1}{2},\mathbf{x}_0,\frac{1}{2}\right) = \inf_{f\in\mathcal{H}:\ \underline{M}(f,\mathbf{x}_0,\gamma)\leq 0\leq\overline{M}(f,\mathbf{x}_0,\gamma)} \Delta\mathcal{C}_{\tilde{\phi},\mathcal{H}}(f,\mathbf{x}_0,\frac{1}{2}).$$

Next we first consider the case where $\mathcal{H} = \mathcal{H}_{\text{lin}}$. Take distinguishing $\mathbf{x}_0 \in \mathcal{X}$ and $f_0 \in \mathcal{H}_{\text{lin}}$ such that $f_0(\mathbf{x}_0) = 0$. As shown by Awasthi et al. (2020), for $f \in \mathcal{H}_{\text{lin}} = \{\mathbf{x} \to \mathbf{w} \cdot \mathbf{x} \mid \|\mathbf{w}\| = 1\}$,

$$\underline{M}(f, \mathbf{x}, \gamma) = \inf_{\mathbf{x}': \|\mathbf{x} - \mathbf{x}'\| \le \gamma} f(\mathbf{x}') = \inf_{\mathbf{x}': \|\mathbf{x} - \mathbf{x}'\| \le \gamma} (\mathbf{w} \cdot \mathbf{x}') = \mathbf{w} \cdot \mathbf{x} - \gamma \|\mathbf{w}\| = f(\mathbf{x}) - \gamma,$$

$$\overline{M}(f, \mathbf{x}, \gamma) = -\inf_{\mathbf{x}': \|\mathbf{x} - \mathbf{x}'\| \le \gamma} -f(\mathbf{x}') = -\inf_{\mathbf{x}': \|\mathbf{x} - \mathbf{x}'\| \le \gamma} (-\mathbf{w} \cdot \mathbf{x}') = \mathbf{w} \cdot \mathbf{x} + \gamma \|\mathbf{w}\| = f(\mathbf{x}) + \gamma.$$

Suppose that $\tilde{\phi}$ is $\mathcal{H}_{\text{lin}}$-calibrated with respect to $\ell_\gamma$. By Proposition 4, $\tilde{\phi}$ is $\mathcal{H}_{\text{lin}}$-calibrated with respect to $\ell_\gamma$ if and only if its calibration function $\delta_{\max}$ satisfies $\delta_{\max}(\epsilon, \mathbf{x}, \eta) > 0$ for all $\mathbf{x} \in \mathcal{X}$, $\eta \in [0, 1]$ and $\epsilon > 0$. In particular, the condition requires $\delta_{\max}\left(\frac{1}{2}, \mathbf{x}_0, \frac{1}{2}\right) > 0$, that is,

$$\inf_{f \in \mathcal{H}_{\text{lin}}: -\gamma \le f(\mathbf{x}_0) \le \gamma} \Delta \mathcal{C}_{\tilde{\phi}, \mathcal{H}_{\text{lin}}}(f, \mathbf{x}_0, \frac{1}{2}) > 0,$$

which is equivalent to

$$\inf_{f \in \mathcal{H}_{\text{lin}}: -\gamma \le f(\mathbf{x}_0) \le \gamma} \mathcal{C}_{\tilde{\phi}}(f, \mathbf{x}_0, \frac{1}{2}) > \inf_{f \in \mathcal{H}_{\text{lin}}} \mathcal{C}_{\tilde{\phi}}(f, \mathbf{x}_0, \frac{1}{2}), \tag{14}$$

By (19), for $f \in \mathcal{H}_{\text{lin}}$,

$$\mathcal{C}_{\tilde{\phi}}(f, \mathbf{x}_0, \frac{1}{2}) = \frac{1}{2} \phi(f(\mathbf{x}_0) - \gamma) + \frac{1}{2} \phi(-f(\mathbf{x}_0) - \gamma). \tag{15}$$

Since $\phi$ is convex, by Jensen's inequality, for any $f \in \mathcal{H}_{\text{lin}}$, the following holds:

$$\mathcal{C}_{\tilde{\phi}}(f, \mathbf{x}_0, \frac{1}{2}) \ge \phi\left(\frac{1}{2}(f(\mathbf{x}_0) - \gamma) - \frac{1}{2}(f(\mathbf{x}_0) + \gamma)\right) = \phi(-\gamma).$$

For $f = f_0$, we have $f_0(\mathbf{x}_0) = 0$ and by (15),

$$\mathcal{C}_{\tilde{\phi}}(f_0, \mathbf{x}_0, \frac{1}{2}) = \frac{1}{2}(\phi(-\gamma) + \phi(-\gamma)) = \phi(-\gamma).$$

Moreover, when $f = f_0$, $-\gamma \le f_0(\mathbf{x}_0) = 0 \le \gamma$. Thus

$$\inf_{f \in \mathcal{H}: -\gamma \le f(\mathbf{x}_0) \le \gamma} \mathcal{C}_{\tilde{\phi}}(f, \mathbf{x}_0, \frac{1}{2}) = \inf_{f \in \mathcal{H}} \mathcal{C}_{\tilde{\phi}}(f, \mathbf{x}_0, \frac{1}{2}) = \phi(-\gamma),$$

where the minimum can be achieved by $f = f_0$, contradicting (14). Therefore, $\tilde{\phi}$ is not $\mathcal{H}_{\text{lin}}$-calibrated with respect to $\ell_\gamma$.

Then we consider the case where $\mathcal{H} = \mathcal{H}_g$. By the assumption on $g$, $0 \in \mathcal{X}$ is distinguishing. As shown by Awasthi et al. (2020), for $f \in \mathcal{H}_g$,

$$\underline{M}(f, \mathbf{x}, \gamma) = g(\mathbf{w} \cdot \mathbf{x} - \gamma) + b, \quad \overline{M}(f, \mathbf{x}, \gamma) = g(\mathbf{w} \cdot \mathbf{x} + \gamma) + b.$$

Suppose that $\tilde{\phi}$ is $\mathcal{H}_g$-calibrated with respect to $\ell_\gamma$. By Proposition 4, $\tilde{\phi}$ is $\mathcal{H}_g$-calibrated with respect to $\ell_\gamma$ if and only if its calibration function $\delta_{\max}$ satisfies $\delta_{\max}(\epsilon, \mathbf{x}, \eta) > 0$ for all $\mathbf{x} \in \mathcal{X}$, $\eta \in [0, 1]$ and $\epsilon > 0$. In particular, the condition requires $\delta_{\max}\left(\frac{1}{2}, 0, \frac{1}{2}\right) > 0$, that is,

$$\inf_{f \in \mathcal{H}_g: g(-\gamma) + b \le 0 \le g(\gamma) + b} \Delta \mathcal{C}_{\tilde{\phi}, \mathcal{H}_g}(f, 0, \frac{1}{2}) > 0,$$

which is equivalent to

$$\inf_{f \in \mathcal{H}_g: g(-\gamma) + b \le 0 \le g(\gamma) + b} \mathcal{C}_{\tilde{\phi}}(f, 0, \frac{1}{2}) > \inf_{f \in \mathcal{H}_g} \mathcal{C}_{\tilde{\phi}}(f, 0, \frac{1}{2}), \tag{16}$$

By (19), for $f \in \mathcal{H}_g$,

$$\mathcal{C}_{\tilde{\phi}}(f, 0, \frac{1}{2}) = \frac{1}{2} \phi(g(-\gamma) + b) + \frac{1}{2} \phi(-g(\gamma) - b). \tag{17}$$

Since $\phi$ is convex, by Jensen's inequality, for any $f \in \mathcal{H}_g$, the following holds:

$$\mathcal{C}_{\tilde{\phi}}(f, 0, \frac{1}{2}) \ge \phi\left(\frac{1}{2}(g(-\gamma) + b) + \frac{1}{2}(-g(\gamma) - b)\right) = \phi\left(\frac{g(-\gamma) - g(\gamma)}{2}\right).$$

Take $f_0 \in \mathcal{H}_g$ with $b_0 = \frac{-g(\gamma)-g(-\gamma)}{2}$, we have $g(-\gamma) + b_0 = -g(\gamma) - b_0 = \frac{g(-\gamma)-g(\gamma)}{2}$ and by (17),

$$\mathcal{C}_{\tilde{\phi}}(f_0, 0, \tfrac{1}{2}) = \frac{1}{2}\phi(g(-\gamma) + b_0) + \frac{1}{2}\phi(-g(\gamma) - b_0) = \phi\left(\frac{g(-\gamma) - g(\gamma)}{2}\right).$$

Moreover, when $f = f_0$, $g(-\gamma) + b_0 \le 0 \le g(\gamma) + b_0$. Thus

$$\inf_{f \in \mathcal{H}_g:\, g(-\gamma)+b \le 0 \le g(\gamma)+b} \mathcal{C}_{\tilde{\phi}}(f, 0, \tfrac{1}{2}) = \inf_{f \in \mathcal{H}_g} \mathcal{C}_{\tilde{\phi}}(f, 0, \tfrac{1}{2}) = \phi\left(\frac{g(-\gamma) - g(\gamma)}{2}\right),$$

where the minimum can be achieved by $f = f_0$, contradicting (16). Therefore, $\tilde{\phi}$ is not $\mathcal{H}_g$-calibrated with respect to $\ell_\gamma$. $\qquad\square$

**Theorem 10.** *Let $\mathcal{H}$ be a hypothesis set containing $0$ that is regular for adversarial calibration. If a margin-based loss $\phi$ is convex and non-increasing, then the surrogate loss defined by $\tilde{\phi}(f, \mathbf{x}, y) = \sup_{\mathbf{x}':\|\mathbf{x}-\mathbf{x}'\| \le \gamma} \phi(y f(\mathbf{x}'))$ is not $\mathcal{H}$-calibrated with respect to $\ell_\gamma$.*

*Proof.* By Lemma 25, for distinguishing $\mathbf{x}_0 \in \mathcal{X}$, the calibration function $\delta_{\max}(\epsilon, \mathbf{x}, \eta)$ of losses $(\tilde{\phi}, \ell_\gamma)$ satisfies

$$\delta_{\max}\left(\frac{1}{2}, \mathbf{x}_0, \frac{1}{2}\right) = \inf_{f \in \mathcal{H}:\, \underline{M}(f,\mathbf{x}_0,\gamma) \le 0 \le \overline{M}(f,\mathbf{x}_0,\gamma)} \Delta\mathcal{C}_{\tilde{\phi},\mathcal{H}}(f, \mathbf{x}_0, \tfrac{1}{2}).$$

Suppose that $\tilde{\phi}$ is $\mathcal{H}$-calibrated with respect to $\ell_\gamma$. By Proposition 4, $\tilde{\phi}$ is $\mathcal{H}$-calibrated with respect to $\ell_\gamma$ if and only if its calibration function $\delta_{\max}$ satisfies $\delta_{\max}(\epsilon, \mathbf{x}, \eta) > 0$ for all $\mathbf{x} \in \mathcal{X}$, $\eta \in [0, 1]$ and $\epsilon > 0$. In particular, the condition requires $\delta_{\max}\left(\frac{1}{2}, \mathbf{x}_0, \frac{1}{2}\right) > 0$, that is,

$$\inf_{f \in \mathcal{H}:\, \underline{M}(f,\mathbf{x}_0,\gamma) \le 0 \le \overline{M}(f,\mathbf{x}_0,\gamma)} \Delta\mathcal{C}_{\tilde{\phi},\mathcal{H}}(f, \mathbf{x}_0, \tfrac{1}{2}) > 0,$$

which is equivalent to

$$\inf_{f \in \mathcal{H}:\, \underline{M}(f,\mathbf{x}_0,\gamma) \le 0 \le \overline{M}(f,\mathbf{x}_0,\gamma)} \mathcal{C}_{\tilde{\phi}}(f, \mathbf{x}_0, \tfrac{1}{2}) > \inf_{f \in \mathcal{H}} \mathcal{C}_{\tilde{\phi}}(f, \mathbf{x}_0, \tfrac{1}{2}), \tag{18}$$

As shown by Awasthi et al. (2020), $\tilde{\phi}$ has the equivalent form

$$\tilde{\phi}(f, \mathbf{x}, y) = \phi\left(\inf_{\|\mathbf{x}'-\mathbf{x}\| \le \gamma} (y f(\mathbf{x}'))\right).$$

By the definition of inner risk (4),

$$\mathcal{C}_{\tilde{\phi}}(f, \mathbf{x}_0, \tfrac{1}{2}) = \frac{1}{2}\left(\phi(\underline{M}(f, \mathbf{x}_0, \gamma)) + \phi(-\overline{M}(f, \mathbf{x}_0, \gamma))\right). \tag{19}$$

Since $\phi$ is convex, by Jensen's inequality, for any $f \in \mathcal{H}$, the following holds:

$$\mathcal{C}_{\tilde{\phi}}(f, \mathbf{x}_0, \tfrac{1}{2}) \ge \phi\left(\frac{1}{2}\underline{M}(f, \mathbf{x}_0, \gamma) - \frac{1}{2}\overline{M}(f, \mathbf{x}_0, \gamma)\right) = \phi\left(\frac{1}{2}(\underline{M}(f, \mathbf{x}_0, \gamma) - \overline{M}(f, \mathbf{x}_0, \gamma))\right) \ge \phi(0),$$

where the last inequality used the fact that

$$\frac{1}{2}(\underline{M}(f, \mathbf{x}_0, \gamma) - \overline{M}(f, \mathbf{x}_0, \gamma)) \le 0$$

and $\phi$ is non-increasing. For $f = 0$, we have $\underline{M}(f, \mathbf{x}_0, \gamma) = \overline{M}(f, \mathbf{x}_0, \gamma) = 0$ and by (19),

$$\mathcal{C}_{\tilde{\phi}}(f, \mathbf{x}_0, \tfrac{1}{2}) = \frac{1}{2}(\phi(0) + \phi(0)) = \phi(0).$$

Moreover, when $\underline{M}(f, \mathbf{x}_0, \gamma) = \overline{M}(f, \mathbf{x}_0, \gamma) = 0$, $\underline{M}(f, \mathbf{x}_0, \gamma) \le 0 \le \overline{M}(f, \mathbf{x}_0, \gamma)$ is satisfied. Thus

$$\inf_{f \in \mathcal{H}:\, \underline{M}(f,\mathbf{x}_0,\gamma) \le 0 \le \overline{M}(f,\mathbf{x}_0,\gamma)} \mathcal{C}_{\tilde{\phi}}(f, \mathbf{x}_0, \tfrac{1}{2}) = \inf_{f \in \mathcal{H}} \mathcal{C}_{\tilde{\phi}}(f, \mathbf{x}_0, \tfrac{1}{2}) = \phi(0),$$

where the minimum can be achieved by $f = 0$, contradicting (18). Therefore, $\tilde{\phi}$ is not $\mathcal{H}$-calibrated with respect to $\ell_\gamma$. $\qquad\square$

### E.3 Properties of generic conditional $\phi$-risk

In this section, we characterize the properties of the generic conditional $\phi$-risk $\bar{\mathcal{C}}_\phi(t,\eta)$ when margin-based loss $\phi$ is bounded, continuous, non-increasing and satisfy $\bar{\mathcal{C}}_\phi(t,\eta)$ is quasi-concave in $t \in \mathbb{R}$ for all $\eta \in [0,1]$, which would be useful in the proof of Theorem 12 and Theorem 13. Without loss of generality, assume that $g$ is continuous, non-decreasing and satisfies $g(-1-\gamma) + G > 0$, $g(1+\gamma) - G < 0$.

**Lemma 26.** *Let $\phi$ be a margin-based loss. If $\phi$ is bounded, continuous, non-increasing and satisfy $\bar{\mathcal{C}}_\phi(t,\eta)$ is quasi-concave in $t \in \mathbb{R}$ for all $\eta \in [0,1]$, then*

1. *$\bar{\mathcal{C}}_\phi(t, \frac{1}{2})$ is even and non-increasing in $t$ when $t \geq 0$.*

2. *For $l, u \in \mathbb{R}(l \leq u)$, $\inf_{t \in [l,u]} \bar{\mathcal{C}}_\phi(t,\eta) = \min\{\bar{\mathcal{C}}_\phi(l,\eta), \bar{\mathcal{C}}_\phi(u,\eta)\}$ for all $\eta \in [0,1]$.*

3. *For all $\eta \in (\frac{1}{2}, 1]$, $\bar{\mathcal{C}}_\phi(t,\eta)$ is non-increasing in $t$ when $t \geq 0$.*

4. *For all $\eta \in [0, \frac{1}{2})$, $\bar{\mathcal{C}}_\phi(t,\eta)$ is non-decreasing in $t$ when $t \leq 0$.*

5. *If $\phi(-t) > \phi(t)$ for any $\gamma < t \leq 1$, then, for all $\eta \in (\frac{1}{2}, 1]$ and any $\gamma < t \leq 1$, $\bar{\mathcal{C}}_\phi(-t,\eta) > \bar{\mathcal{C}}_\phi(t,\eta)$.*

6. *If $\phi(-t) > \phi(t)$ for any $\gamma < t \leq 1$, then, for all $\eta \in [0, \frac{1}{2})$ and any $\gamma < t \leq 1$, $\bar{\mathcal{C}}_\phi(-t,\eta) < \bar{\mathcal{C}}_\phi(t,\eta)$.*

7. *If $\phi(g(-t)-G) > \phi(G-g(-t))$, $g(-t)+g(t) \geq 0$ for any $0 \leq t \leq 1$, then, for all $\eta \in (\frac{1}{2}, 1]$ and any $0 \leq t \leq 1$, $\bar{\mathcal{C}}_\phi(g(-t)-G,\eta) > \bar{\mathcal{C}}_\phi(g(t)+G,\eta)$.*

8. *If $\phi(g(-t)-G) > \phi(G-g(-t))$, $g(-t)+g(t) \geq 0$ for any $0 \leq t \leq 1$, then, for any $0 \leq t \leq 1$, $\bar{\mathcal{C}}_\phi(g(-t)-G,\eta) < \bar{\mathcal{C}}_\phi(g(t)+G,\eta)$ for all $\eta \in [0, \frac{1}{2})$ if and only if $\phi(G-g(-t))+\phi(g(-t)-G) = \phi(g(t)+G)+\phi(-g(t)-G)$.*

*Proof.* Part 1 and Part 3 of Lemma 26 are stated in (Bao et al., 2020, Lemma 13). Part 2 is implied straightforwardly by the assumption $\bar{\mathcal{C}}_\phi(t,\eta)$ is quasi-concave in $t \in \mathbb{R}$ for all $\eta \in [0,1]$ and the characterization of continuous and quasi-convex functions in (Boyd and Vandenberghe, 2014).

Consider Part 4. For $\eta \in [0, \frac{1}{2})$, and $t_1, t_2 \leq 0$. Suppose that $t_1 < t_2$, then

$$\phi(t_1) - \phi(-t_1) - \phi(t_2) + \phi(-t_2)$$
$$\geq \phi(t_2) - \phi(-t_2) - \phi(t_2) + \phi(-t_2)$$
$$= 0$$

since $\phi$ is non-increasing. By Part 1 of Lemma 26, $\phi(t) + \phi(-t)$ is non-decreasing in $t$ when $t \leq 0$. Therefore, for $\eta \in [0, \frac{1}{2})$,

$$\bar{\mathcal{C}}_\phi(t_1,\eta) - \bar{\mathcal{C}}_\phi(t_2,\eta)$$
$$= (\phi(t_1) - \phi(-t_1) - \phi(t_2) + \phi(-t_2))\eta + \phi(-t_1) - \phi(-t_2)$$
$$\leq (\phi(t_1) - \phi(-t_1) - \phi(t_2) + \phi(-t_2))\frac{1}{2} + \phi(-t_1) - \phi(-t_2)$$
$$= \frac{1}{2}(\phi(t_1) + \phi(-t_1) - \phi(t_2) - \phi(-t_2))$$
$$\leq 0.$$

Consider Part 5, For $\eta \in (\frac{1}{2}, 1]$ and any $\gamma < t \leq 1$,

$$\bar{\mathcal{C}}_\phi(-t,\eta) - \bar{\mathcal{C}}_\phi(t,\eta) = \eta\phi(-t) + (1-\eta)\phi(t) - \eta\phi(t) - (1-\eta)\phi(-t)$$
$$= (2\eta - 1)[\phi(-t) - \phi(t)] > 0$$

since $\eta > \frac{1}{2}$ and $\phi(-t) > \phi(t)$ for any $\gamma < t \leq 1$.

Consider Part 6, For $\eta \in [0, \frac{1}{2})$ and any $\gamma < t \leq 1$,

$$\bar{\mathcal{C}}_\phi(t, \eta) - \bar{\mathcal{C}}_\phi(-t, \eta) = \eta\phi(t) + (1 - \eta)\phi(-t) - \eta\phi(-t) - (1 - \eta)\phi(t)$$
$$= (1 - 2\eta)\left[\phi(-t) - \phi(t)\right] > 0$$

since $\eta < \frac{1}{2}$ and $\phi(-t) > \phi(t)$ for any $\gamma < t \leq 1$.

Consider Part 7. For $\eta \in (\frac{1}{2}, 1]$ and any $0 \leq t \leq 1$,

$$\begin{aligned}
&\bar{\mathcal{C}}_\phi(g(-t) - G, \eta) - \bar{\mathcal{C}}_\phi(g(t) + G, \eta) \\
\geq &\bar{\mathcal{C}}_\phi(g(-t) - G, \eta) - \bar{\mathcal{C}}_\phi(G - g(-t), \eta) \qquad (g(-t) + g(t) \geq 0, \text{ Part 3 of Lemma 26}) \\
= &(2\eta - 1)[\phi(g(-t) - G) - \phi(G - g(-t))] \\
> &0 \qquad\qquad\qquad\qquad\qquad\qquad\qquad\qquad\qquad (\phi(g(-t) - G) > \phi(G - g(-t)))
\end{aligned}$$

Consider Part 8. Since $\phi$ is non-increasing, for any $0 \leq t \leq 1$,

$$\begin{aligned}
&\phi(g(-t) - G) - \phi(G - g(-t)) + \phi(-g(t) - G) - \phi(g(t) + G) \\
\geq &\phi(g(-t) - G) - \phi(G - g(-t)) + \phi(g(t) + G) - \phi(g(t) + G) \qquad\qquad (g(t) + G > 0) \\
= &\phi(g(-t) - G) - \phi(G - g(-t)) \\
> &0 \qquad\qquad\qquad\qquad\qquad\qquad\qquad\qquad\qquad (\phi(g(-t) - G) > \phi(G - g(-t)))
\end{aligned}$$

$\Longleftarrow$: Suppose $\phi(G - g(-t)) + \phi(g(-t) - G) = \phi(g(t) + G) + \phi(-g(t) - G)$, then for $\eta \in [0, \frac{1}{2})$,

$$\begin{aligned}
&\bar{\mathcal{C}}_\phi(g(-t) - G, \eta) - \bar{\mathcal{C}}_\phi(g(t) + G, \eta) \\
= &(\phi(g(-t) - G) - \phi(G - g(-t)) + \phi(-g(t) - G) - \phi(g(t) + G))\eta \\
&\quad + \phi(G - g(-t)) - \phi(-g(t) - G) \\
< &(\phi(g(-t) - G) - \phi(G - g(-t)) + \phi(-g(t) - G) - \phi(g(t) + G))\frac{1}{2} \\
&\quad + \phi(G - g(-t)) - \phi(-g(t) - G) \\
= &\frac{1}{2}(\phi(G - g(-t)) + \phi(g(-t) - G) - \phi(g(t) + G) - \phi(-g(t) - G)) \\
= &0.
\end{aligned}$$

$\Longrightarrow$: Suppose $\bar{\mathcal{C}}_\phi(g(-t) - G, \eta) < \bar{\mathcal{C}}_\phi(g(t) + G, \eta)$ for $\eta \in [0, \frac{1}{2})$, then

$$\begin{aligned}
&\bar{\mathcal{C}}_\phi(g(-t) - G, \eta) - \bar{\mathcal{C}}_\phi(g(t) + G, \eta) \\
= &(\phi(g(-t) - G) - \phi(G - g(-t)) + \phi(-g(t) - G) - \phi(g(t) + G))\eta \\
&\quad + \phi(G - g(-1)) - \phi(-g(1) - G) \\
< &0
\end{aligned}$$

for $\eta \in [0, \frac{1}{2})$. By taking $\eta \to \frac{1}{2}$, we have

$$\begin{aligned}
&\frac{1}{2}(\phi(G - g(-t)) + \phi(g(-t) - G) - \phi(g(t) + G) - \phi(-g(t) - G)) \\
= &(\phi(g(-t) - G) - \phi(G - g(-t)) + \phi(-g(t) - G) - \phi(g(t) + G))\frac{1}{2} \\
&\quad + \phi(G - g(-t)) - \phi(-g(t) - G) \\
\leq &0.
\end{aligned}$$

By Part 1 of Lemma 26, we have

$$\begin{aligned}
&\phi(G - g(-t)) + \phi(g(-t) - G) - \phi(g(t) + G) - \phi(-g(t) - G) \\
\geq &\phi(g(t) + G) + \phi(-g(t) - G) - \phi(g(t) + G) - \phi(-g(t) - G) \qquad (g(-t) + g(t) \geq 0) \\
= &0.
\end{aligned}$$

Therefore, $\phi(G - g(-t)) + \phi(g(-t) - G) - \phi(g(t) + G) - \phi(-g(t) - G) = 0$, i.e., $\phi(G - g(-t)) + \phi(g(-t) - G) = \phi(g(t) + G) + \phi(-g(t) - G)$. $\qquad \square$

## E.4 Proof of Theorem 12 and Theorem 16

We will make use of general form (9) of the adversarial 0/1 loss:

$$\ell_\gamma(f, \mathbf{x}, y) = \sup_{\mathbf{x}': \|\mathbf{x}-\mathbf{x}'\| \le \gamma} \mathbb{1}_{yf(\mathbf{x}')\le 0} = \mathbb{1}_{\inf_{\mathbf{x}': \|\mathbf{x}-\mathbf{x}'\| \le \gamma} yf(\mathbf{x}')\le 0}.$$

Next, we first characterize the calibration function $\delta_{\max}(\epsilon, \mathbf{x}, \eta)$ of losses $(\ell, \ell_\gamma)$ given a symmetric hypothesis set $\mathcal{H}$.

**Lemma 27.** *Let $\mathcal{H}$ be a symmetric hypothesis set. For a surrogate loss $\ell$, the calibration function $\delta_{\max}(\epsilon, \mathbf{x}, \eta)$ of the losses $(\ell, \ell_\gamma)$ is*

$$\delta_{\max}(\epsilon, \mathbf{x}, \eta) = \begin{cases} +\infty & \text{if } \mathbf{x} \in \mathcal{X}_1 \text{ or } \mathbf{x} \in \mathcal{X}_2, \ \epsilon > \max\{\eta, 1-\eta\}, \\ \inf_{f \in \mathcal{H}: \ \underline{M}(f,\mathbf{x},\gamma) \le 0 \le \overline{M}(f,\mathbf{x},\gamma)} \Delta\mathcal{C}_{\ell,\mathcal{H}}(f, \mathbf{x}, \eta) & \text{if } \mathbf{x} \in \mathcal{X}_2, \ |2\eta - 1| < \epsilon \le \max\{\eta, 1-\eta\}, \\ \inf_{f \in \mathcal{H}: \ \underline{M}(f,\mathbf{x},\gamma) \le 0 \le \overline{M}(f,\mathbf{x},\gamma) \ or \ (2\eta-1)(\underline{M}(f,\mathbf{x},\gamma)) \le 0} \Delta\mathcal{C}_{\ell,\mathcal{H}}(f, \mathbf{x}, \eta) & \text{if } \mathbf{x} \in \mathcal{X}_2, \ \epsilon \le |2\eta - 1|, \end{cases}$$

*where $\mathcal{X}_1 = \{\mathbf{x} \in \mathcal{X} : \underline{M}(f, \mathbf{x}, \gamma) \le 0 \le \overline{M}(f, \mathbf{x}, \gamma), \ \forall f \in \mathcal{H}\}$, $\mathcal{X}_2 = \{\mathbf{x} \in \mathcal{X} : \text{there exists } f' \in \mathcal{H} \text{ such that } \underline{M}(f', \mathbf{x}, \gamma) > 0\}$ and $\mathcal{X} = \mathcal{X}_1 \cup \mathcal{X}_2, \ \mathcal{X}_1 \cap \mathcal{X}_2 = \varnothing.$*

*Proof.* By the definition of inner risk (4) and adversarial 0-1 loss $\ell_\gamma$ (9), the inner $\ell_\gamma$-risk is

$$\mathcal{C}_{\ell_\gamma}(f, \mathbf{x}, \eta) = \eta \mathbb{1}_{\{\underline{M}(f,\mathbf{x},\gamma)\le 0\}} + (1-\eta)\mathbb{1}_{\{\overline{M}(f,\mathbf{x},\gamma)\ge 0\}}$$

$$= \begin{cases} 1 & \text{if } \underline{M}(f, \mathbf{x}, \gamma) \le 0 \le \overline{M}(f, \mathbf{x}, \gamma), \\ \eta & \text{if } \overline{M}(f, \mathbf{x}, \gamma) < 0, \\ 1 - \eta & \text{if } \underline{M}(f, \mathbf{x}, \gamma) > 0. \end{cases}$$

Let $\mathcal{X}_1 = \{\mathbf{x} \in \mathcal{X} : \underline{M}(f, \mathbf{x}, \gamma) \le 0 \le \overline{M}(f, \mathbf{x}, \gamma), \ \forall f \in \mathcal{H}\}$, $\mathcal{X}_2 = \{\mathbf{x} \in \mathcal{X} : \text{there exists } f' \in \mathcal{H} \text{ such that } \underline{M}(f', \mathbf{x}, \gamma) > 0\}$. It is obvious that $\mathcal{X}_1 \cap \mathcal{X}_2 = \varnothing$. Since $\mathcal{H}$ is symmetric, for any $\mathbf{x} \in \mathcal{X}$, either there exists $f' \in \mathcal{H}$ such that $\underline{M}(f', \mathbf{x}, \gamma) > 0$ and $\overline{M}(-f', \mathbf{x}, \gamma) < 0$, or $\underline{M}(f, \mathbf{x}, \gamma) \le 0 \le \overline{M}(f, \mathbf{x}, \gamma)$ for any $f \in \mathcal{H}$. Thus $\mathcal{X} = \mathcal{X}_1 \cup \mathcal{X}_2$. Note when $\mathbf{x} \in \mathcal{X}_1$, $\{f \in \mathcal{H} : \overline{M}(f, \mathbf{x}, \gamma) < 0\}$ and $\{f \in \mathcal{H} : \underline{M}(f, \mathbf{x}, \gamma) > 0\}$ are both empty sets. Therefore, the minimal inner $\ell_\gamma$-risk is

$$\mathcal{C}^*_{\ell_\gamma, \mathcal{H}}(\mathbf{x}, \eta) = \begin{cases} 1, & \mathbf{x} \in \mathcal{X}_1, \\ \min\{\eta, 1-\eta\}, & \mathbf{x} \in \mathcal{X}_2. \end{cases}$$

Note when $\mathbf{x} \in \mathcal{X}_1$, $\mathcal{C}_{\ell_\gamma}(f, \mathbf{x}, \eta) = 1$ for any $f \in \mathcal{H}$, thus $\Delta\mathcal{C}_{\ell_\gamma, \mathcal{H}}(f, \mathbf{x}, \eta) = 0$. When $\mathbf{x} \in \mathcal{X}_2$, for $f \in \{f \in \mathcal{H} : \underline{M}(f, \mathbf{x}, \gamma) \le 0 \le \overline{M}(f, \mathbf{x}, \gamma)\}$, $\Delta\mathcal{C}_{\ell_\gamma, \mathcal{H}}(f, \mathbf{x}, \eta) = 1 - \min\{\eta, 1-\eta\} = \max\{\eta, 1-\eta\}$; for $f \in \{f \in \mathcal{H} : \overline{M}(f, \mathbf{x}, \gamma) < 0\}$, $\Delta\mathcal{C}_{\ell_\gamma, \mathcal{H}}(f, \mathbf{x}, \eta) = \eta - \min\{\eta, 1-\eta\} = \max\{0, 2\eta - 1\} = |2\eta - 1|\mathbb{1}_{(2\eta-1)(\underline{M}(f,\mathbf{x},\gamma))\le 0}$ since $\underline{M}(f, \mathbf{x}, \gamma) \le \overline{M}(f, \mathbf{x}, \gamma) < 0$; for $f \in \{f \in \mathcal{H} : \underline{M}(f, \mathbf{x}, \gamma) > 0\}$, $\Delta\mathcal{C}_{\ell_\gamma, \mathcal{H}}(f, \mathbf{x}, \eta) = 1 - \eta - \min\{\eta, 1-\eta\} = \max\{0, 1-2\eta\} = |2\eta - 1|\mathbb{1}_{(2\eta-1)(\underline{M}(f,\mathbf{x},\gamma))\le 0}$ since $\underline{M}(f, \mathbf{x}, \gamma) > 0$. Therefore,

$$\Delta\mathcal{C}_{\ell_\gamma, \mathcal{H}}(f, \mathbf{x}, \eta) = \begin{cases} \max\{\eta, 1-\eta\} & \text{if } \mathbf{x} \in \mathcal{X}_2, \ \underline{M}(f, \mathbf{x}, \gamma) \le 0 \le \overline{M}(f, \mathbf{x}, \gamma), \\ |2\eta - 1|\mathbb{1}_{(2\eta-1)(\underline{M}(f,\mathbf{x},\gamma))\le 0} & \text{if } \mathbf{x} \in \mathcal{X}_2, \ \underline{M}(f, \mathbf{x}, \gamma) > 0 \text{ or } \overline{M}(f, \mathbf{x}, \gamma) < 0, \\ 0 & \text{if } \mathbf{x} \in \mathcal{X}_1. \end{cases}$$

$$(20)$$

By (6), for a fixed $\eta \in [0, 1]$ and $\mathbf{x} \in \mathcal{X}$, the calibration function of losses $(\ell, \ell_\gamma)$ is

$$\delta_{\max}(\epsilon, \mathbf{x}, \eta) = \inf_{f \in \mathcal{H}} \left\{ \Delta\mathcal{C}_{\ell, \mathcal{H}}(f, \mathbf{x}, \eta) \mid \Delta\mathcal{C}_{\ell_\gamma, \mathcal{H}}(f, \mathbf{x}, \eta) \ge \epsilon \right\}$$

If $\mathbf{x} \in \mathcal{X}_1$, then for all $f \in \mathcal{H}$, $\Delta\mathcal{C}_{\ell_\gamma, \mathcal{H}}(f, \mathbf{x}, \eta) = 0 < \epsilon$, which implies that $\delta_{\max}(\epsilon, \mathbf{x}, \eta) = +\infty$. Next we consider case where $\mathbf{x} \in \mathcal{X}_2$. By the observation (11), if $\epsilon > \max\{\eta, 1-\eta\}$, then for all $f \in \mathcal{H}$, $\Delta\mathcal{C}_{\ell_\gamma, \mathcal{H}}(f, \mathbf{x}, \eta) < \epsilon$, which implies that $\delta_{\max}(\epsilon, \mathbf{x}, \eta) = +\infty$; if $|2\eta - 1| < \epsilon \le \max\{\eta, 1-\eta\}$, then $\Delta\mathcal{C}_{\ell_\gamma, \mathcal{H}}(f, \mathbf{x}, \eta) \ge \epsilon$ if and only if $\underline{M}(f, \mathbf{x}, \gamma) \le 0 \le \overline{M}(f, \mathbf{x}, \gamma)$, which leads to

$$\delta_{\max}(\epsilon, \mathbf{x}, \eta) = \inf_{f \in \mathcal{H}: \ \underline{M}(f,\mathbf{x},\gamma) \le 0 \le \overline{M}(f,\mathbf{x},\gamma)} \Delta\mathcal{C}_{\ell, \mathcal{H}}(f, \mathbf{x}, \eta);$$

if $\epsilon \leq |2\eta - 1|$, then $\Delta\mathcal{C}_{\ell_\gamma,\mathcal{H}}(f,\mathbf{x},\eta) \geq \epsilon$ if and only if $\underline{M}(f,\mathbf{x},\gamma) \leq 0 \leq \overline{M}(f,\mathbf{x},\gamma)$ or $(2\eta - 1)(\underline{M}(f,\mathbf{x},\gamma)) \leq 0$, which leads to

$$\delta_{\max}(\epsilon,\mathbf{x},\eta) = \inf_{f\in\mathcal{H}:\ \underline{M}(f,\mathbf{x},\gamma)\leq 0\leq\overline{M}(f,\mathbf{x},\gamma)\ \text{or}\ (2\eta-1)(\underline{M}(f,\mathbf{x},\gamma))\leq 0} \Delta\mathcal{C}_{\ell,\mathcal{H}}(f,\mathbf{x},\eta).$$

$\square$

We then give the equivalent conditions of calibration based on inner $\ell$-risk for any symmetric hypothesis set $\mathcal{H}$.

**Lemma 28.** *Let $\mathcal{H}$ be a symmetric hypothesis set and $\ell$ be a surrogate loss function. If $\mathcal{X}_2 = \varnothing$, any loss $\ell$ is $\mathcal{H}$-calibrated with respect to $\ell_\gamma$. If $\mathcal{X}_2 \neq \varnothing$, then $\ell$ is $\mathcal{H}$-calibrated with respect to $\ell_\gamma$ if and only if for any $\mathbf{x} \in \mathcal{X}_2$,*

$$\inf_{f\in\mathcal{H}:\ \underline{M}(f,\mathbf{x},\gamma)\leq 0\leq\overline{M}(f,\mathbf{x},\gamma)} \mathcal{C}_\ell(f,\mathbf{x},\tfrac{1}{2}) > \inf_{f\in\mathcal{H}} \mathcal{C}_\ell(f,\mathbf{x},\tfrac{1}{2}),\ and$$

$$\inf_{f\in\mathcal{H}:\ \underline{M}(f,\mathbf{x},\gamma)\leq 0} \mathcal{C}_\ell(f,\mathbf{x},\eta) > \inf_{f\in\mathcal{H}} \mathcal{C}_\ell(f,\mathbf{x},\eta)\ for\ all\ \eta \in (\tfrac{1}{2},1],\ and$$

$$\inf_{f\in\mathcal{H}:\ \overline{M}(f,\mathbf{x},\gamma)\geq 0} \mathcal{C}_\ell(f,\mathbf{x},\eta) > \inf_{f\in\mathcal{H}} \mathcal{C}_\ell(f,\mathbf{x},\eta)\ for\ all\ \eta \in [0,\tfrac{1}{2}).$$

*where $\mathcal{X}_2 = \{\mathbf{x} \in \mathcal{X}:\ there\ exists\ f' \in \mathcal{H}\ such\ that\ \underline{M}(f',\mathbf{x},\gamma) > 0\}$.*

*Proof.* Let $\delta_{\max}$ be the calibration function of $(\ell,\ell_\gamma)$ given hypothesis set $\mathcal{H}$. By Lemma 27,

$$\delta_{\max}(\epsilon,\mathbf{x},\eta) = \begin{cases} +\infty & \text{if } \mathbf{x} \in \mathcal{X}_1 \text{ or } \mathbf{x} \in \mathcal{X}_2,\ \epsilon > \max\{\eta,1-\eta\}, \\ \inf\limits_{f\in\mathcal{H}:\ \underline{M}(f,\mathbf{x},\gamma)\leq 0\leq\overline{M}(f,\mathbf{x},\gamma)} \Delta\mathcal{C}_{\ell,\mathcal{H}}(f,\mathbf{x},\eta) & \text{if } \mathbf{x} \in \mathcal{X}_2,\ |2\eta-1| < \epsilon \leq \max\{\eta,1-\eta\}, \\ \inf\limits_{f\in\mathcal{H}:\ \underline{M}(f,\mathbf{x},\gamma)\leq 0\leq\overline{M}(f,\mathbf{x},\gamma)\ \text{or}\ (2\eta-1)(\underline{M}(f,\mathbf{x},\gamma))\leq 0} \Delta\mathcal{C}_{\ell,\mathcal{H}}(f,\mathbf{x},\eta) & \text{if } \mathbf{x} \in \mathcal{X}_2,\ \epsilon \leq |2\eta-1|, \end{cases}$$

where $\mathcal{X}_1 = \{\mathbf{x} \in \mathcal{X}: \underline{M}(f,\mathbf{x},\gamma) \leq 0 \leq \overline{M}(f,\mathbf{x},\gamma),\ \forall f \in \mathcal{H}\}$, $\mathcal{X}_2 = \{\mathbf{x} \in \mathcal{X}:\ \text{there exists } f' \in \mathcal{H} \text{ such that } \underline{M}(f',\mathbf{x},\gamma) > 0\}$ and $\mathcal{X} = \mathcal{X}_1 \cup \mathcal{X}_2$, $\mathcal{X}_1 \cap \mathcal{X}_2 = \varnothing$. By Proposition 4, $\ell$ is $\mathcal{H}$-calibrated with respect to $\ell_\gamma$ if and only if its calibration function $\delta_{\max}$ satisfies $\delta_{\max}(\epsilon,\mathbf{x},\eta) > 0$ for all $\mathbf{x} \in \mathcal{X}$, $\eta \in [0,1]$ and $\epsilon > 0$. Since $\delta(\epsilon,\mathbf{x},\eta) = +\infty > 0$ when $\mathbf{x} \notin \mathcal{X}_2$, any loss $\ell$ is $\mathcal{H}$-calibrated with respect to $\ell_\gamma$ when $\mathcal{X}_2 = \varnothing$. Furtheremore, when $\mathcal{X}_2 \neq \varnothing$, we only need to analyze $\delta(\epsilon,\mathbf{x},\eta)$ when $\mathbf{x} \in \mathcal{X}_2$. For $\eta = \tfrac{1}{2}$, we have for any $\mathbf{x} \in \mathcal{X}_2$,

$$\delta_{\max}(\epsilon,\mathbf{x},\tfrac{1}{2}) > 0 \text{ for all } \epsilon > 0 \Leftrightarrow \inf_{f\in\mathcal{H}:\ \underline{M}(f,\mathbf{x},\gamma)\leq 0\leq\overline{M}(f,\mathbf{x},\gamma)} \mathcal{C}_\ell(f,\mathbf{x},\tfrac{1}{2}) > \inf_{f\in\mathcal{H}} \mathcal{C}_\ell(f,\mathbf{x},\tfrac{1}{2}). \quad (21)$$

For $1 \geq \eta > \tfrac{1}{2}$, we have $|2\eta-1| = 2\eta-1$, $\max\{\eta,1-\eta\} = \eta$, and

$$\inf_{f\in\mathcal{H}:\ \underline{M}(f,\mathbf{x},\gamma)\leq 0\leq\overline{M}(f,\mathbf{x},\gamma)\ \text{or}\ (2\eta-1)(\underline{M}(f,\mathbf{x},\gamma))\leq 0} \Delta\mathcal{C}_{\ell,\mathcal{H}}(f,\mathbf{x},\eta) = \inf_{f\in\mathcal{H}:\ \underline{M}(f,\mathbf{x},\gamma)\leq 0} \Delta\mathcal{C}_{\ell,\mathcal{H}}(f,\mathbf{x},\eta).$$

Therefore, $\delta_{\max}(\epsilon,\mathbf{x},\tfrac{1}{2}) > 0$ for all $\mathbf{x} \in \mathcal{X}_2,\epsilon > 0$ and $\eta \in (\tfrac{1}{2},1]$ if and only if for all $\mathbf{x} \in \mathcal{X}_2$,

$$\begin{cases} \inf\limits_{f\in\mathcal{H}:\ \underline{M}(f,\mathbf{x},\gamma)\leq 0\leq\overline{M}(f,\mathbf{x},\gamma)} \mathcal{C}_\ell(f,\mathbf{x},\eta) > \inf\limits_{f\in\mathcal{H}} \mathcal{C}_\ell(f,\mathbf{x},\eta) & \text{for all } \eta \in (\tfrac{1}{2},1] \text{ such that } 2\eta-1 < \epsilon \leq \eta, \\ \inf\limits_{f\in\mathcal{H}:\ \underline{M}(f,\mathbf{x},\gamma)\leq 0} \mathcal{C}_\ell(f,\mathbf{x},\eta) > \inf\limits_{f\in\mathcal{H}} \mathcal{C}_\ell(f,\mathbf{x},\eta) & \text{for all } \eta \in (\tfrac{1}{2},1] \text{ such that } \epsilon \leq 2\eta-1, \end{cases}$$

for all $\epsilon > 0$, which is equivalent to for all $\mathbf{x} \in \mathcal{X}_2$,

$$\begin{cases} \inf\limits_{f\in\mathcal{H}:\ \underline{M}(f,\mathbf{x},\gamma)\leq 0\leq\overline{M}(f,\mathbf{x},\gamma)} \mathcal{C}_\ell(f,\mathbf{x},\eta) > \inf\limits_{f\in\mathcal{H}} \mathcal{C}_\ell(f,\mathbf{x},\eta) & \text{for all } \eta \in (\tfrac{1}{2},1] \text{ such that } \epsilon \leq \eta < \tfrac{\epsilon+1}{2}, \\ \inf\limits_{f\in\mathcal{H}:\ \underline{M}(f,\mathbf{x},\gamma)\leq 0} \mathcal{C}_\ell(f,\mathbf{x},\eta) > \inf\limits_{f\in\mathcal{H}} \mathcal{C}_\ell(f,\mathbf{x},\eta) & \text{for all } \eta \in (\tfrac{1}{2},1] \text{ such that } \tfrac{\epsilon+1}{2} \leq \eta, \end{cases} \quad (22)$$

for all $\epsilon > 0$. Observe that

$$\left\{\eta \in (\tfrac{1}{2},1] \Big| \epsilon \leq \eta < \frac{\epsilon+1}{2}, \epsilon > 0\right\} = \left\{\tfrac{1}{2} < \eta \leq 1\right\},\ \text{and}$$

$$\left\{\eta \in (\tfrac{1}{2},1] \Big| \frac{\epsilon+1}{2} \leq \eta, \epsilon > 0\right\} = \left\{\tfrac{1}{2} < \eta \leq 1\right\},\ \text{and}$$

$$\inf_{f\in\mathcal{H}:\ \underline{M}(f,\mathbf{x},\gamma)\leq 0\leq\overline{M}(f,\mathbf{x},\gamma)} \mathcal{C}_\ell(f,\mathbf{x},\eta) \geq \inf_{f\in\mathcal{H}:\ \underline{M}(f,\mathbf{x},\gamma)\leq 0} \mathcal{C}_\ell(f,\mathbf{x},\eta)\ \text{for all } \eta.$$

Therefore, we reduce the above condition (22) as for all $\mathbf{x} \in \mathcal{X}_2$,

$$\inf_{f \in \mathcal{H}: \underline{M}(f,\mathbf{x},\gamma) \leq 0} \mathcal{C}_\ell(f,\mathbf{x},\eta) > \inf_{f \in \mathcal{H}} \mathcal{C}_\ell(f,\mathbf{x},\eta) \text{ for all } \eta \in (\tfrac{1}{2},1]. \tag{23}$$

For $\frac{1}{2} > \eta \geq 0$, we have $|2\eta - 1| = 1 - 2\eta$, $\max\{\eta, 1-\eta\} = 1 - \eta$, and

$$\inf_{f \in \mathcal{H}: \underline{M}(f,\mathbf{x},\gamma) \leq 0 \leq \overline{M}(f,\mathbf{x},\gamma) \text{ or } (2\eta-1)(\underline{M}(f,\mathbf{x},\gamma)) \leq 0} \Delta\mathcal{C}_{\ell,\mathcal{H}}(f,\mathbf{x},\eta) = \inf_{f \in \mathcal{H}: \overline{M}(f,\mathbf{x},\gamma) \geq 0} \Delta\mathcal{C}_{\ell,\mathcal{H}}(f,\mathbf{x},\eta).$$

Therefore, $\delta_{\max}(\epsilon, \mathbf{x}, \frac{1}{2}) > 0$ for all $\mathbf{x} \in \mathcal{X}_2, \epsilon > 0$ and $\eta \in [0, \frac{1}{2})$ if and only if for all $\mathbf{x} \in \mathcal{X}_2$,

$$\begin{cases} \inf\limits_{f \in \mathcal{H}: \underline{M}(f,\mathbf{x},\gamma) \leq 0 \leq \overline{M}(f,\mathbf{x},\gamma)} \mathcal{C}_\ell(f,\mathbf{x},\eta) > \inf\limits_{f \in \mathcal{H}} \mathcal{C}_\ell(f,\mathbf{x},\eta) & \text{for all } \eta \in [0,\tfrac{1}{2}) \text{ such that } 1 - 2\eta < \epsilon \leq 1 - \eta, \\ \inf\limits_{f \in \mathcal{H}: \overline{M}(f,\mathbf{x},\gamma) \geq 0} \mathcal{C}_\ell(f,\mathbf{x},\eta) > \inf\limits_{f \in \mathcal{H}} \mathcal{C}_\ell(f,\mathbf{x},\eta) & \text{for all } \eta \in [0,\tfrac{1}{2}) \text{ such that } \epsilon \leq 1 - 2\eta, \end{cases}$$

for all $\epsilon > 0$, which is equivalent to for all $\mathbf{x} \in \mathcal{X}_2$,

$$\begin{cases} \inf\limits_{f \in \mathcal{H}: \underline{M}(f,\mathbf{x},\gamma) \leq 0 \leq \overline{M}(f,\mathbf{x},\gamma)} \mathcal{C}_\ell(f,\mathbf{x},\eta) > \inf\limits_{f \in \mathcal{H}} \mathcal{C}_\ell(f,\mathbf{x},\eta) & \text{for all } \eta \in [0,\tfrac{1}{2}) \text{ such that } \tfrac{1-\epsilon}{2} < \eta \leq 1 - \epsilon, \\ \inf\limits_{f \in \mathcal{H}: \overline{M}(f,\mathbf{x},\gamma) \geq 0} \mathcal{C}_\ell(f,\mathbf{x},\eta) > \inf\limits_{f \in \mathcal{H}} \mathcal{C}_\ell(f,\mathbf{x},\eta) & \text{for all } \eta \in [0,\tfrac{1}{2}) \text{ such that } \eta \leq \tfrac{1-\epsilon}{2}, \end{cases} \tag{24}$$

for all $\epsilon > 0$. Observe that

$$\left\{ \eta \in [0, \tfrac{1}{2}) \,\Big|\, \tfrac{1-\epsilon}{2} < \eta \leq 1 - \epsilon, \epsilon > 0 \right\} = \left\{ 0 \leq \eta < \tfrac{1}{2} \right\}, \text{ and}$$

$$\left\{ \eta \in [0, \tfrac{1}{2}) \,\Big|\, \eta \leq \tfrac{1-\epsilon}{2}, \epsilon > 0 \right\} = \left\{ 0 \leq \eta < \tfrac{1}{2} \right\}, \text{ and}$$

$$\inf_{f \in \mathcal{H}: \underline{M}(f,\mathbf{x},\gamma) \leq 0 \leq \overline{M}(f,\mathbf{x},\gamma)} \mathcal{C}_\ell(f,\mathbf{x},\eta) \geq \inf_{f \in \mathcal{H}: \overline{M}(f,\mathbf{x},\gamma) \geq 0} \mathcal{C}_\ell(f,\mathbf{x},\eta) \text{ for all } \eta.$$

Therefore, we reduce the above condition (24) as for all $\mathbf{x} \in \mathcal{X}_2$,

$$\inf_{f \in \mathcal{H}: \overline{M}(f,\mathbf{x},\gamma) \geq 0} \mathcal{C}_\ell(f,\mathbf{x},\eta) > \inf_{f \in \mathcal{H}} \mathcal{C}_\ell(f,\mathbf{x},\eta) \text{ for all } \eta \in [0,\tfrac{1}{2}). \tag{25}$$

To sum up, by (21), (23) and (25), we conclude the proof. $\qquad\square$

Since $\mathcal{H}_{\mathrm{lin}}$ is a symmetric hypothesis set, we could make use of Lemma 27 and Lemma 28 for proving Theorem 12.

**Theorem 12.** *Let a margin-based loss $\phi$ be bounded, continuous, non-increasing, and satisfy the property that $\bar{\mathcal{C}}_\phi(t,\eta)$ is quasi-concave in $t \in \mathbb{R}$ for all $\eta \in [0,1]$. Assume that $\phi(-t) > \phi(t)$ for any $\gamma < t \leq 1$. Then $\phi$ is $\mathcal{H}_{\mathrm{lin}}$-calibrated with respect to $\ell_\gamma$ if and only if for any $\gamma < t \leq 1$,*

$$\phi(\gamma) + \phi(-\gamma) > \phi(t) + \phi(-t). \tag{10}$$

*Proof.* As shown by Awasthi et al. (2020), for $f \in \mathcal{H}_{\mathrm{lin}} = \{\mathbf{x} \to \mathbf{w} \cdot \mathbf{x} \mid \|\mathbf{w}\| = 1\}$,

$$\underline{M}(f,\mathbf{x},\gamma) = \inf_{\mathbf{x}':\|\mathbf{x}-\mathbf{x}'\| \leq \gamma} f(\mathbf{x}') = \inf_{\mathbf{x}':\|\mathbf{x}-\mathbf{x}'\| \leq \gamma} (\mathbf{w} \cdot \mathbf{x}') = \mathbf{w} \cdot \mathbf{x} - \gamma\|\mathbf{w}\| = f(\mathbf{x}) - \gamma,$$

$$\overline{M}(f,\mathbf{x},\gamma) = -\inf_{\mathbf{x}':\|\mathbf{x}-\mathbf{x}'\| \leq \gamma} -f(\mathbf{x}') = -\inf_{\mathbf{x}':\|\mathbf{x}-\mathbf{x}'\| \leq \gamma} (-\mathbf{w} \cdot \mathbf{x}') = \mathbf{w} \cdot \mathbf{x} + \gamma\|\mathbf{w}\| = f(\mathbf{x}) + \gamma.$$

Thus for $\mathcal{H}_{\mathrm{lin}}$, $\mathcal{X}_2 = \{\mathbf{x} \in \mathcal{X} : \text{there exists } f' \in \mathcal{H}_{\mathrm{lin}} \text{ such that } \underline{M}(f',\mathbf{x},\gamma) > 0\} = \{\mathbf{x} \in \mathcal{X} : \text{there exists } f' \in \mathcal{H}_{\mathrm{lin}} \text{ such that } f'(\mathbf{x}) > \gamma\} = \{\mathbf{x} : \gamma < \|\mathbf{x}\| \leq 1\}$ since $f(\mathbf{x}) = \mathbf{w} \cdot \mathbf{x} \in [-\|\mathbf{x}\|, \|\mathbf{x}\|]$ when $f \in \mathcal{H}_{\mathrm{lin}}$. Note $\mathcal{H}_{\mathrm{lin}}$ is a symmetric hypothesis set. Therefore, by Lemma 28, $\phi$ is $\mathcal{H}_{\mathrm{lin}}$-calibrated with respect to $\ell_\gamma$ if and only if for any $\mathbf{x} \in \mathcal{X}$ such that $\gamma < \|\mathbf{x}\| \leq 1$,

$$\inf_{f \in \mathcal{H}_{\mathrm{lin}}: |f(\mathbf{x})| \leq \gamma} \mathcal{C}_\phi(f,\mathbf{x},\tfrac{1}{2}) > \inf_{f \in \mathcal{H}_{\mathrm{lin}}} \mathcal{C}_\phi(f,\mathbf{x},\tfrac{1}{2}), \text{ and}$$

$$\inf_{f \in \mathcal{H}_{\mathrm{lin}}: f(\mathbf{x}) \leq \gamma} \mathcal{C}_\phi(f,\mathbf{x},\eta) > \inf_{f \in \mathcal{H}_{\mathrm{lin}}} \mathcal{C}_\phi(f,\mathbf{x},\eta) \text{ for all } \eta \in (\tfrac{1}{2},1], \text{ and} \tag{26}$$

$$\inf_{f \in \mathcal{H}_{\mathrm{lin}}: f(\mathbf{x}) \geq -\gamma} \mathcal{C}_\phi(f,\mathbf{x},\eta) > \inf_{f \in \mathcal{H}_{\mathrm{lin}}} \mathcal{C}_\phi(f,\mathbf{x},\eta) \text{ for all } \eta \in [0,\tfrac{1}{2}).$$

By the definition of inner risk (4), the inner $\phi$-risk is
$$\mathcal{C}_\phi(f, \mathbf{x}, \eta) = \eta\phi(f(\mathbf{x})) + (1-\eta)\phi(-f(\mathbf{x})).$$
Note $f(\mathbf{x}) = \mathbf{w} \cdot \mathbf{x} \in [-\|\mathbf{x}\|, \|\mathbf{x}\|]$ when $f \in \mathcal{H}_{\text{lin}}$. Therefore, (26) is equivalent to for any $\mathbf{x} \in \mathcal{X}$ such that $\gamma < \|\mathbf{x}\| \leq 1$,

$$\inf_{-\gamma \leq t \leq \gamma} \bar{\mathcal{C}}_\phi(t, \frac{1}{2}) > \inf_{-\|\mathbf{x}\| \leq t \leq \|\mathbf{x}\|} \bar{\mathcal{C}}_\phi(t, \frac{1}{2}), \text{ and}$$

$$\inf_{-\|\mathbf{x}\| \leq t \leq \gamma} \bar{\mathcal{C}}_\phi(t, \eta) > \inf_{-\|\mathbf{x}\| \leq t \leq \|\mathbf{x}\|} \bar{\mathcal{C}}_\phi(t, \eta) \text{ for all } \eta \in (\frac{1}{2}, 1], \text{ and} \qquad (27)$$

$$\inf_{-\gamma \leq t \leq \|\mathbf{x}\|} \bar{\mathcal{C}}_\phi(t, \eta) > \inf_{-\|\mathbf{x}\| \leq t \leq \|\mathbf{x}\|} \bar{\mathcal{C}}_\phi(t, \eta) \text{ for all } \eta \in [0, \frac{1}{2}).$$

Suppose that $\phi$ is $\mathcal{H}_{\text{lin}}$-calibrated with respect to $\ell_\gamma$. Since by Part 1 of Lemma 26,

$$\inf_{-\gamma \leq t \leq \gamma} \bar{\mathcal{C}}_\phi(t, \frac{1}{2}) = \bar{\mathcal{C}}_\phi(\gamma, \frac{1}{2}), \qquad \inf_{-\|\mathbf{x}\| \leq t \leq \|\mathbf{x}\|} \bar{\mathcal{C}}_\phi(t, \frac{1}{2}) = \bar{\mathcal{C}}_\phi(\|\mathbf{x}\|, \frac{1}{2}),$$

we obtain $\phi(\gamma) + \phi(-\gamma) = 2\bar{\mathcal{C}}_\phi(\gamma, \frac{1}{2}) > 2\bar{\mathcal{C}}_\phi(t, \frac{1}{2}) = \phi(t) + \phi(-t)$ for any $\gamma < t \leq 1$.

Now for the other direction, assume that $\phi(\gamma) + \phi(-\gamma) > \phi(t) + \phi(-t)$ for any $\gamma < t \leq 1$. For $\eta = \frac{1}{2}$, by Part 1 of Lemma 26, we obtain for any $\mathbf{x} \in \mathcal{X}$ such that $\gamma < \|\mathbf{x}\| \leq 1$,

$$\inf_{-\gamma \leq t \leq \gamma} \bar{\mathcal{C}}_\phi(t, \frac{1}{2}) = \bar{\mathcal{C}}_\phi(\gamma, \frac{1}{2}) > \bar{\mathcal{C}}_\phi(\|\mathbf{x}\|, \frac{1}{2}) = \inf_{-\|\mathbf{x}\| \leq t \leq \|\mathbf{x}\|} \bar{\mathcal{C}}_\phi(t, \frac{1}{2}).$$

For $\eta \in (\frac{1}{2}, 1]$ and any $\mathbf{x} \in \mathcal{X}$ such that $\gamma < \|\mathbf{x}\| \leq 1$,

$$\inf_{-\|\mathbf{x}\| \leq t \leq \gamma} \bar{\mathcal{C}}_\phi(t, \eta) = \min\{\bar{\mathcal{C}}_\phi(\gamma, \eta), \bar{\mathcal{C}}_\phi(-\|\mathbf{x}\|, \eta)\} \quad \text{(Part 2 of Lemma 26)}$$

$$\inf_{-\|\mathbf{x}\| \leq t \leq \|\mathbf{x}\|} \bar{\mathcal{C}}_\phi(t, \eta) = \min\{\bar{\mathcal{C}}_\phi(\|\mathbf{x}\|, \eta), \bar{\mathcal{C}}_\phi(-\|\mathbf{x}\|, \eta)\} \quad \text{(Part 2 of Lemma 26)}$$

$$= \bar{\mathcal{C}}_\phi(\|\mathbf{x}\|, \eta) \quad \text{(Part 5 of Lemma 26)}$$

Note for $\eta \in (\frac{1}{2}, 1]$ and any $\mathbf{x} \in \mathcal{X}$ such that $\gamma < \|\mathbf{x}\| \leq 1$, since $\phi$ is non-increasing,

$$\phi(\gamma) - \phi(-\gamma) - \phi(\|\mathbf{x}\|) + \phi(-\|\mathbf{x}\|) \geq \phi(\|\mathbf{x}\|) - \phi(-\|\mathbf{x}\|) - \phi(\|\mathbf{x}\|) + \phi(-\|\mathbf{x}\|) = 0.$$

Thus

$$\begin{aligned}
\bar{\mathcal{C}}_\phi(\gamma, \eta) - \bar{\mathcal{C}}_\phi(\|\mathbf{x}\|, \eta) &= \eta\phi(\gamma) + (1-\eta)\phi(-\gamma) - \eta\phi(\|\mathbf{x}\|) - (1-\eta)\phi(-\|\mathbf{x}\|) \\
&= (\phi(\gamma) - \phi(-\gamma) - \phi(\|\mathbf{x}\|) + \phi(-\|\mathbf{x}\|))\eta + \phi(-\gamma) - \phi(-\|\mathbf{x}\|) \\
&\geq (\phi(\gamma) - \phi(-\gamma) - \phi(\|\mathbf{x}\|) + \phi(-\|\mathbf{x}\|))\frac{1}{2} + \phi(-\gamma) - \phi(-\|\mathbf{x}\|) \\
&= \frac{1}{2}[\phi(\gamma) + \phi(-\gamma) - \phi(\|\mathbf{x}\|) - \phi(-\|\mathbf{x}\|)] \\
&> 0.
\end{aligned}$$

In addition, we have for $\eta \in (\frac{1}{2}, 1]$ and any $\mathbf{x} \in \mathcal{X}$ such that $\gamma < \|\mathbf{x}\| \leq 1$,

$$\bar{\mathcal{C}}_\phi(-\|\mathbf{x}\|, \eta) > \bar{\mathcal{C}}_\phi(\|\mathbf{x}\|, \eta). \quad \text{(Part 5 of Lemma 26)}$$

Therefore for $\eta \in (\frac{1}{2}, 1]$ and any $\mathbf{x} \in \mathcal{X}$ such that $\gamma < \|\mathbf{x}\| \leq 1$,

$$\inf_{-\|\mathbf{x}\| \leq t \leq \gamma} \bar{\mathcal{C}}_\phi(t, \eta) = \min\{\bar{\mathcal{C}}_\phi(\gamma, \eta), \bar{\mathcal{C}}_\phi(-\|\mathbf{x}\|, \eta)\} > \bar{\mathcal{C}}_\phi(\|\mathbf{x}\|, \eta) = \inf_{-\|\mathbf{x}\| \leq t \leq \|\mathbf{x}\|} \bar{\mathcal{C}}_\phi(t, \eta).$$

For $\eta \in [0, \frac{1}{2})$ and any $\mathbf{x} \in \mathcal{X}$ such that $\gamma < \|\mathbf{x}\| \leq 1$,

$$\inf_{-\gamma \leq t \leq \|\mathbf{x}\|} \bar{\mathcal{C}}_\phi(t, \eta) = \min\{\bar{\mathcal{C}}_\phi(-\gamma, \eta), \bar{\mathcal{C}}_\phi(\|\mathbf{x}\|, \eta)\} \quad \text{(Part 2 of Lemma 26)}$$

$$\inf_{-\|\mathbf{x}\| \leq t \leq \|\mathbf{x}\|} \bar{\mathcal{C}}_\phi(t, \eta) = \min\{\bar{\mathcal{C}}_\phi(\|\mathbf{x}\|, \eta), \bar{\mathcal{C}}_\phi(-\|\mathbf{x}\|, \eta)\} \quad \text{(Part 2 of Lemma 26)}$$

$$= \bar{\mathcal{C}}_\phi(-\|\mathbf{x}\|, \eta) \quad \text{(Part 6 of Lemma 26)}$$

Note for $\eta \in [0, \frac{1}{2})$ and any $\mathbf{x} \in \mathcal{X}$ such that $\gamma < \|\mathbf{x}\| \leq 1$, since $\phi$ is non-increasing,

$$\phi(-\gamma) - \phi(\gamma) - \phi(-\|\mathbf{x}\|) + \phi(\|\mathbf{x}\|) \leq \phi(-\|\mathbf{x}\|) - \phi(\|\mathbf{x}\|) - \phi(-\|\mathbf{x}\|) + \phi(\|\mathbf{x}\|) = 0.$$

Thus

$$
\begin{aligned}
\bar{\mathcal{C}}_\phi(-\gamma, \eta) - \bar{\mathcal{C}}_\phi(-\|\mathbf{x}\|, \eta) &= \eta\phi(-\gamma) + (1-\eta)\phi(\gamma) - \eta\phi(-\|\mathbf{x}\|) - (1-\eta)\phi(\|\mathbf{x}\|) \\
&= \left(\phi(-\gamma) - \phi(\gamma) - \phi(-\|\mathbf{x}\|) + \phi(\|\mathbf{x}\|)\right)\eta + \phi(\gamma) - \phi(\|\mathbf{x}\|) \\
&\geq \left(\phi(-\gamma) - \phi(\gamma) - \phi(-\|\mathbf{x}\|) + \phi(\|\mathbf{x}\|)\right)\frac{1}{2} + \phi(\gamma) - \phi(\|\mathbf{x}\|) \\
&= \frac{1}{2}\left[\phi(\gamma) + \phi(-\gamma) - \phi(\|\mathbf{x}\|) - \phi(-\|\mathbf{x}\|)\right] \\
&> 0.
\end{aligned}
$$

In addition, we have for $\eta \in [0, \frac{1}{2})$ and any $\mathbf{x} \in \mathcal{X}$ such that $\gamma < \|\mathbf{x}\| \leq 1$,

$$\bar{\mathcal{C}}_\phi(\|\mathbf{x}\|, \eta) > \bar{\mathcal{C}}_\phi(-\|\mathbf{x}\|, \eta). \quad \text{(Part 6 of Lemma 26)}$$

Therefore for $\eta \in [0, \frac{1}{2})$ and any $\mathbf{x} \in \mathcal{X}$ such that $\gamma < \|\mathbf{x}\| \leq 1$,

$$\inf_{-\gamma \leq t \leq \|\mathbf{x}\|} \bar{\mathcal{C}}_\phi(t, \eta) = \min\{\bar{\mathcal{C}}_\phi(-\gamma, \eta), \bar{\mathcal{C}}_\phi(\|\mathbf{x}\|, \eta)\} > \bar{\mathcal{C}}_\phi(-\|\mathbf{x}\|, \eta) = \inf_{-\|\mathbf{x}\| \leq t \leq \|\mathbf{x}\|} \bar{\mathcal{C}}_\phi(t, \eta).$$

$\square$

**Theorem 16.** *Let $\mathcal{H}$ be a symmetric hypothesis set, then $\tilde{\phi}_\rho$ is $\mathcal{H}$-calibrated with respect to $\ell_\gamma$.*

*Proof.* By Lemma 28, if $\mathcal{X}_2 = \varnothing$, $\tilde{\phi}_\rho$ is $\mathcal{H}$-calibrated with respect to $\ell_\gamma$. Next consider the case where $\mathcal{X}_2 \neq \varnothing$. By Lemma 28, $\tilde{\phi}_\rho$ is $\mathcal{H}$-calibrated with respect to $\ell_\gamma$ if and only if for all $\mathbf{x} \in \mathcal{X}_2$,

$$\inf_{f \in \mathcal{H}: \, \underline{M}(f,\mathbf{x},\gamma) \leq 0 \leq \overline{M}(f,\mathbf{x},\gamma)} \mathcal{C}_{\tilde{\phi}_\rho}(f, \mathbf{x}, \frac{1}{2}) > \inf_{f \in \mathcal{H}} \mathcal{C}_{\tilde{\phi}_\rho}(f, \mathbf{x}, \frac{1}{2}), \text{ and}$$

$$\inf_{f \in \mathcal{H}: \, \underline{M}(f,\mathbf{x},\gamma) \leq 0} \mathcal{C}_{\tilde{\phi}_\rho}(f, \mathbf{x}, \eta) > \inf_{f \in \mathcal{H}} \mathcal{C}_{\tilde{\phi}_\rho}(f, \mathbf{x}, \eta) \text{ for all } \eta \in (\frac{1}{2}, 1], \text{ and}$$

$$\inf_{f \in \mathcal{H}: \, \overline{M}(f,\mathbf{x},\gamma) \geq 0} \mathcal{C}_{\tilde{\phi}_\rho}(f, \mathbf{x}, \eta) > \inf_{f \in \mathcal{H}} \mathcal{C}_{\tilde{\phi}_\rho}(f, \mathbf{x}, \eta) \text{ for all } \eta \in [0, \frac{1}{2}),$$

where $\mathcal{X}_2 = \{\mathbf{x} \in \mathcal{X} : \text{ there exists } f' \in \mathcal{H} \text{ such that } \underline{M}(f', \mathbf{x}, \gamma) > 0\}$. As shown by Awasthi et al. (2020), $\tilde{\phi}_\rho$ has the equivalent form

$$\tilde{\phi}_\rho(f, \mathbf{x}, y) = \phi_\rho\left(\inf_{\mathbf{x}': \|\mathbf{x}-\mathbf{x}'\| \leq \gamma} (yf(\mathbf{x}'))\right).$$

Thus by the definition of inner risk (4), the inner $\tilde{\phi}_\rho$-risk is

$$\mathcal{C}_{\tilde{\phi}_\rho}(f, \mathbf{x}, \eta) = \eta\phi_\rho(\underline{M}(f, \mathbf{x}, \gamma)) + (1-\eta)\phi_\rho(-\overline{M}(f, \mathbf{x}, \gamma)).$$

For any $\mathbf{x} \in \mathcal{X}_2$, let $M_\mathbf{x} = \sup_{f \in \mathcal{H}} \underline{M}(f, \mathbf{x}, \gamma) > 0$. Since $\mathcal{H}$ is symmetric, we have $-M_\mathbf{x} = \inf_{f \in \mathcal{H}} \overline{M}(f, \mathbf{x}, \gamma) < 0$. Since $\phi_\rho$ is continuous, for any $\mathbf{x} \in \mathcal{X}_2$ and $\epsilon > 0$, there exists $f_\mathbf{x}^\epsilon \in \mathcal{H}$ such that $\phi_\rho(\underline{M}(f_\mathbf{x}^\epsilon, \mathbf{x}, \gamma)) < \phi_\rho(M_\mathbf{x}) + \epsilon$ and $\overline{M}(f_\mathbf{x}^\epsilon, \mathbf{x}, \gamma) \geq \underline{M}(f_\mathbf{x}^\epsilon, \mathbf{x}, \gamma) > 0$, $\underline{M}(-f_\mathbf{x}^\epsilon, \mathbf{x}, \gamma) \leq \overline{M}(-f_\mathbf{x}^\epsilon, \mathbf{x}, \gamma) = -\underline{M}(f_\mathbf{x}^\epsilon, \mathbf{x}, \gamma) < 0$. Next we analyze three cases:

- When $\eta = \frac{1}{2}$, since $\phi_\rho$ is non-increasing,

$$
\begin{aligned}
&\inf_{f \in \mathcal{H}: \, \underline{M}(f,\mathbf{x},\gamma) \leq 0 \leq \overline{M}(f,\mathbf{x},\gamma)} \mathcal{C}_{\tilde{\phi}_\rho}(f, \mathbf{x}, \frac{1}{2}) \\
&= \inf_{f \in \mathcal{H}: \, \underline{M}(f,\mathbf{x},\gamma) \leq 0 \leq \overline{M}(f,\mathbf{x},\gamma)} \frac{1}{2}\phi_\rho(\underline{M}(f, \mathbf{x}, \gamma)) + \frac{1}{2}\phi_\rho(-\overline{M}(f, \mathbf{x}, \gamma)) \\
&\geq \frac{1}{2}\phi_\rho(0) + \frac{1}{2}\phi_\rho(0) = \phi_\rho(0) = 1.
\end{aligned}
$$

For any $\mathbf{x} \in \mathcal{X}_2$, there exists $f' \in \mathcal{H}$ such that $\underline{M}(f', \mathbf{x}, \gamma) > 0$ and $-\overline{M}(f', \mathbf{x}, \gamma) \leq -\underline{M}(f', \mathbf{x}, \gamma) < 0$, we obtain

$$\mathcal{C}_{\tilde{\phi}_\rho}(f', \mathbf{x}, \tfrac{1}{2}) = \frac{1}{2}\phi_\rho(\underline{M}(f', \mathbf{x}, \gamma)) + \frac{1}{2}\phi_\rho(-\overline{M}(f', \mathbf{x}, \gamma)) = \frac{1}{2}\phi_\rho(\underline{M}(f', \mathbf{x}, \gamma)) + \frac{1}{2} < 1.$$

Therefore for any $\mathbf{x} \in \mathcal{X}_2$,

$$\inf_{f \in \mathcal{H}} \mathcal{C}_{\tilde{\phi}_\rho}(f, \mathbf{x}, \tfrac{1}{2}) \leq \mathcal{C}_{\tilde{\phi}_\rho}(f', \mathbf{x}, \tfrac{1}{2}) < 1 \leq \inf_{f \in \mathcal{H}:\ \underline{M}(f,\mathbf{x},\gamma) \leq 0 \leq \overline{M}(f,\mathbf{x},\gamma)} \mathcal{C}_{\tilde{\phi}_\rho}(f, \mathbf{x}, \tfrac{1}{2}). \quad (28)$$

- When $\eta \in (\tfrac{1}{2}, 1]$, since $\phi_\rho$ is non-increasing, for any $\mathbf{x} \in \mathcal{X}_2$,

$$\inf_{f \in \mathcal{H}:\ \underline{M}(f,\mathbf{x},\gamma) \leq 0} \mathcal{C}_{\tilde{\phi}_\rho}(f, \mathbf{x}, \eta) = \inf_{f \in \mathcal{H}:\ \underline{M}(f,\mathbf{x},\gamma) \leq 0} \eta\phi_\rho(\underline{M}(f, \mathbf{x}, \gamma)) + (1-\eta)\phi_\rho(-\overline{M}(f, \mathbf{x}, \gamma))$$

$$= \eta + \inf_{f \in \mathcal{H}:\ \underline{M}(f,\mathbf{x},\gamma) \leq 0} (1-\eta)\phi_\rho(-\overline{M}(f, \mathbf{x}, \gamma))$$

$$\geq \eta + (1-\eta)\phi_\rho(M_\mathbf{x}).$$

On the other hand, for any $\mathbf{x} \in \mathcal{X}_2$ and $\epsilon > 0$,

$$\mathcal{C}_{\tilde{\phi}_\rho}(f_\mathbf{x}^\epsilon, \mathbf{x}, \eta) = \eta\phi_\rho(\underline{M}(f_\mathbf{x}^\epsilon, \mathbf{x}, \gamma)) + (1-\eta)\phi_\rho(-\overline{M}(f_\mathbf{x}^\epsilon, \mathbf{x}, \gamma))$$

$$< \eta\phi_\rho(M_\mathbf{x}) + \epsilon + (1-\eta).$$

Since $\eta > \tfrac{1}{2}$ and $M_\mathbf{x} > 0$, we have

$$\inf_{f \in \mathcal{H}:\ \underline{M}(f,\mathbf{x},\gamma) \leq 0} \mathcal{C}_{\tilde{\phi}_\rho}(f, \mathbf{x}, \eta) - \mathcal{C}_{\tilde{\phi}_\rho}(f_\mathbf{x}^\epsilon, \mathbf{x}, \eta)$$

$$> [\eta + (1-\eta)\phi_\rho(M_\mathbf{x})] - [\eta\phi_\rho(M_\mathbf{x}) + \epsilon + (1-\eta)]$$

$$= (2\eta - 1)(1 - \phi_\rho(M_\mathbf{x})) - \epsilon$$

$$> 0,$$

where we take $0 < \epsilon < (2\eta - 1)(1 - \phi_\rho(M_\mathbf{x}))$.

Therefore for any $\eta \in (\tfrac{1}{2}, 1]$ and $\mathbf{x} \in \mathcal{X}_2$, there exists $0 < \epsilon < (2\eta - 1)(1 - \phi_\rho(M_\mathbf{x}))$ such that

$$\inf_{f \in \mathcal{H}} \mathcal{C}_{\tilde{\phi}_\rho}(f, \mathbf{x}, \eta) \leq \mathcal{C}_{\tilde{\phi}_\rho}(f_\mathbf{x}^\epsilon, \mathbf{x}, \eta) < \inf_{f \in \mathcal{H}:\ \underline{M}(f,\mathbf{x},\gamma) \leq 0} \mathcal{C}_{\tilde{\phi}_\rho}(f, \mathbf{x}, \eta). \quad (29)$$

- When $\eta \in [0, \tfrac{1}{2})$, since $\phi_\rho$ is non-increasing, for any $\mathbf{x} \in \mathcal{X}_2$,

$$\inf_{f \in \mathcal{H}:\ \overline{M}(f,\mathbf{x},\gamma) \geq 0} \mathcal{C}_{\tilde{\phi}_\rho}(f, \mathbf{x}, \eta) = \inf_{f \in \mathcal{H}:\ \overline{M}(f,\mathbf{x},\gamma) \geq 0} \eta\phi_\rho(\underline{M}(f, \mathbf{x}, \gamma)) + (1-\eta)\phi_\rho(-\overline{M}(f, \mathbf{x}, \gamma))$$

$$= 1 - \eta + \inf_{f \in \mathcal{H}:\ \overline{M}(f,\mathbf{x},\gamma) \geq 0} \eta\phi_\rho(\underline{M}(f, \mathbf{x}, \gamma))$$

$$\geq 1 - \eta + \eta\phi_\rho(M_\mathbf{x})$$

On the other hand, for any $\mathbf{x} \in \mathcal{X}_2$ and $\epsilon > 0$,

$$\mathcal{C}_{\tilde{\phi}_\rho}(-f_\mathbf{x}^\epsilon, \mathbf{x}, \eta) = \eta\phi_\rho(\underline{M}(-f_\mathbf{x}^\epsilon, \mathbf{x}, \gamma)) + (1-\eta)\phi_\rho(-\overline{M}(-f_\mathbf{x}^\epsilon, \mathbf{x}, \gamma))$$

$$= \eta + (1-\eta)\phi_\rho(\underline{M}(f_\mathbf{x}^\epsilon, \mathbf{x}, \gamma))$$

$$< \eta + (1-\eta)\phi_\rho(M_\mathbf{x}) + \epsilon$$

Since $\eta < \tfrac{1}{2}$ and $M_\mathbf{x} > 0$, we have

$$\inf_{f \in \mathcal{H}:\ \overline{M}(f,\mathbf{x},\gamma) \geq 0} \mathcal{C}_{\tilde{\phi}_\rho}(f, \mathbf{x}, \eta) - \mathcal{C}_{\tilde{\phi}_\rho}(-f_\mathbf{x}^\epsilon, \mathbf{x}, \eta)$$

$$> [1 - \eta + \eta\phi_\rho(M_\mathbf{x})] - [\eta + (1-\eta)\phi_\rho(M_\mathbf{x}) + \epsilon]$$

$$= (1 - 2\eta)(1 - \phi_\rho(M_\mathbf{x})) - \epsilon$$

$$> 0$$

where we take $0 < \epsilon < (1 - 2\eta)(1 - \phi_\rho(M_\mathbf{x}))$.

Therefore for any $\eta \in [0, \tfrac{1}{2})$ and $\mathbf{x} \in \mathcal{X}_2$, there exists $0 < \epsilon < (1 - 2\eta)(1 - \phi_\rho(M_\mathbf{x}))$ such that

$$\inf_{f \in \mathcal{H}} \mathcal{C}_{\tilde{\phi}_\rho}(f, \mathbf{x}, \eta) \leq \mathcal{C}_{\tilde{\phi}_\rho}(-f_\mathbf{x}^\epsilon, \mathbf{x}, \eta) < \inf_{f \in \mathcal{H}:\ \overline{M}(f,\mathbf{x},\gamma) \geq 0} \mathcal{C}_{\tilde{\phi}_\rho}(f, \mathbf{x}, \eta). \quad (30)$$

To sum up, by (28), (29) and (30), we conclude that $\tilde{\phi}_\rho$ is $\mathcal{H}$-calibrated with respect to $\ell_\gamma$. $\qquad \square$

## E.5   Proof of Theorem 13 and Corollary 14

As shown by Awasthi et al. (2020), for $f \in \mathcal{H}_g$, the adversarial $0/1$ loss has the equivalent form

$$\ell_\gamma(f, \mathbf{x}, y) = \mathbb{1}_{\inf_{\mathbf{x}': \|\mathbf{x}-\mathbf{x}'\| \le \gamma} (yg(\mathbf{w} \cdot \mathbf{x}') + by) \le 0} = \mathbb{1}_{yg(\mathbf{w} \cdot \mathbf{x} - \gamma y \|\mathbf{w}\|) + by \le 0} = \mathbb{1}_{yg(\mathbf{w} \cdot \mathbf{x} - \gamma y) + by \le 0}. \qquad (31)$$

The proofs of Theorem 13 will closely follow the proofs of Theorem 12 and Theorem 16. We will first prove Lemma 29 and Lemma 30 analogous to Lemma 27 and Lemma 28 respectively. Without loss of generality, assume that $g$ is continuous and satisfies $g(-1-\gamma) + G > 0$, $g(1+\gamma) - G < 0$. Then observe that $g(-\gamma) + G > 0$, $g(\gamma) - G < 0$ since $g$ is non-decreasing.

**Lemma 29.** *For a surrogate loss $\ell$ and hypothesis set $\mathcal{H}_g$, the calibration function of losses $(\ell, \ell_\gamma)$ is*

$$\delta_{\max}(\epsilon, \mathbf{x}, \eta) = \begin{cases} +\infty & \text{if } \epsilon > \max\{\eta, 1-\eta\}, \\ \inf_{f \in \mathcal{H}_g: \, g(\mathbf{w} \cdot \mathbf{x} - \gamma) + b \le 0 \le g(\mathbf{w} \cdot \mathbf{x} + \gamma) + b} \Delta\mathcal{C}_{\ell, \mathcal{H}_g}(f, \mathbf{x}, \eta) & \text{if } |2\eta - 1| < \epsilon \le \max\{\eta, 1-\eta\}, \\ \inf_{f \in \mathcal{H}_g: \, g(\mathbf{w} \cdot \mathbf{x} - \gamma) + b \le 0 \le g(\mathbf{w} \cdot \mathbf{x} + \gamma) + b \text{ or } (2\eta-1)[g(\mathbf{w} \cdot \mathbf{x} - \gamma) + b] \le 0} \Delta\mathcal{C}_{\ell, \mathcal{H}_g}(f, \mathbf{x}, \eta) & \text{if } \epsilon \le |2\eta - 1|. \end{cases}$$

*Proof.* As with the proof of Lemma 27, we first characterize the inner $\ell$-risk and minimal inner $\ell_\gamma$-risk for $\mathcal{H}_g$. By the definition of inner risk (4) and equivalent form of adversarial 0-1 loss $\ell_\gamma$ for $\mathcal{H}_g$ (31), the inner $\ell_\gamma$-risk is

$$\mathcal{C}_{\ell_\gamma}(f, \mathbf{x}, \eta) = \eta \mathbb{1}_{g(\mathbf{w} \cdot \mathbf{x} - \gamma) + b \le 0} + (1-\eta) \mathbb{1}_{g(\mathbf{w} \cdot \mathbf{x} + \gamma) + b \ge 0}$$

$$= \begin{cases} 1 & \text{if } g(\mathbf{w} \cdot \mathbf{x} - \gamma) + b \le 0 \le g(\mathbf{w} \cdot \mathbf{x} + \gamma) + b, \\ \eta & \text{if } g(\mathbf{w} \cdot \mathbf{x} + \gamma) + b < 0, \\ 1 - \eta & \text{if } g(\mathbf{w} \cdot \mathbf{x} - \gamma) + b > 0. \end{cases}$$

where we used the fact that $g$ is non-decreasing and $g(\mathbf{w} \cdot \mathbf{x} - \gamma) \le g(\mathbf{w} \cdot \mathbf{x} + \gamma)$. Note for any $\mathbf{x} \in \mathcal{X}$, $\mathbf{w} \cdot \mathbf{x} \in [-\|\mathbf{x}\|, \|\mathbf{x}\|]$. Thus we have $g(\mathbf{w} \cdot \mathbf{x} - \gamma) + b \in [g(-\|\mathbf{x}\| - \gamma) - G, g(\|\mathbf{x}\| - \gamma) + G]$ and $g(\mathbf{w} \cdot \mathbf{x} + \gamma) + b \in [g(-\|\mathbf{x}\| + \gamma) - G, g(\|\mathbf{x}\| + \gamma) + G]$ since $g$ is non-decreasing. By the fact that $g(-\gamma) + G > 0$ and $g(\gamma) - G < 0$, we obtain the minimal inner $\ell_\gamma$-risk, which is for any $\mathbf{x} \in \mathcal{X}$,

$$\mathcal{C}^*_{\ell_\gamma, \mathcal{H}_g}(\mathbf{x}, \eta) = \min\{\eta, 1-\eta\}.$$

As with the derivation of $\Delta\mathcal{C}_{\ell_\gamma, \mathcal{H}}(f, \mathbf{x}, \eta)$ (20), we derive $\Delta\mathcal{C}_{\ell_\gamma, \mathcal{H}_g}(f, \mathbf{x}, \eta)$ as follows. By the observation (11), for any $\mathbf{x} \in \mathcal{X}$, for $f \in \mathcal{H}_g$ such that $g(\mathbf{w} \cdot \mathbf{x} - \gamma) + b \le 0 \le g(\mathbf{w} \cdot \mathbf{x} + \gamma) + b$, $\Delta\mathcal{C}_{\ell_\gamma, \mathcal{H}_g}(f, \mathbf{x}, \eta) = 1 - \min\{\eta, 1-\eta\} = \max\{\eta, 1-\eta\}$; for $f \in \mathcal{H}_g$ such that $g(\mathbf{w} \cdot \mathbf{x} + \gamma) + b < 0$, $\Delta\mathcal{C}_{\ell_\gamma, \mathcal{H}_g}(f, \mathbf{x}, \eta) = \eta - \min\{\eta, 1-\eta\} = \max\{0, 2\eta - 1\} = |2\eta - 1| \mathbb{1}_{(2\eta-1)[g(\mathbf{w} \cdot \mathbf{x} - \gamma) + b] \le 0}$ since $g(\mathbf{w} \cdot \mathbf{x} - \gamma) + b \le g(\mathbf{w} \cdot \mathbf{x} + \gamma) + b < 0$; for $f \in \mathcal{H}_g$ such that $g(\mathbf{w} \cdot \mathbf{x} - \gamma) + b > 0$, $\Delta\mathcal{C}_{\ell_\gamma, \mathcal{H}_g}(f, \mathbf{x}, \eta) = 1 - \eta - \min\{\eta, 1-\eta\} = \max\{0, 1-2\eta\} = |2\eta - 1| \mathbb{1}_{(2\eta-1)[g(\mathbf{w} \cdot \mathbf{x} - \gamma) + b] \le 0}$ since $g(\mathbf{w} \cdot \mathbf{x} - \gamma) + b > 0$. Therefore,

$$\Delta\mathcal{C}_{\ell_\gamma, \mathcal{H}_g}(f, \mathbf{x}, \eta) = \begin{cases} \max\{\eta, 1-\eta\} & \text{if } g(\mathbf{w} \cdot \mathbf{x} - \gamma) + b \le 0 \le g(\mathbf{w} \cdot \mathbf{x} + \gamma) + b, \\ |2\eta - 1| \mathbb{1}_{(2\eta-1)[g(\mathbf{w} \cdot \mathbf{x} - \gamma) + b] \le 0} & \text{if } g(\mathbf{w} \cdot \mathbf{x} + \gamma) + b < 0 \text{ or } g(\mathbf{w} \cdot \mathbf{x} - \gamma) + b > 0. \end{cases}$$

By (6), for a fixed $\eta \in [0, 1]$ and $\mathbf{x} \in \mathcal{X}$, the calibration function of losses $(\ell, \ell_\gamma)$ given $\mathcal{H}_g$ is

$$\delta_{\max}(\epsilon, \mathbf{x}, \eta) = \inf_{f \in \mathcal{H}_g} \left\{ \Delta\mathcal{C}_{\ell, \mathcal{H}_g}(f, \mathbf{x}, \eta) \mid \Delta\mathcal{C}_{\ell_\gamma, \mathcal{H}_g}(f, \mathbf{x}, \eta) \ge \epsilon \right\}.$$

As with the proof of Lemma 27, we then make use of the observation (11) for deriving the the calibration function. By the observation (11), if $\epsilon > \max\{\eta, 1-\eta\}$, then for all $f \in \mathcal{H}_g$, $\Delta\mathcal{C}_{\ell_\gamma, \mathcal{H}_g}(f, \mathbf{x}, \eta) < \epsilon$, which implies that $\delta_{\max}(\epsilon, \mathbf{x}, \eta) = \infty$; if $|2\eta - 1| < \epsilon \le \max\{\eta, 1-\eta\}$, then $\Delta\mathcal{C}_{\ell_\gamma, \mathcal{H}_g}(f, \mathbf{x}, \eta) \ge \epsilon$ if and only if $g(\mathbf{w} \cdot \mathbf{x} - \gamma) + b \le 0 \le g(\mathbf{w} \cdot \mathbf{x} + \gamma) + b$, which leads to

$$\delta_{\max}(\epsilon, \mathbf{x}, \eta) = \inf_{f \in \mathcal{H}_g: \, g(\mathbf{w} \cdot \mathbf{x} - \gamma) + b \le 0 \le g(\mathbf{w} \cdot \mathbf{x} + \gamma) + b} \Delta\mathcal{C}_{\ell, \mathcal{H}_g}(f, \mathbf{x}, \eta);$$

if $\epsilon \le |2\eta - 1|$, then $\Delta\mathcal{C}_{\ell_\gamma, \mathcal{H}_g}(f, \mathbf{x}, \eta) \ge \epsilon$ if and only if $g(\mathbf{w} \cdot \mathbf{x} - \gamma) + b \le 0 \le g(\mathbf{w} \cdot \mathbf{x} + \gamma) + b$ or $(2\eta - 1)[g(\mathbf{w} \cdot \mathbf{x} - \gamma) + b] \le 0$, which leads to

$$\delta_{\max}(\epsilon, \mathbf{x}, \eta) = \inf_{f \in \mathcal{H}_g: \, g(\mathbf{w} \cdot \mathbf{x} - \gamma) + b \le 0 \le g(\mathbf{w} \cdot \mathbf{x} + \gamma) + b \text{ or } (2\eta-1)[g(\mathbf{w} \cdot \mathbf{x} - \gamma) + b] \le 0} \Delta\mathcal{C}_{\ell, \mathcal{H}_g}(f, \mathbf{x}, \eta).$$

$\square$

**Lemma 30.** *Let $\ell$ be a surrogate loss function. Then $\ell$ is $\mathcal{H}_g$-calibrated with respect to $\ell_\gamma$ if and only if for any $\mathbf{x} \in \mathcal{X}$,*

$$\inf_{f \in \mathcal{H}_g:\, g(\mathbf{w}\cdot\mathbf{x}-\gamma)+b \leq 0 \leq g(\mathbf{w}\cdot\mathbf{x}+\gamma)+b} \mathcal{C}_\ell\left(f, \mathbf{x}, \frac{1}{2}\right) > \inf_{f \in \mathcal{H}_g} \mathcal{C}_\ell\left(f, \mathbf{x}, \frac{1}{2}\right), and$$

$$\inf_{f \in \mathcal{H}_g:\, g(\mathbf{w}\cdot\mathbf{x}-\gamma)+b \leq 0} \mathcal{C}_\ell(f, \mathbf{x}, \eta) > \inf_{f \in \mathcal{H}_g} \mathcal{C}_\ell(f, \mathbf{x}, \eta) \text{ for all } \eta \in \left(\frac{1}{2}, 1\right], and$$

$$\inf_{f \in \mathcal{H}_g:\, g(\mathbf{w}\cdot\mathbf{x}+\gamma)+b \geq 0} \mathcal{C}_\ell(f, \mathbf{x}, \eta) > \inf_{f \in \mathcal{H}_g} \mathcal{C}_\ell(f, \mathbf{x}, \eta) \text{ for all } \eta \in \left[0, \frac{1}{2}\right).$$

*Proof.* As the proof of Lemma 28 first makes use of Lemma 27 and Proposition 4, we also first make use of Lemma 29 and Proposition 4 in the following proof. Let $\delta_{\max}$ be the calibration function of $(\ell, \ell_\gamma)$ for hypothesis set $\mathcal{H}_g$. By Lemma 29,

$$\delta_{\max}(\epsilon, \mathbf{x}, \eta) = \begin{cases} +\infty & \text{if } \epsilon > \max\{\eta, 1-\eta\}, \\ \inf_{f \in \mathcal{H}_g:\, g(\mathbf{w}\cdot\mathbf{x}-\gamma)+b \leq 0 \leq g(\mathbf{w}\cdot\mathbf{x}+\gamma)+b} \Delta\mathcal{C}_{\ell,\mathcal{H}_g}(f, \mathbf{x}, \eta) & \text{if } |2\eta-1| < \epsilon \leq \max\{\eta, 1-\eta\}, \\ \inf_{f \in \mathcal{H}_g:\, g(\mathbf{w}\cdot\mathbf{x}-\gamma)+b \leq 0 \leq g(\mathbf{w}\cdot\mathbf{x}+\gamma)+b \text{ or } (2\eta-1)[g(\mathbf{w}\cdot\mathbf{x}-\gamma)+b] \leq 0} \Delta\mathcal{C}_{\ell,\mathcal{H}_g}(f, \mathbf{x}, \eta) & \text{if } \epsilon \leq |2\eta-1|. \end{cases}$$

By Proposition 4, $\ell$ is $\mathcal{H}_g$-calibrated with respect to $\ell_\gamma$ if and only if its calibration function $\delta_{\max}$ satisfies $\delta_{\max}(\epsilon, \mathbf{x}, \eta) > 0$ for all $\mathbf{x} \in \mathcal{X}$, $\eta \in [0,1]$ and $\epsilon > 0$. The following steps are similar to the steps in the proof of Lemma 28, where we analyze by considering three cases.
For $\eta = \frac{1}{2}$, we have for any $\mathbf{x} \in \mathcal{X}$,

$$\delta_{\max}\left(\epsilon, \mathbf{x}, \frac{1}{2}\right) > 0 \text{ for all } \epsilon > 0 \Leftrightarrow \inf_{f \in \mathcal{H}_g:\, g(\mathbf{w}\cdot\mathbf{x}-\gamma)+b \leq 0 \leq g(\mathbf{w}\cdot\mathbf{x}+\gamma)+b} \mathcal{C}_\ell\left(f, \mathbf{x}, \frac{1}{2}\right) > \inf_{f \in \mathcal{H}_g} \mathcal{C}_\ell\left(f, \mathbf{x}, \frac{1}{2}\right).$$

$$(32)$$

For $1 \geq \eta > \frac{1}{2}$, we have $|2\eta - 1| = 2\eta - 1$, $\max\{\eta, 1-\eta\} = \eta$, and

$$\inf_{f \in \mathcal{H}_g:\, g(\mathbf{w}\cdot\mathbf{x}-\gamma)+b \leq 0 \leq g(\mathbf{w}\cdot\mathbf{x}+\gamma)+b \text{ or } (2\eta-1)[g(\mathbf{w}\cdot\mathbf{x}-\gamma)+b] \leq 0} \Delta\mathcal{C}_{\ell,\mathcal{H}_g}(f, \mathbf{x}, \eta) = \inf_{f \in \mathcal{H}_g:\, g(\mathbf{w}\cdot\mathbf{x}-\gamma)+b \leq 0} \Delta\mathcal{C}_{\ell,\mathcal{H}_g}(f, \mathbf{x}, \eta).$$

Therefore, $\delta_{\max}(\epsilon, \mathbf{x}, \frac{1}{2}) > 0$ for any $\mathbf{x} \in \mathcal{X}$, $\epsilon > 0$ and $\eta \in (\frac{1}{2}, 1]$ if and only if for any $\mathbf{x} \in \mathcal{X}$,

$$\begin{cases} \inf_{f \in \mathcal{H}_g:\, g(\mathbf{w}\cdot\mathbf{x}-\gamma)+b \leq 0 \leq g(\mathbf{w}\cdot\mathbf{x}+\gamma)+b} \mathcal{C}_\ell(f, \mathbf{x}, \eta) > \inf_{f \in \mathcal{H}_g} \mathcal{C}_\ell(f, \mathbf{x}, \eta) & \text{for all } \eta \in (\frac{1}{2}, 1] \text{ such that } 2\eta - 1 < \epsilon \leq \eta, \\ \inf_{f \in \mathcal{H}_g:\, g(\mathbf{w}\cdot\mathbf{x}-\gamma)+b \leq 0} \mathcal{C}_\ell(f, \mathbf{x}, \eta) > \inf_{f \in \mathcal{H}_g} \mathcal{C}_\ell(f, \mathbf{x}, \eta) & \text{for all } \eta \in (\frac{1}{2}, 1] \text{ such that } \epsilon \leq 2\eta - 1, \end{cases}$$

for all $\epsilon > 0$, which is equivalent to for any $\mathbf{x} \in \mathcal{X}$,

$$\begin{cases} \inf_{f \in \mathcal{H}_g:\, g(\mathbf{w}\cdot\mathbf{x}-\gamma)+b \leq 0 \leq g(\mathbf{w}\cdot\mathbf{x}+\gamma)+b} \mathcal{C}_\ell(f, \mathbf{x}, \eta) > \inf_{f \in \mathcal{H}_g} \mathcal{C}_\ell(f, \mathbf{x}, \eta) & \text{for all } \eta \in (\frac{1}{2}, 1] \text{ such that } \epsilon \leq \eta < \frac{\epsilon+1}{2}, \\ \inf_{f \in \mathcal{H}_g:\, g(\mathbf{w}\cdot\mathbf{x}-\gamma)+b \leq 0} \mathcal{C}_\ell(f, \mathbf{x}, \eta) > \inf_{f \in \mathcal{H}_g} \mathcal{C}_\ell(f, \mathbf{x}, \eta) & \text{for all } \eta \in (\frac{1}{2}, 1] \text{ such that } \frac{\epsilon+1}{2} \leq \eta, \end{cases}$$

$$(33)$$

for all $\epsilon > 0$. Observe that

$$\left\{\eta \in \left(\frac{1}{2}, 1\right] \,\middle|\, \epsilon \leq \eta < \frac{\epsilon+1}{2}, \epsilon > 0\right\} = \left\{\frac{1}{2} < \eta \leq 1\right\}, \text{ and}$$

$$\left\{\eta \in \left(\frac{1}{2}, 1\right] \,\middle|\, \frac{\epsilon+1}{2} \leq \eta, \epsilon > 0\right\} = \left\{\frac{1}{2} < \eta \leq 1\right\}, \text{ and}$$

$$\inf_{f \in \mathcal{H}_g:\, g(\mathbf{w}\cdot\mathbf{x}-\gamma)+b \leq 0 \leq g(\mathbf{w}\cdot\mathbf{x}+\gamma)+b} \mathcal{C}_\ell(f, \mathbf{x}, \eta) \geq \inf_{f \in \mathcal{H}_g:\, g(\mathbf{w}\cdot\mathbf{x}-\gamma)+b \leq 0} \mathcal{C}_\ell(f, \mathbf{x}, \eta) \text{ for all } \eta.$$

Therefore, we reduce the above condition (33) as for any $\mathbf{x} \in \mathcal{X}$,

$$\inf_{f \in \mathcal{H}_g:\, g(\mathbf{w}\cdot\mathbf{x}-\gamma)+b \leq 0} \mathcal{C}_\ell(f, \mathbf{x}, \eta) > \inf_{f \in \mathcal{H}_g} \mathcal{C}_\ell(f, \mathbf{x}, \eta) \text{ for all } \eta \in \left(\frac{1}{2}, 1\right]. \qquad (34)$$

For $\frac{1}{2} > \eta \geq 0$, we have $|2\eta - 1| = 1 - 2\eta$, $\max\{\eta, 1-\eta\} = 1 - \eta$, and

$$\inf_{f \in \mathcal{H}_g:\, g(\mathbf{w}\cdot\mathbf{x}-\gamma)+b \leq 0 \leq g(\mathbf{w}\cdot\mathbf{x}+\gamma)+b \text{ or } (2\eta-1)[g(\mathbf{w}\cdot\mathbf{x}-\gamma)+b] \leq 0} \Delta\mathcal{C}_{\ell,\mathcal{H}_g}(f, \mathbf{x}, \eta) = \inf_{f \in \mathcal{H}_g:\, g(\mathbf{w}\cdot\mathbf{x}+\gamma)+b \geq 0} \Delta\mathcal{C}_{\ell,\mathcal{H}_g}(f, \mathbf{x}, \eta).$$

Therefore, $\delta_{\max}(\epsilon, \mathbf{x}, \frac{1}{2}) > 0$ for any $\mathbf{x} \in \mathcal{X}$, $\epsilon > 0$ and $\eta \in [0, \frac{1}{2})$ if and only if for any $\mathbf{x} \in \mathcal{X}$,

$$\begin{cases} \displaystyle\inf_{f \in \mathcal{H}_g\colon g(\mathbf{w}\cdot\mathbf{x}-\gamma)+b \leq 0 \leq g(\mathbf{w}\cdot\mathbf{x}+\gamma)+b} \mathcal{C}_\ell(f,\mathbf{x},\eta) > \inf_{f \in \mathcal{H}_g} \mathcal{C}_\ell(f,\mathbf{x},\eta) & \text{for all } \eta \in [0,\tfrac{1}{2}) \text{ such that } 1-2\eta < \epsilon \leq 1-\eta, \\ \displaystyle\inf_{f \in \mathcal{H}_g\colon g(\mathbf{w}\cdot\mathbf{x}+\gamma)+b \geq 0} \mathcal{C}_\ell(f,\mathbf{x},\eta) > \inf_{f \in \mathcal{H}_g} \mathcal{C}_\ell(f,\mathbf{x},\eta) & \text{for all } \eta \in [0,\tfrac{1}{2}) \text{ such that } \epsilon \leq 1-2\eta, \end{cases}$$

for all $\epsilon > 0$, which is equivalent to for any $\mathbf{x} \in \mathcal{X}$,

$$\begin{cases} \displaystyle\inf_{f \in \mathcal{H}_g\colon g(\mathbf{w}\cdot\mathbf{x}-\gamma)+b \leq 0 \leq g(\mathbf{w}\cdot\mathbf{x}+\gamma)+b} \mathcal{C}_\ell(f,\mathbf{x},\eta) > \inf_{f \in \mathcal{H}_g} \mathcal{C}_\ell(f,\mathbf{x},\eta) & \text{for all } \eta \in [0,\tfrac{1}{2}) \text{ such that } \tfrac{1-\epsilon}{2} < \eta \leq 1-\epsilon, \\ \displaystyle\inf_{f \in \mathcal{H}_g\colon g(\mathbf{w}\cdot\mathbf{x}+\gamma)+b \geq 0} \mathcal{C}_\ell(f,\mathbf{x},\eta) > \inf_{f \in \mathcal{H}_g} \mathcal{C}_\ell(f,\mathbf{x},\eta) & \text{for all } \eta \in [0,\tfrac{1}{2}) \text{ such that } \eta \leq \tfrac{1-\epsilon}{2}, \end{cases}$$

(35)

for all $\epsilon > 0$. Observe that

$$\left\{ \eta \in [0,\tfrac{1}{2}) \,\middle|\, \tfrac{1-\epsilon}{2} < \eta \leq 1-\epsilon, \epsilon > 0 \right\} = \left\{ 0 \leq \eta < \tfrac{1}{2} \right\}, \text{ and}$$

$$\left\{ \eta \in [0,\tfrac{1}{2}) \,\middle|\, \eta \leq \tfrac{1-\epsilon}{2}, \epsilon > 0 \right\} = \left\{ 0 \leq \eta < \tfrac{1}{2} \right\}, \text{ and}$$

$$\inf_{f \in \mathcal{H}_g\colon g(\mathbf{w}\cdot\mathbf{x}-\gamma)+b \leq 0 \leq g(\mathbf{w}\cdot\mathbf{x}+\gamma)+b} \mathcal{C}_\ell(f,\mathbf{x},\eta) \geq \inf_{f \in \mathcal{H}_g\colon g(\mathbf{w}\cdot\mathbf{x}+\gamma)+b \geq 0} \mathcal{C}_\ell(f,\mathbf{x},\eta) \text{ for all } \eta.$$

Therefore we reduce the above condition (35) as for any $\mathbf{x} \in \mathcal{X}$,

$$\inf_{f \in \mathcal{H}_g\colon g(\mathbf{w}\cdot\mathbf{x}+\gamma)+b \geq 0} \mathcal{C}_\ell(f,\mathbf{x},\eta) > \inf_{f \in \mathcal{H}_g} \mathcal{C}_\ell(f,\mathbf{x},\eta) \text{ for all } \eta \in [0, \tfrac{1}{2}). \tag{36}$$

To sum up, by (32), (34) and (36), we conclude the proof. $\qquad\square$

**Theorem 13.** *Let $g$ be a non-decreasing and continuous function such that $g(1+\gamma) < G$ and $g(-1-\gamma) > -G$ for some $G \geq 0$. Let a margin-based loss $\phi$ be bounded, continuous, non-increasing, and satisfy the property that $\bar{C}_\phi(t,\eta)$ is quasi-concave in $t \in \mathbb{R}$ for all $\eta \in [0,1]$. Assume that $\phi(g(-t) - G) > \phi(G - g(-t))$ and $g(-t) + g(t) \geq 0$ for any $0 \leq t \leq 1$. Then $\phi$ is $\mathcal{H}_g$-calibrated with respect to $\ell_\gamma$ if and only if for any $0 \leq t \leq 1$,*

$$\phi(G - g(-t)) + \phi(g(-t) - G) = \phi(g(t) + G) + \phi(-g(t) - G)$$

*and* $\quad \min\{\phi(\overline{A}(t)) + \phi(-\overline{A}(t)), \phi(\underline{A}(t)) + \phi(-\underline{A}(t))\} > \phi(G - g(-t)) + \phi(g(-t) - G),$

*where $\overline{A}(t) = \max_{s \in [-t,t]} g(s) - g(s-\gamma)$ and $\underline{A}(t) = \min_{s \in [-t,t]} g(s) - g(s+\gamma)$.*

*Proof.* By Lemma 30, $\phi$ is $\mathcal{H}_g$-calibrated with respect to $\ell_\gamma$ if and only if for any $\mathbf{x} \in \mathcal{X}$,

$$\inf_{f \in \mathcal{H}_g\colon g(\mathbf{w}\cdot\mathbf{x}-\gamma)+b \leq 0 \leq g(\mathbf{w}\cdot\mathbf{x}+\gamma)+b} \mathcal{C}_\phi(f,\mathbf{x},\tfrac{1}{2}) > \inf_{f \in \mathcal{H}_g} \mathcal{C}_\phi(f,\mathbf{x},\tfrac{1}{2}), \text{ and}$$

$$\inf_{f \in \mathcal{H}_g\colon g(\mathbf{w}\cdot\mathbf{x}-\gamma)+b \leq 0} \mathcal{C}_\phi(f,\mathbf{x},\eta) > \inf_{f \in \mathcal{H}_g} \mathcal{C}_\phi(f,\mathbf{x},\eta) \text{ for all } \eta \in (\tfrac{1}{2}, 1], \text{ and} \tag{37}$$

$$\inf_{f \in \mathcal{H}_g\colon g(\mathbf{w}\cdot\mathbf{x}+\gamma)+b \geq 0} \mathcal{C}_\phi(f,\mathbf{x},\eta) > \inf_{f \in \mathcal{H}_g} \mathcal{C}_\phi(f,\mathbf{x},\eta) \text{ for all } \eta \in [0, \tfrac{1}{2}).$$

By the definition of inner risk (4), the inner $\phi$-risk is

$$\mathcal{C}_\phi(f,\mathbf{x},\eta) = \eta\phi(f(\mathbf{x})) + (1-\eta)\phi(-f(\mathbf{x})).$$

and $f(\mathbf{x}) = g(\mathbf{w} \cdot \mathbf{x}) + b \in [g(-\|\mathbf{x}\|) - G, g(\|\mathbf{x}\|) + G]$ when $f \in \mathcal{H}_g$ since $g$ is continuous and non-decreasing. Specifically, by the assumption that $g(-1-\gamma) + G > 0$, $g(1+\gamma) - G < 0$, when $f \in \{f \in \mathcal{H}_g : g(\mathbf{w}\cdot\mathbf{x}-\gamma)+b \leq 0 \leq g(\mathbf{w}\cdot\mathbf{x}+\gamma)+b\}$, $f(\mathbf{x}) = g(\mathbf{w}\cdot\mathbf{x})+b \in [\min_{-\|\mathbf{x}\| \leq s \leq \|\mathbf{x}\|} g(s) - g(s+\gamma), \max_{-\|\mathbf{x}\| \leq s \leq \|\mathbf{x}\|} g(s) - g(s-\gamma)]$; when $f \in \{f \in \mathcal{H}_g : g(\mathbf{w}\cdot\mathbf{x}-\gamma)+b \leq 0\}$, $f(\mathbf{x}) = g(\mathbf{w}\cdot\mathbf{x})+b \in [g(-\|\mathbf{x}\|) - G, \max_{-\|\mathbf{x}\| \leq s \leq \|\mathbf{x}\|} g(s) - g(s-\gamma)]$; when $f \in \{f \in \mathcal{H}_g : g(\mathbf{w} \cdot \mathbf{x} + \gamma) + b \geq 0\}$, $f(\mathbf{x}) = g(\mathbf{w} \cdot \mathbf{x}) + b \in [\min_{-\|\mathbf{x}\| \leq s \leq \|\mathbf{x}\|} g(s) - g(s+\gamma), g(\|\mathbf{x}\|) + G]$. For convenience, we denote

$\overline{A}(t) = \max_{-t \le s \le t} g(s) - g(s - \gamma) \ge 0$ and $\underline{A}(t) = \min_{-t \le s \le t} g(s) - g(s + \gamma) \le 0$ for any $0 \le t \le 1$. Therefore, for any $\mathbf{x} \in \mathcal{X}$, (37) is equivalent to

$$\inf_{\underline{A}(\|\mathbf{x}\|) \le t \le \overline{A}(\|\mathbf{x}\|)} \bar{\mathcal{C}}_\phi(t, \frac{1}{2}) > \inf_{g(-\|\mathbf{x}\|) - G \le t \le g(\|\mathbf{x}\|) + G} \bar{\mathcal{C}}_\phi(t, \frac{1}{2}), \text{ and}$$

$$\inf_{g(-\|\mathbf{x}\|) - G \le t \le \overline{A}(\|\mathbf{x}\|)} \bar{\mathcal{C}}_\phi(t, \eta) > \inf_{g(-\|\mathbf{x}\|) - G \le t \le g(\|\mathbf{x}\|) + G} \bar{\mathcal{C}}_\phi(t, \eta) \text{ for all } \eta \in (\frac{1}{2}, 1], \text{ and} \qquad (38)$$

$$\inf_{\underline{A}(\|\mathbf{x}\|) \le t \le g(\|\mathbf{x}\|) + G} \bar{\mathcal{C}}_\phi(t, \eta) > \inf_{g(-\|\mathbf{x}\|) - G \le t \le g(\|\mathbf{x}\|) + G} \bar{\mathcal{C}}_\phi(t, \eta) \text{ for all } \eta \in [0, \frac{1}{2}).$$

Suppose that $\phi$ is $\mathcal{H}_g$-calibrated with respect to $\ell_\gamma$. Since for $\eta \in [0, \frac{1}{2})$,

$$\inf_{\underline{A}(\|\mathbf{x}\|) \le t \le g(\|\mathbf{x}\|) + G} \bar{\mathcal{C}}_\phi(t, \eta) = \min\{\bar{\mathcal{C}}_\phi(\underline{A}(\|\mathbf{x}\|), \eta), \bar{\mathcal{C}}_\phi(g(\|\mathbf{x}\|) + G, \eta)\} \qquad \text{(Part 2 of Lemma 26)}$$

$$\inf_{g(-\|\mathbf{x}\|) - G \le t \le g(\|\mathbf{x}\|) + G} \bar{\mathcal{C}}_\phi(t, \eta) = \min\{\bar{\mathcal{C}}_\phi(g(-\|\mathbf{x}\|) - G, \eta), \bar{\mathcal{C}}_\phi(g(\|\mathbf{x}\|) + G, \eta)\} \quad \text{(Part 2 of Lemma 26)}$$

we have $\bar{\mathcal{C}}_\phi(g(-\|\mathbf{x}\|) - G, \eta) < \bar{\mathcal{C}}_\phi(g(\|\mathbf{x}\|) + G, \eta)$ for any $\mathbf{x} \in \mathcal{X}$, otherwise

$$\inf_{\underline{A}(\|\mathbf{x}\|) \le t \le g(\|\mathbf{x}\|) + G} \bar{\mathcal{C}}_\phi(t, \eta) \le \bar{\mathcal{C}}_\phi(g(\|\mathbf{x}\|) + G, \eta) = \inf_{g(-\|\mathbf{x}\|) - G \le t \le g(\|\mathbf{x}\|) + G} \bar{\mathcal{C}}_\phi(t, \eta).$$

By Part 8 of Lemma 26, $\phi(G - g(-t)) + \phi(g(-t) - G) = \phi(g(t) + G) + \phi(-g(t) - G)$ for all $0 \le t \le 1$. Also, for any $0 \le t \le 1$,

$$\frac{1}{2} \min\{\phi(\overline{A}(t)) + \phi(-\overline{A}(t)), \phi(\underline{A}(t)) + \phi(-\underline{A}(t))\}$$

$$= \inf_{\underline{A}(t) \le t \le \overline{A}(t)} \bar{\mathcal{C}}_\phi(t, \frac{1}{2}) \qquad \text{(Part 2 of Lemma 26)}$$

$$> \inf_{g(-t) - G \le t \le g(t) + G} \bar{\mathcal{C}}_\phi(t, \frac{1}{2}) \qquad (38)$$

$$= \frac{1}{2} \min\{\phi(G - g(-t)) + \phi(g(-t) - G), \phi(g(t) + G) + \phi(-g(t) - G)\} \quad \text{(Part 2 of Lemma 26)}$$

$$= \frac{1}{2}(\phi(G - g(-t)) + \phi(g(-t) - G))$$

Now for the other direction, assume that for any $0 \le t \le 1$,

$$\phi(G - g(-t)) + \phi(g(-t) - G) = \phi(g(t) + G) + \phi(-g(t) - G)$$

and $\quad \min\{\phi(\overline{A}(t)) + \phi(-\overline{A}(t)), \phi(\underline{A}(t)) + \phi(-\underline{A}(t))\} > \phi(G - g(-t)) + \phi(g(-t) - G).$

Then for $\eta = \frac{1}{2}$ and any $\mathbf{x} \in \mathcal{X}$,

$$\inf_{\underline{A}(\|\mathbf{x}\|) \le t \le \overline{A}(\|\mathbf{x}\|)} \bar{\mathcal{C}}_\phi(t, \frac{1}{2})$$

$$= \frac{1}{2} \min\{\phi(\overline{A}(\|\mathbf{x}\|)) + \phi(-\overline{A}(\|\mathbf{x}\|)), \phi(\underline{A}(\|\mathbf{x}\|)) + \phi(-\underline{A}(\|\mathbf{x}\|))\} \qquad \text{(Part 2 of Lemma 26)}$$

$$> \frac{1}{2}(\phi(G - g(-\|\mathbf{x}\|)) + \phi(g(-\|\mathbf{x}\|) - G)) \qquad \text{(by assumption)}$$

$$= \frac{1}{2} \min\{\phi(G - g(-\|\mathbf{x}\|)) + \phi(g(-\|\mathbf{x}\|) - G), \phi(g(\|\mathbf{x}\|) + G) + \phi(-g(\|\mathbf{x}\|) - G)\} \qquad \text{(by assumption)}$$

$$= \inf_{g(-\|\mathbf{x}\|) - G \le t \le g(\|\mathbf{x}\|) + G} \bar{\mathcal{C}}_\phi(t, \frac{1}{2}). \qquad \text{(Part 2 of Lemma 26)}$$

For $\eta \in (\frac{1}{2}, 1]$ and any $\mathbf{x} \in \mathcal{X}$,

$$\inf_{g(-\|\mathbf{x}\|) - G \le t \le \overline{A}(\|\mathbf{x}\|)} \bar{\mathcal{C}}_\phi(t, \eta) = \min\{\bar{\mathcal{C}}_\phi(g(-\|\mathbf{x}\|) - G, \eta), \bar{\mathcal{C}}_\phi(\overline{A}(\|\mathbf{x}\|), \eta)\} \qquad \text{(Part 2 of Lemma 26)}$$

$$\inf_{g(-\|\mathbf{x}\|) - G \le t \le g(\|\mathbf{x}\|) + G} \bar{\mathcal{C}}_\phi(t, \eta) = \min\{\bar{\mathcal{C}}_\phi(g(-\|\mathbf{x}\|) - G, \eta), \bar{\mathcal{C}}_\phi(g(\|\mathbf{x}\|) + G, \eta)\} \qquad \text{(Part 2 of Lemma 26)}$$

$$= \bar{\mathcal{C}}_\phi(g(\|\mathbf{x}\|) + G, \eta) \qquad \text{(Part 7 of Lemma 26)}$$

Since $\phi$ is non-increasing, we have for any $\mathbf{x} \in \mathcal{X}$,

$$\phi(-g(\|\mathbf{x}\|) - G) - \phi(g(\|\mathbf{x}\|) + G) + \phi(\overline{A}(\|\mathbf{x}\|)) - \phi(-\overline{A}(\|\mathbf{x}\|))$$
$$\geq \phi(-g(\|\mathbf{x}\|) - G) - \phi(g(\|\mathbf{x}\|) + G) + \phi(g(\|\mathbf{x}\|) + G) - \phi(-g(\|\mathbf{x}\|) - G)$$
$$= 0.$$

Then for $\eta \in (\frac{1}{2}, 1]$ and any $\mathbf{x} \in \mathcal{X}$,

$$\bar{\mathcal{C}}_\phi(\overline{A}(\|\mathbf{x}\|), \eta) - \bar{\mathcal{C}}_\phi(g(\|\mathbf{x}\|) + G, \eta)$$
$$= (\phi(\overline{A}(\|\mathbf{x}\|)) - \phi(-\overline{A}(\|\mathbf{x}\|)) + \phi(-g(\|\mathbf{x}\|) - G) - \phi(g(\|\mathbf{x}\|) + G))\eta + \phi(-\overline{A}(\|\mathbf{x}\|)) - \phi(-g(\|\mathbf{x}\|) - G)$$
$$\geq (\phi(\overline{A}(\|\mathbf{x}\|)) - \phi(-\overline{A}(\|\mathbf{x}\|)) + \phi(-g(\|\mathbf{x}\|) - G) - \phi(g(\|\mathbf{x}\|) + G))\frac{1}{2} + \phi(-\overline{A}(\|\mathbf{x}\|)) - \phi(-g(\|\mathbf{x}\|) - G)$$
$$= \frac{1}{2}(\phi(\overline{A}(\|\mathbf{x}\|)) - \phi(-\overline{A}(\|\mathbf{x}\|)) - \phi(-g(\|\mathbf{x}\|) - G) - \phi(g(\|\mathbf{x}\|) + G))$$
$$> 0.$$

In addition, by Part 7 of Lemma 26, for all $\eta \in (\frac{1}{2}, 1]$ and any $\mathbf{x} \in \mathcal{X}$, $\bar{\mathcal{C}}_\phi(g(-\|\mathbf{x}\|) - G, \eta) - \bar{\mathcal{C}}_\phi(g(\|\mathbf{x}\|) + G, \eta) > 0$. As a result, for $\eta \in (\frac{1}{2}, 1]$ and any $\mathbf{x} \in \mathcal{X}$,

$$\inf_{g(-\|\mathbf{x}\|) - G \leq t \leq \overline{A}(\|\mathbf{x}\|)} \bar{\mathcal{C}}_\phi(t, \eta) - \inf_{g(-\|\mathbf{x}\|) - G \leq t \leq g(\|\mathbf{x}\|) + G} \bar{\mathcal{C}}_\phi(t, \eta)$$
$$= \min\{\bar{\mathcal{C}}_\phi(g(-\|\mathbf{x}\|) - G, \eta) - \bar{\mathcal{C}}_\phi(g(\|\mathbf{x}\|) + G, \eta), \bar{\mathcal{C}}_\phi(\overline{A}(\|\mathbf{x}\|), \eta) - \bar{\mathcal{C}}_\phi(g(\|\mathbf{x}\|) + G, \eta)\}$$
$$> 0.$$

Finally, for $\eta \in [0, \frac{1}{2})$, by Part 8 of Lemma 26, we have $\bar{\mathcal{C}}_\phi(g(-\|\mathbf{x}\|) - G, \eta) < \bar{\mathcal{C}}_\phi(g(\|\mathbf{x}\|) + G, \eta)$ and

$$\inf_{\underline{A}(\|\mathbf{x}\|) \leq t \leq g(\|\mathbf{x}\|) + G} \bar{\mathcal{C}}_\phi(t, \eta) = \min\{\bar{\mathcal{C}}_\phi(\underline{A}(\|\mathbf{x}\|), \eta), \bar{\mathcal{C}}_\phi(g(\|\mathbf{x}\|) + G, \eta)\} \qquad \text{(Part 2 of Lemma 26)}$$

$$\inf_{g(-\|\mathbf{x}\|) - G \leq t \leq g(\|\mathbf{x}\|) + G} \bar{\mathcal{C}}_\phi(t, \eta) = \min\{\bar{\mathcal{C}}_\phi(g(-\|\mathbf{x}\|) - G, \eta), \bar{\mathcal{C}}_\phi(g(\|\mathbf{x}\|) + G, \eta)\} \quad \text{(Part 2 of Lemma 26)}$$
$$= \bar{\mathcal{C}}_\phi(g(-\|\mathbf{x}\|) - G, \eta) \qquad \text{(Part 8 of Lemma 26)}$$

Since $\phi(\underline{A}(\|\mathbf{x}\|)) + \phi(-\underline{A}(\|\mathbf{x}\|)) > \phi(G - g(-\|\mathbf{x}\|)) + \phi(g(-\|\mathbf{x}\|) - G)$ and $\phi$ is non-increasing, we have for any $\mathbf{x} \in \mathcal{X}$,

$$\phi(G - g(-\|\mathbf{x}\|)) - \phi(g(-\|\mathbf{x}\|) - G) + \phi(\underline{A}(\|\mathbf{x}\|)) - \phi(-\underline{A}(\|\mathbf{x}\|))$$
$$= \phi(G - g(-\|\mathbf{x}\|)) - \phi(-\underline{A}(\|\mathbf{x}\|)) + \phi(\underline{A}(\|\mathbf{x}\|)) - \phi(g(-\|\mathbf{x}\|) - G)$$
$$< \phi(\underline{A}(\|\mathbf{x}\|)) - \phi(g(-\|\mathbf{x}\|) - G) + \phi(\underline{A}(\|\mathbf{x}\|)) - \phi(g(-\|\mathbf{x}\|) - G)$$
$$= 2[\phi(\underline{A}(\|\mathbf{x}\|)) - \phi(g(-\|\mathbf{x}\|) - G)]$$
$$\leq 0.$$

Then for $\eta \in [0, \frac{1}{2})$ and any $\mathbf{x} \in \mathcal{X}$.

$$\bar{\mathcal{C}}_\phi(\underline{A}(\|\mathbf{x}\|), \eta) - \bar{\mathcal{C}}_\phi(g(-\|\mathbf{x}\|) - G, \eta)$$
$$= [\phi(G - g(-\|\mathbf{x}\|)) - \phi(g(-\|\mathbf{x}\|) - G) + \phi(\underline{A}(\|\mathbf{x}\|)) - \phi(-\underline{A}(\|\mathbf{x}\|))]\eta + \phi(-\underline{A}(\|\mathbf{x}\|)) - \phi(G - g(-\|\mathbf{x}\|))$$
$$\geq [\phi(G - g(-\|\mathbf{x}\|)) - \phi(g(-\|\mathbf{x}\|) - G) + \phi(\underline{A}(\|\mathbf{x}\|)) - \phi(-\underline{A}(\|\mathbf{x}\|))]\frac{1}{2} + \phi(-\underline{A}(\|\mathbf{x}\|)) - \phi(G - g(-\|\mathbf{x}\|))$$
$$= \frac{1}{2}[\phi(\underline{A}(\|\mathbf{x}\|)) + \phi(-\underline{A}(\|\mathbf{x}\|)) - \phi(g(-\|\mathbf{x}\|) - G) - \phi(G - g(-\|\mathbf{x}\|))]$$
$$> 0.$$

In addition, by Part 8 of Lemma 26, for all $\eta \in [0, \frac{1}{2})$ and any $\mathbf{x} \in \mathcal{X}$, $\bar{\mathcal{C}}_\phi(g(\|\mathbf{x}\|) + G, \eta) - \bar{\mathcal{C}}_\phi(g(-\|\mathbf{x}\|) - G, \eta) > 0$. As a result, for $\eta \in [0, \frac{1}{2})$ and any $\mathbf{x} \in \mathcal{X}$,

$$\inf_{\underline{A}(\|\mathbf{x}\|) \leq t \leq g(\|\mathbf{x}\|) + G} \bar{\mathcal{C}}_\phi(t, \eta) - \inf_{g(-\|\mathbf{x}\|) - G \leq t \leq g(\|\mathbf{x}\|) + G} \bar{\mathcal{C}}_\phi(t, \eta)$$
$$= \min\{\bar{\mathcal{C}}_\phi(g(\|\mathbf{x}\|) + G, \eta) - \bar{\mathcal{C}}_\phi(g(-\|\mathbf{x}\|) - G, \eta), \bar{\mathcal{C}}_\phi(\underline{A}(\|\mathbf{x}\|), \eta) - \bar{\mathcal{C}}_\phi(g(-\|\mathbf{x}\|) - G, \eta)\}$$
$$> 0.$$

$\square$

**Corollary 14.** *Assume that $G > 1 + \gamma$. Let a margin-based loss $\phi$ be bounded, continuous, non-increasing, and satisfy the property that $\bar{C}_\phi(t, \eta)$ is quasi-concave in $t \in \mathbb{R}$ for all $\eta \in [0, 1]$. Assume that $\phi(-G) > \phi(G)$. Then $\phi$ is $\mathcal{H}_{\mathrm{relu}}$-calibrated with respect to $\ell_\gamma$ if and only if for any $0 \le t \le 1$,*

$$\phi(G) + \phi(-G) = \phi(t + G) + \phi(-t - G) \quad \text{and} \quad \phi(\gamma) + \phi(-\gamma) > \phi(G) + \phi(-G).$$

*Proof.* For hypothesis set $\mathcal{H}_{\mathrm{relu}}$, that is, $g = (\cdot)_+$ in $\mathcal{H}_g$, we have

$$\overline{A}(t) = \max_{s \in [-t, t]} (s)_+ - (s - \gamma)_+ = \begin{cases} t, 0 \le t < \gamma, \\ \gamma, \gamma \le t \le 1. \end{cases} \quad \text{and} \quad \underline{A}(t) = \min_{s \in [-t, t]} (s)_+ - (s + \gamma)_+ = -\gamma.$$

As a result, using the fact that $\phi(t) + \phi(-t) \ge \phi(\gamma) + \phi(-\gamma)$ when $0 \le t \le \gamma$ by Part 1 of Lemma 26, we conclude the proof by Theorem 13. $\qquad\square$

### E.6 Proof of Theorem 18

**Theorem 18.** *No continuous margin-based loss function $\phi$ is $\mathcal{H}_{\mathrm{lin}}$-consistent with respect to $\ell_\gamma$. Furthermore, for any continuous and non-increasing margin-based loss $\phi$, surrogates of the form*

$$\tilde{\phi}(f, \mathbf{x}, y) = \sup_{\mathbf{x}':\|\mathbf{x} - \mathbf{x}'\| \le \gamma} \phi(yf(\mathbf{x}'))$$

*are not $\mathcal{H}_{\mathrm{lin}}$-consistent with respect to $\ell_\gamma$.*

*Proof.* Let $\mathbf{x}$ follow the uniform distribution on the unit circle. Denote $\mathbf{x} = (\cos(\theta), \sin(\theta))^\top, \theta \in [0, 2\pi)$ and $f(\mathbf{x}) = \mathbf{w} \cdot \mathbf{x}, \mathbf{w} = (\cos(t), \sin(t))^\top, t \in [0, 2\pi), f \in \mathcal{H}_{\mathrm{lin}} = \{\mathbf{x} \to \mathbf{w} \cdot \mathbf{x} \mid \|\mathbf{w}\|_2 = 1\}$. We set the label of a point $\mathbf{x}$ as follows: if $\theta \in (\sigma, \pi)$, where $\sigma \in (0, \pi)$, then set $y = -1$ with probability $\frac{3}{4}$ and $y = 1$ with probability $\frac{1}{4}$; if $\theta \in (0, \sigma)$ or $(\sigma + \pi, 2\pi)$, then set $y = 1$; if $\theta \in (\pi, \sigma + \pi)$, then set $y = -1$.

Let $\eta: \mathcal{X} \to [0, 1]$ be a measurable function such that $\eta(X) = \mathbb{P}(Y = 1 \mid X)$. For $\ell_\gamma(\tau) = \mathbb{1}_{\tau \le \gamma}$, we want to solve

$$\mathcal{R}^*_{\ell_\gamma, \mathcal{H}_{\mathrm{lin}}} = \min_{f \in \mathcal{H}_{\mathrm{lin}}} \mathcal{R}_{\ell_\gamma}(f) = \min_{f \in \mathcal{H}_{\mathrm{lin}}} \mathbb{E}_X [\ell_\gamma(f(X))\eta + \ell_\gamma(-f(X))(1 - \eta)].$$

Let $\eta': \Theta \to [0, 1]$ be a measurable function such that $\eta' = \mathbb{P}(Y = 1|\Theta)$, $\Theta \sim \mathcal{U}(0, 2\pi)$. In our example, we have

$$\eta' = \begin{cases} \frac{1}{4} & \theta \in (\sigma, \pi), \\ 1 & \theta \in (0, \sigma) \text{ or } \theta \in (\sigma + \pi, 2\pi), \\ 0 & \theta \in (\pi, \sigma + \pi). \end{cases}$$

Therefore we obtain

$$\mathcal{R}^*_{\ell_\gamma, \mathcal{H}_{\mathrm{lin}}} = \min_{t \in [0, 2\pi)} \mathbb{E}_\Theta [\ell_\gamma(\cos(\Theta - t))\eta' + \ell_\gamma(-\cos(\Theta - t))(1 - \eta')]$$

$$= \frac{1}{2\pi} \min_{t \in [0, 2\pi)} \int_\sigma^\pi \frac{1}{4} \ell_\gamma(\cos(\theta - t)) + \frac{3}{4} \ell_\gamma(-\cos(\theta - t)) \, d\theta + \int_{\sigma-\pi}^\sigma \ell_\gamma(\cos(\theta - t)) \, d\theta$$

$$+ \int_{-\pi}^{\sigma-\pi} \ell_\gamma(-\cos(\theta - t)) \, d\theta$$

$$= \frac{1}{2\pi} \min_{t \in [0, 2\pi)} \int_\sigma^\pi \frac{1}{4} \ell_\gamma(\cos(\theta - t)) + \frac{3}{4} \ell_\gamma(-\cos(\theta - t)) d\theta + \int_{\sigma-\pi}^0 \ell_\gamma(\cos(\theta - t)) d\theta$$

$$+ \int_0^\sigma \ell_\gamma(\cos(\theta - t)) d\theta + \int_0^\sigma \ell_\gamma(\cos(\theta - t)) d\theta$$

$$= \frac{1}{2\pi} \min_{t \in [0, 2\pi)} \int_\sigma^\pi \frac{1}{4} \ell_\gamma(\cos(\theta - t)) \, d\theta + \int_{\sigma-\pi}^0 \frac{3}{4} \ell_\gamma(\cos(\theta - t)) \, d\theta + \int_{\sigma-\pi}^0 \ell_\gamma(\cos(\theta - t)) \, d\theta$$

$$+ \int_0^\sigma 2\ell_\gamma(\cos(\theta - t)) \, d\theta$$

$$= \frac{1}{2\pi} \min_{t \in [0, 2\pi)} \int_\sigma^\pi \frac{1}{4} \ell_\gamma(\cos(\theta - t)) \, d\theta + \int_{\sigma-\pi}^0 \frac{7}{4} \ell_\gamma(\cos(\theta - t)) \, d\theta + \int_0^\sigma \frac{7}{4} \ell_\gamma(\cos(\theta - t)) \, d\theta$$

$$+ \int_0^\sigma \frac{1}{4} \ell_\gamma(\cos(\theta - t))\, d\theta$$

$$= \frac{1}{2\pi} \min_{t \in [0, 2\pi]} \int_0^\pi \frac{1}{4} \ell_\gamma(\cos(\theta - t))\, d\theta + \int_{\sigma - \pi}^\sigma \frac{7}{4} \ell_\gamma(\cos(\theta - t))\, d\theta \qquad (39)$$

$$= \frac{1}{2\pi} \min_{t \in [0, 2\pi]} \int_0^\pi \frac{1}{4} \ell_\gamma(\cos(\theta - t))\, d\theta + \int_0^\pi \frac{7}{4} \ell_\gamma(-\cos(\theta - t + \sigma))\, d\theta$$

$$= \frac{1}{2\pi} \min_{t \in [0, 2\pi]} \int_{-t}^{\pi - t} \frac{1}{4} \ell_\gamma(\cos(\theta))\, d\theta + \frac{7}{4} \ell_\gamma(-\cos(\theta + \sigma))\, d\theta$$

$$= \frac{1}{2\pi} \min_{t \in [0, 2\pi]} \int_{-t}^{\pi - t} \frac{1}{4} \mathbb{1}_{\cos(\theta) \leq \gamma} + \frac{7}{4} \mathbb{1}_{-\cos(\theta + \sigma) \leq \gamma}\, d\theta\,.$$

Take $\gamma = \cos(\frac{\sigma}{2}) \in (0, 1)$. For $\sigma \in (0, \frac{\pi}{2}]$, we analyze six cases:

- When $-t \in \left[ -\frac{3\sigma}{2}, -\frac{\sigma}{2} \right]$,

$$\int_{-t}^{\pi - t} \frac{1}{4} \mathbb{1}_{\cos(\theta) \leq \gamma} + \frac{7}{4} \mathbb{1}_{-\cos(\theta + \sigma) \leq \gamma}\, d\theta$$

$$= \int_{-t}^{-\frac{\sigma}{2}} \frac{1}{4} + \frac{7}{4}\, d\theta + \int_{-\frac{\sigma}{2}}^{\frac{\sigma}{2}} \frac{7}{4}\, d\theta + \int_{\frac{\sigma}{2}}^{-\frac{3\sigma}{2} + \pi} \frac{1}{4} + \frac{7}{4}\, d\theta + \int_{-\frac{3\sigma}{2} + \pi}^{\pi - t} \frac{1}{4}\, d\theta$$

$$= 2\pi - \frac{23}{8}\sigma + \frac{7}{4}t \geq 2\pi - 2\sigma$$

where the equality is achieved when $t = \frac{\sigma}{2}$.

- When $-t \in \left[ -\frac{\sigma}{2}, \frac{\sigma}{2} \right]$,

$$\int_{-t}^{\pi - t} \frac{1}{4} \mathbb{1}_{\cos(\theta) \leq \gamma} + \frac{7}{4} \mathbb{1}_{-\cos(\theta + \sigma) \leq \gamma}\, d\theta$$

$$= \int_{-t}^{\frac{\sigma}{2}} \frac{7}{4}\, d\theta + \int_{\frac{\sigma}{2}}^{-\frac{3\sigma}{2} + \pi} \frac{1}{4} + \frac{7}{4}\, d\theta + \int_{-\frac{3\sigma}{2} + \pi}^{-\frac{\sigma}{2} + \pi} \frac{1}{4}\, d\theta + \int_{-\frac{\sigma}{2} + \pi}^{\pi - t} \frac{1}{4} + \frac{7}{4}\, d\theta$$

$$= 2\pi - \frac{15}{8}\sigma - \frac{1}{4}t \geq 2\pi - 2\sigma$$

where the equality is achieved when $t = \frac{\sigma}{2}$.

- When $-t \in \left[ \frac{\sigma}{2}, -\frac{3\sigma}{2} + \pi \right]$,

$$\int_{-t}^{\pi - t} \frac{1}{4} \mathbb{1}_{\cos(\theta) \leq \gamma} + \frac{7}{4} \mathbb{1}_{-\cos(\theta + \sigma) \leq \gamma}\, d\theta$$

$$= \int_{-t}^{-\frac{3\sigma}{2} + \pi} \frac{1}{4} + \frac{7}{4}\, d\theta + \int_{-\frac{3\sigma}{2} + \pi}^{-\frac{\sigma}{2} + \pi} \frac{1}{4}\, d\theta + \int_{-\frac{\sigma}{2} + \pi}^{\pi - t} \frac{1}{4} + \frac{7}{4}\, d\theta$$

$$= 2\pi - \frac{7}{4}\sigma\,.$$

- When $-t \in \left[ -\frac{3\sigma}{2} + \pi, -\frac{\sigma}{2} + \pi \right]$,

$$\int_{-t}^{\pi - t} \frac{1}{4} \mathbb{1}_{\cos(\theta) \leq \gamma} + \frac{7}{4} \mathbb{1}_{-\cos(\theta + \sigma) \leq \gamma}\, d\theta$$

$$= \int_{-t}^{-\frac{\sigma}{2} + \pi} \frac{1}{4}\, d\theta + \int_{-\frac{\sigma}{2} + \pi}^{\pi - t} \frac{1}{4} + \frac{7}{4}\, d\theta$$

$$= \frac{\pi}{4} + \frac{7}{8}\sigma - \frac{7}{4}t \geq 2\pi - \frac{7}{4}\sigma$$

where the equality is achieved when $t = \frac{3\sigma}{2} - \pi$.

- When $-t \in \left[ -\frac{\sigma}{2} + \pi, \frac{\sigma}{2} + \pi \right]$,

$$\int_{-t}^{\pi-t} \frac{1}{4} \mathbb{1}_{\cos(\theta) \leq \gamma} + \frac{7}{4} \mathbb{1}_{-\cos(\theta+\sigma) \leq \gamma} \, d\theta$$

$$= \int_{-t}^{-\frac{\sigma}{2}+2\pi} \frac{1}{4} + \frac{7}{4} \, d\theta + \int_{-\frac{\sigma}{2}+2\pi}^{\pi-t} \frac{7}{4} \, d\theta$$

$$= \frac{9\pi}{4} - \frac{1}{8}\sigma + \frac{1}{4}t \geq 2\pi - \frac{1}{4}\sigma$$

where the equality is achieved when $t = -\frac{\sigma}{2} - \pi$.

- When $-t \in \left[ \frac{\sigma}{2} + \pi, -\frac{3\sigma}{2} + 2\pi \right]$,

$$\int_{-t}^{\pi-t} \frac{1}{4} \mathbb{1}_{\cos(\theta) \leq \gamma} + \frac{7}{4} \mathbb{1}_{-\cos(\theta+\sigma) \leq \gamma} \, d\theta$$

$$= \int_{-t}^{-\frac{\sigma}{2}+2\pi} \frac{1}{4} + \frac{7}{4} \, d\theta + \int_{-\frac{\sigma}{2}+2\pi}^{\frac{\sigma}{2}+2\pi} \frac{7}{4} \, d\theta + \int_{\frac{\sigma}{2}+2\pi}^{\pi-t} \frac{1}{4} + \frac{7}{4} \, d\theta$$

$$= 2\pi - \frac{1}{4}\sigma \, .$$

Similarly for $\sigma \in \left[ \frac{\pi}{2}, \pi \right)$, we analyze six cases:

- When $-t \in \left[ -\frac{3\sigma}{2}, \frac{\sigma}{2} - \pi \right]$,

$$\int_{-t}^{\pi-t} \frac{1}{4} \mathbb{1}_{\cos(\theta) \leq \gamma} + \frac{7}{4} \mathbb{1}_{-\cos(\theta+\sigma) \leq \gamma} \, d\theta$$

$$= \int_{-t}^{-\frac{\sigma}{2}} \frac{1}{4} + \frac{7}{4} \, d\theta + \int_{-\frac{\sigma}{2}}^{-\frac{3\sigma}{2}+\pi} \frac{7}{4} \, d\theta$$

$$= \frac{7}{4}\pi - \frac{11}{4}\sigma + 2t \geq \frac{15}{4}\pi - \frac{15}{4}\sigma$$

where the equality is achieved when $t = \pi - \frac{\sigma}{2}$.

- When $-t \in \left[ \frac{\sigma}{2} - \pi, -\frac{\sigma}{2} \right]$,

$$\int_{-t}^{\pi-t} \frac{1}{4} \mathbb{1}_{\cos(\theta) \leq \gamma} + \frac{7}{4} \mathbb{1}_{-\cos(\theta+\sigma) \leq \gamma} \, d\theta$$

$$= \int_{-t}^{-\frac{\sigma}{2}} \frac{1}{4} + \frac{7}{4} \, d\theta + \int_{-\frac{\sigma}{2}}^{-\frac{3\sigma}{2}+\pi} \frac{7}{4} \, d\theta + \int_{\frac{\sigma}{2}}^{\pi-t} \frac{1}{4} \, d\theta$$

$$= 2\pi - \frac{23}{8}\sigma + \frac{7}{4}t \geq 2\pi - 2\sigma$$

where the equality is achieved when $t = \frac{\sigma}{2}$.

- When $-t \in \left[ -\frac{\sigma}{2}, -\frac{3\sigma}{2} + \pi \right]$,

$$\int_{-t}^{\pi-t} \frac{1}{4} \mathbb{1}_{\cos(\theta) \leq \gamma} + \frac{7}{4} \mathbb{1}_{-\cos(\theta+\sigma) \leq \gamma} \, d\theta$$

$$= \int_{-t}^{-\frac{3\sigma}{2}+\pi} \frac{7}{4} \, d\theta + \int_{\frac{\sigma}{2}}^{-\frac{\sigma}{2}+\pi} \frac{1}{4} \, d\theta + \int_{-\frac{\sigma}{2}+\pi}^{\pi-t} \frac{1}{4} + \frac{7}{4} \, d\theta$$

$$= 2\pi - \frac{15}{8}\sigma - \frac{1}{4}t \geq 2\pi - 2\sigma$$

where the equality is achieved when $t = \frac{\sigma}{2}$.

- When $-t \in \left[ -\frac{3\sigma}{2} + \pi, \frac{\sigma}{2} \right]$,

$$\int_{-t}^{\pi-t} \frac{1}{4} \mathbb{1}_{\cos(\theta) \leq \gamma} + \frac{7}{4} \mathbb{1}_{-\cos(\theta+\sigma) \leq \gamma} \, d\theta$$

$$= \int_{\frac{\sigma}{2}}^{-\frac{\sigma}{2}+\pi} \frac{1}{4} \, d\theta + \int_{-\frac{\sigma}{2}+\pi}^{\pi-t} \frac{1}{4} + \frac{7}{4} \, d\theta$$

$$= \frac{\pi}{4} + \frac{3}{4}\sigma - 2t \geq \frac{9}{4}\pi - \frac{9}{4}\sigma$$

where the equality is achieved when $t = \frac{3\sigma}{2} - \pi$.

- When $-t \in \left[ \frac{\sigma}{2}, -\frac{\sigma}{2} + \pi \right]$,

$$\int_{-t}^{\pi-t} \frac{1}{4} \mathbb{1}_{\cos(\theta) \leq \gamma} + \frac{7}{4} \mathbb{1}_{-\cos(\theta+\sigma) \leq \gamma} \, d\theta$$

$$= \int_{-t}^{-\frac{\sigma}{2}+\pi} \frac{7}{4} \, d\theta + \int_{-\frac{\sigma}{2}+\pi}^{\pi-t} \frac{1}{4} + \frac{7}{4} \, d\theta$$

$$= \frac{7\pi}{4} + \frac{1}{8}\sigma - \frac{1}{4}t \geq \frac{7\pi}{4} + \frac{1}{4}\sigma$$

where the equality is achieved when $t = -\frac{\sigma}{2}$.

- When $-t \in \left[ -\frac{\sigma}{2} + \pi, -\frac{3\sigma}{2} + 2\pi \right]$,

$$\int_{-t}^{\pi-t} \frac{1}{4} \mathbb{1}_{\cos(\theta) \leq \gamma} + \frac{7}{4} \mathbb{1}_{-\cos(\theta+\sigma) \leq \gamma} \, d\theta$$

$$= \int_{-t}^{-\frac{\sigma}{2}+2\pi} \frac{1}{4} + \frac{7}{4} \, d\theta + \int_{-\frac{\sigma}{2}+2\pi}^{\pi-t} \frac{7}{4} \, d\theta$$

$$= \frac{9}{4}\pi - \frac{1}{8}\sigma + \frac{1}{4}t \geq \frac{7}{4}\pi + \frac{1}{4}\sigma$$

where the equality is achieved when $t = \frac{3\sigma}{2} - 2\pi$.

Therefore for $\sigma \in (0, \pi)$,

$$\min_{t \in [0, 2\pi)} \int_{-t}^{\pi-t} \frac{1}{4} \mathbb{1}_{\cos(\theta) \leq \gamma} + \frac{7}{4} \mathbb{1}_{-\cos(\theta+\sigma) \leq \gamma} \, d\theta = 2\pi - 2\sigma$$

where the equality is achieved when $t = \frac{\sigma}{2}$. Therefore

$$\mathcal{R}_{\ell_\gamma, \mathcal{H}_{\text{lin}}}^* = \frac{1}{2\pi} \times (2\pi - 2\sigma) = 1 - \frac{\sigma}{\pi},$$

where the unique Bayes classifier satisfies $t_1^* = \frac{\sigma}{2}$.

For continuous margin-based loss $\phi$, by (39) we have

$$\mathcal{R}_{\phi, \mathcal{H}_{\text{lin}}}^* = \frac{1}{2\pi} \min_{t \in [0, 2\pi]} \int_0^\pi \frac{1}{4}\phi(\cos(\theta - t)) \, d\theta + \int_0^\pi \frac{7}{4}\phi(\sin(\theta - t)) \, d\theta$$

$$= \frac{1}{2\pi} \min_{t \in [0, 2\pi]} \int_{-t}^{\pi-t} \frac{1}{4}\phi(\cos(\theta)) + \frac{7}{4}\phi(-\cos(\theta + \sigma)) \, d\theta. \tag{40}$$

If $t^* = \frac{\sigma}{2}$ is the minimizer of $g(t) = \int_{-t}^{\pi-t} \frac{1}{4}\phi(\cos(\theta)) + \frac{7}{4}\phi(-\cos(\theta + \sigma)) \, d\theta$, $t \in [0, 2\pi]$, since $\frac{\sigma}{2}$ is not at the boundary of $[0, 2\pi]$, we need

$$g'\left(\frac{\sigma}{2}\right) = 0.$$

Since $\phi$ is continuous, by Leibniz Integral Rule, we have

$$g'\left(\frac{\sigma}{2}\right) = -\frac{1}{4}\phi\left(\cos\left(\pi - \frac{\sigma}{2}\right)\right) - \frac{7}{4}\phi\left(-\cos\left(\pi + \frac{\sigma}{2}\right)\right) + \frac{1}{4}\phi\left(\cos\left(-\frac{\sigma}{2}\right)\right) + \frac{7}{4}\phi\left(-\cos\left(\frac{\sigma}{2}\right)\right)$$

$$= -\frac{1}{4}\phi\left(-\cos\left(\frac{\sigma}{2}\right)\right) - \frac{7}{4}\phi\left(\cos\left(\frac{\sigma}{2}\right)\right) + \frac{1}{4}\phi\left(\cos\left(\frac{\sigma}{2}\right)\right) + \frac{7}{4}\phi\left(-\cos\left(\frac{\sigma}{2}\right)\right)$$

$$= \frac{3}{2}\phi\left(-\cos\left(\frac{\sigma}{2}\right)\right) - \frac{3}{2}\phi\left(\cos\left(\frac{\sigma}{2}\right)\right).$$

Thus if $t^* = \frac{\sigma}{2}$ is the minimizer of $\mathcal{R}^*_{\phi,\mathcal{H}_{\mathrm{lin}}}$, we need $\phi$ satisfies

$$\phi\left(-\cos\left(\frac{\sigma}{2}\right)\right) = \phi\left(\cos\left(\frac{\sigma}{2}\right)\right). \tag{41}$$

Therefore, if $\phi$ is $\mathcal{H}_{\mathrm{lin}}$-consistent with respect to $\ell_\gamma$, we need $\phi$ satisfies (41) for any $\sigma \in (0, \pi)$. Namely $\phi$ satisfies

$$\phi(-\tau) = \phi(\tau), \quad \tau \in [0, 1).$$

Note in our example, $\tau \in [-1, 1]$, $\phi$ is continuous. We obtain that if $\phi$ is $\mathcal{H}_{\mathrm{lin}}$-consistent with respect to $\ell_\gamma$, $\phi$ must be even function in $[-1, 1]$. Next we claim that if $\phi$ is even function in $[-1, 1]$, $\phi$ is not $\mathcal{H}_{\mathrm{lin}}$-consistent with respect to $\ell_\gamma$. Indeed, for the distribution $y = 1$ if $\theta \in (0, \pi)$ and $y = -1$ if $\theta \in (\pi, 2\pi)$, we have

$$\begin{aligned}
\mathcal{R}^*_{\phi,\mathcal{H}_{\mathrm{lin}}} &= \frac{1}{2\pi} \min_{t \in [0, 2\pi]} \int_0^\pi \phi(\cos(\theta - t)) + \int_\pi^{2\pi} \phi(-\cos(\theta - t)) \, d\theta \\
&= \frac{1}{\pi} \min_{t \in [0, 2\pi]} \int_0^\pi \phi(\cos(\theta - t)) \, d\theta \\
&= \frac{1}{\pi} \min_{t \in [0, 2\pi]} \int_{-t}^{\pi - t} \phi(\cos(\theta)) \, d\theta.
\end{aligned} \tag{42}$$

Note that when $\phi$ is even function in $[-1, 1]$, $h(t) = \int_{-t}^{\pi - t} \phi(\cos(\theta)) \, d\theta$ satisfies

$$h'(t) = -\phi(-\cos(t)) + \phi(\cos(t)) = 0, \quad t \in [0, 2\pi].$$

Thus $h(t)$ is a constant for $t \in [0, 2\pi]$ and $\mathcal{R}^*_{\phi,\mathcal{H}_{\mathrm{lin}}}$ can be attained for any classifier $t \in [0, 2\pi]$. However, $\mathcal{R}^*_{\ell_\gamma,\mathcal{H}_{\mathrm{lin}}}$ can not be attained for any classifier $t \in [0, 2\pi]$ with respect to this distribution. Therefore when $\phi$ is even function in $[-1, 1]$, $\phi$ is not $\mathcal{H}_{\mathrm{lin}}$-consistent with respect to $\ell_\gamma$. By the claim, we conclude that any continuous margin-based loss $\phi$ is not $\mathcal{H}_{\mathrm{lin}}$-consistent with respect to $\ell_\gamma$.

Furthermore, as shown by Awasthi et al. (2020), for a continuous and non-increasing margin-based loss $\phi$, when $f \in \mathcal{H}_{\mathrm{lin}}$, the supremum-based surrogate loss can be expressed as follows:

$$\tilde{\phi}(f, \mathbf{x}, y) = \sup_{\mathbf{x}': \|\mathbf{x} - \mathbf{x}'\| \leq \gamma} \phi(yf(\mathbf{x}')) = \phi\left(\inf_{\|\mathbf{s}\| \leq 1} (yf(\mathbf{x} + \gamma\mathbf{s}))\right) = \phi(y(\mathbf{w} \cdot \mathbf{x}) - \gamma) = \psi(y(\mathbf{w} \cdot \mathbf{x})),$$

where $\psi(t) = \phi(t - \gamma)$ is also a continuous margin-based loss. In view of the results above, we conclude that the supremum-based surrogate loss $\tilde{\phi}$ is also not $\mathcal{H}_{\mathrm{lin}}$-consistent with respect to $\ell_\gamma$. $\square$

### E.7 Proof of Theorem 20, Theorem 21 and Theorem 23

Since the proofs adopt some results of (Steinwart, 2007), we introduce the notation used in (Steinwart, 2007) to make the proofs more clear. In this section, we denote the loss $\ell(f, \mathbf{x}, y)$ defined on a particular hypothesis set $\mathcal{H}$ as $\ell_{\mathcal{H}}(f, \mathbf{x}, y)$. For a joint distribution $\mathcal{P}$ over $\mathcal{X} \times \mathcal{Y}$, the corresponding conditional distribution and marginal distribution are denoted as $\mathcal{P}(\cdot|\mathbf{x})$ and $\mathcal{P}_X$ respectively. In (Steinwart, 2007), given a distribution $\mathcal{P}$ over $\mathcal{X} \times \mathcal{Y}$, the $\ell_{\mathcal{H}}$-risk and the inner $\ell_{\mathcal{H}}$-risk of a classifier $f \in \mathcal{H}$ for the loss $\ell_{\mathcal{H}}$ are denoted by

$$\mathcal{R}_{\ell_{\mathcal{H}},\mathcal{P}}(f) = \mathbb{E}_{(\mathbf{x},y)\sim\mathcal{P}}[\ell_{\mathcal{H}}(f, \mathbf{x}, y)], \quad \mathcal{C}_{\ell_{\mathcal{H}},\mathcal{P}(\cdot|\mathbf{x}),\mathbf{x}}(f) = \mathbb{E}_{y\sim\mathcal{P}(\cdot|\mathbf{x})}[\ell_{\mathcal{H}}(f, \mathbf{x}, y)].$$

Accordingly, the minimal $\ell_{\mathcal{H}}$-risk and minimal inner $\ell_{\mathcal{H}}$-risk are denoted by $\mathcal{R}^*_{\ell_{\mathcal{H}},\mathcal{P}}$ and $\mathcal{C}^*_{\ell_{\mathcal{H}},\mathcal{P}(\cdot|\mathbf{x}),\mathbf{x}}$. For convenience, we will alternately use the notation of risk and inner risk presented above and in Section 2 for the proofs. Next, we introduce the minimizability proposed in (Steinwart, 2007).

**Definition 31** (minimizability). *Given a distribution $\mathcal{P}$ over $\mathcal{X} \times \mathcal{Y}$ and a hypothesis set $\mathcal{H}$. We say that loss $\ell_{\mathcal{H}}(f, \mathbf{x}, y)$ is $\mathcal{P}$-minimizable if for all $\epsilon > 0$ there exists $f_\epsilon \in \mathcal{H}$ such that for all $\mathbf{x} \in \mathcal{X}$ we have*

$$\mathcal{C}_{\ell_{\mathcal{H}},\mathcal{P}(\cdot|\mathbf{x}),\mathbf{x}}(f_\epsilon) < \mathcal{C}^*_{\ell_{\mathcal{H}},\mathcal{P}(\cdot|\mathbf{x}),\mathbf{x}} + \epsilon.$$

The following lemmas are useful in the proofs of Theorem 20 and Theorem 23.

**Lemma 32.** *Given a distribution $\mathcal{P}$ over $\mathcal{X} \times \mathcal{Y}$ and a hypothesis set $\mathcal{H}$. Let $\phi$ be a margin-based loss. Then $\phi_{\mathcal{H}_{\mathrm{all}}}$ is $\mathcal{P}$-minimizable. If there exists $f^* \in \mathcal{H} \subset \mathcal{H}_{\mathrm{all}}$ such that $\mathcal{R}^*_{\phi_{\mathcal{H}_{\mathrm{all}}},\mathcal{P}} = \mathcal{R}_{\phi_{\mathcal{H}},\mathcal{P}}(f^*)$, then $\phi_{\mathcal{H}}$ is also $\mathcal{P}$-minimizable in the almost surely sense.*

*Proof.* By Theorem 3.2 of (Steinwart, 2007), since $\mathcal{C}^*_{\phi_{\mathcal{H}_{\text{all}}},\mathcal{P}(\cdot|\mathbf{x}),\mathbf{x}} < \infty$ for all $\mathbf{x} \in \mathcal{X}$, $\phi_{\mathcal{H}_{\text{all}}}$ is $\mathcal{P}$-minimizable. Therefore, by Lemma 2.5 of (Steinwart, 2007), we have

$$\mathcal{R}^*_{\phi_{\mathcal{H}_{\text{all}}},\mathcal{P}} = \int_{\mathcal{X}} \mathcal{C}^*_{\phi_{\mathcal{H}_{\text{all}}},\mathcal{P}(\cdot|\mathbf{x}),\mathbf{x}} \, d\mathcal{P}_X(\mathbf{x}).$$

Then by the assumption,

$$\int_{\mathcal{X}} \mathcal{C}_{\phi_{\mathcal{H}},\mathcal{P}(\cdot|\mathbf{x}),\mathbf{x}}(f^*) \, d\mathcal{P}_X(\mathbf{x}) = \mathcal{R}_{\phi_{\mathcal{H}},\mathcal{P}}(f^*) = \mathcal{R}^*_{\phi_{\mathcal{H}_{\text{all}}},\mathcal{P}} = \int_{\mathcal{X}} \mathcal{C}^*_{\phi_{\mathcal{H}_{\text{all}}},\mathcal{P}(\cdot|\mathbf{x}),\mathbf{x}} \, d\mathcal{P}_X(\mathbf{x}).$$

Since

$$\mathcal{C}^*_{\phi_{\mathcal{H}_{\text{all}}},\mathcal{P}(\cdot|\mathbf{x}),\mathbf{x}} \leq \mathcal{C}_{\phi_{\mathcal{H}},\mathcal{P}(\cdot|\mathbf{x}),\mathbf{x}}(f^*),$$

for almost all $\mathbf{x} \in \mathcal{X}$,

$$\mathcal{C}^*_{\phi_{\mathcal{H}_{\text{all}}},\mathcal{P}(\cdot|\mathbf{x}),\mathbf{x}} = \mathcal{C}_{\phi_{\mathcal{H}},\mathcal{P}(\cdot|\mathbf{x}),\mathbf{x}}(f^*).$$

Thus, for all $\epsilon > 0$, for almost all $\mathbf{x} \in \mathcal{X}$ we have

$$\mathcal{C}_{\phi_{\mathcal{H}},\mathcal{P}(\cdot|\mathbf{x}),\mathbf{x}}(f^*) = \mathcal{C}^*_{\phi_{\mathcal{H}_{\text{all}}},\mathcal{P}(\cdot|\mathbf{x}),\mathbf{x}} < \mathcal{C}^*_{\phi_{\mathcal{H}_{\text{all}}},\mathcal{P}(\cdot|\mathbf{x}),\mathbf{x}} + \epsilon \leq \mathcal{C}^*_{\phi_{\mathcal{H}},\mathcal{P}(\cdot|\mathbf{x}),\mathbf{x}} + \epsilon.$$

This completes the proof. $\qquad\square$

**Lemma 33.** *Given a distribution $\mathcal{P}$ over $\mathcal{X} \times \mathcal{Y}$ and a hypothesis set $\mathcal{H}$. Let $\phi$ be a margin-based loss. If for $\eta \geq 0$, there exists $f^* \in \mathcal{H} \subset \mathcal{H}_{\text{all}}$ such that $\mathcal{R}_{\phi_{\mathcal{H}},\mathcal{P}}(f^*) \leq \mathcal{R}^*_{\phi_{\mathcal{H}_{\text{all}}},\mathcal{P}} + \eta$, then $\phi_{\mathcal{H}}$ satisfies*

$$\int_{\mathcal{X}} \mathcal{C}^*_{\phi_{\mathcal{H}},\mathcal{P}(\cdot|\mathbf{x}),\mathbf{x}} \, d\mathcal{P}_X(\mathbf{x}) \leq \mathcal{R}^*_{\phi_{\mathcal{H}},\mathcal{P}} \leq \int_{\mathcal{X}} \mathcal{C}^*_{\phi_{\mathcal{H}},\mathcal{P}(\cdot|\mathbf{x}),\mathbf{x}} \, d\mathcal{P}_X(\mathbf{x}) + \eta.$$

*Proof.* By Lemma 32, $\phi_{\mathcal{H}_{\text{all}}}$ is $\mathcal{P}$-minimizable. Then by Lemma 2.5 of (Steinwart, 2007), we have

$$\mathcal{R}^*_{\phi_{\mathcal{H}_{\text{all}}},\mathcal{P}} = \int_{\mathcal{X}} \mathcal{C}^*_{\phi_{\mathcal{H}_{\text{all}}},\mathcal{P}(\cdot|\mathbf{x}),\mathbf{x}} \, d\mathcal{P}_X(\mathbf{x}).$$

Therefore,

$$\mathcal{R}^*_{\phi_{\mathcal{H}},\mathcal{P}} \leq \mathcal{R}_{\phi_{\mathcal{H}},\mathcal{P}}(f^*) \leq \int_{\mathcal{X}} \mathcal{C}^*_{\phi_{\mathcal{H}_{\text{all}}},\mathcal{P}(\cdot|\mathbf{x}),\mathbf{x}} \, d\mathcal{P}_X(\mathbf{x}) + \eta \leq \int_{\mathcal{X}} \mathcal{C}^*_{\phi_{\mathcal{H}},\mathcal{P}(\cdot|\mathbf{x}),\mathbf{x}} \, d\mathcal{P}_X(\mathbf{x}) + \eta.$$

Also,

$$\int_{\mathcal{X}} \mathcal{C}^*_{\phi_{\mathcal{H}},\mathcal{P}(\cdot|\mathbf{x}),\mathbf{x}} \, d\mathcal{P}_X(\mathbf{x}) \leq \int_{\mathcal{X}} \inf_{f \in \mathcal{H}} \mathcal{C}_{\phi_{\mathcal{H}},\mathcal{P}(\cdot|\mathbf{x}),\mathbf{x}}(f) \, d\mathcal{P}_X(\mathbf{x})$$

$$\leq \inf_{f \in \mathcal{H}} \int_{\mathcal{X}} \mathcal{C}_{\phi_{\mathcal{H}},\mathcal{P}(\cdot|\mathbf{x}),\mathbf{x}}(f) \, d\mathcal{P}_X(x) = \mathcal{R}^*_{\phi_{\mathcal{H}},\mathcal{P}}.$$

$\qquad\square$

**Lemma 34.** *Given a distribution $\mathcal{P}$ over $\mathcal{X} \times \mathcal{Y}$ with random variables $X$ and $Y$ and a hypothesis set $\mathcal{H}$ such that $\mathcal{R}^*_{\ell_\gamma,\mathcal{H}} = \mathcal{R}_{\ell_\gamma}(f^*) = 0$, where $f^* \in \mathcal{H}$ achieves the Bayes risk. Then $f^*$ correctly classifies $\mathbf{x} \in \mathcal{X}$ in the almost surely sense and for almost all $\mathbf{x} \in \mathcal{X}$, any $\mathbf{x}' \in \{\mathbf{x}': \|\mathbf{x}' - \mathbf{x}\| \leq \gamma\}$ has same label as $\mathbf{x}$.*

*Proof.* Since $\mathcal{R}^*_{\ell_{\gamma\mathcal{H}},\mathcal{P}} = \mathcal{R}^*_{\ell_\gamma,\mathcal{H}} = 0$, the distribution $\mathcal{P}$ is $\mathcal{H}$-realizable. Therefore $\mathbb{P}(Y = 1|X = \mathbf{x}) = 1$ or $0$. Thus

$$\mathcal{C}_{\ell_{\gamma\mathcal{H}},\mathcal{P}(\cdot|\mathbf{x}),\mathbf{x}}(f) = \begin{cases} \sup\limits_{\mathbf{x}':\|\mathbf{x}-\mathbf{x}'\|\leq\gamma} \mathbb{1}_{\{f(\mathbf{x}')\leq 0\}}, & \text{if } \mathbb{P}(Y = 1|X = \mathbf{x}) = 1, \\ \sup\limits_{\mathbf{x}':\|\mathbf{x}-\mathbf{x}'\|\leq\gamma} \mathbb{1}_{\{-f(\mathbf{x}')\leq 0\}}, & \text{if } \mathbb{P}(Y = 1|X = \mathbf{x}) = 0, \end{cases}$$

Since $\mathcal{R}_{\ell_{\gamma\mathcal{H}},\mathcal{P}}(f^*) = \mathcal{R}_{\ell_\gamma}(f^*) = 0$, we have $\mathcal{C}_{\ell_{\gamma\mathcal{H}},\mathcal{P}(\cdot|\mathbf{x}),\mathbf{x}}(f^*) = 0$ for almost all $\mathbf{x} \in \mathcal{X}$. When $\mathbb{P}(Y = 1|X = \mathbf{x}) = 1$, we obtain

$$\sup_{\mathbf{x}':\|\mathbf{x}-\mathbf{x}'\|\leq\gamma} \mathbb{1}_{\{f^*(\mathbf{x}')\leq 0\}} = 0 \implies f^*(\mathbf{x}') > 0 \text{ for any } \mathbf{x}' \in \{\mathbf{x}': \|\mathbf{x}' - \mathbf{x}\| \leq \gamma\}. \tag{43}$$

When $\mathbb{P}(Y = 1|X = \mathbf{x}) = 0$, we obtain

$$\sup_{\mathbf{x}':\|\mathbf{x}-\mathbf{x}'\|\leq\gamma} \mathbb{1}_{\{-f^*(\mathbf{x}')\leq 0\}} = 0 \implies f^*(\mathbf{x}') < 0 \text{ for any } \mathbf{x}' \in \{\mathbf{x}': \|\mathbf{x}' - \mathbf{x}\| \leq \gamma\}. \tag{44}$$

Thus $f^*(\mathbf{x}) > 0$ when $\mathbb{P}(Y = 1|X = \mathbf{x}) = 1$ and $f^*(\mathbf{x}) < 0$ when $\mathbb{P}(Y = 1|X = \mathbf{x}) = 0$ for almost all $\mathbf{x} \in \mathcal{X}$. Therefore $f^*$ correctly classify $\mathbf{x} \in \mathcal{X}$ in the almost surely sense. Furthermore, by (43) and (44), for almost all $\mathbf{x} \in \mathcal{X}$, any $\mathbf{x}' \in \{\mathbf{x}': \|\mathbf{x}' - \mathbf{x}\| \leq \gamma\}$ has same label as $\mathbf{x}$. $\qquad\square$

**Lemma 35.** *Given a distribution $\mathcal{P}$ over $\mathcal{X} \times \mathcal{Y}$ and a hypothesis set $\mathcal{H}$ such that $\mathcal{R}^*_{\ell_\gamma, \mathcal{H}} = 0$. Let $\phi$ be a margin-based loss and $\tilde{\phi}(f, \mathbf{x}, y) = \sup_{\mathbf{x}': \|\mathbf{x} - \mathbf{x}'\| \leq \gamma} \phi(yf(\mathbf{x}'))$. If $\phi_{\mathcal{H}}$ is $\mathcal{P}$-minimizable in the almost surely sense, then $\tilde{\phi}_{\mathcal{H}}$ is also $\mathcal{P}$-minimizable in the almost surely sense.*

*Proof.* As shown by Awasthi et al. (2020), $\tilde{\phi}$ has the equivalent form

$$\tilde{\phi}(f, \mathbf{x}, y) = \phi\left(\inf_{\mathbf{x}': \|\mathbf{x} - \mathbf{x}'\| \leq \gamma} (yf(\mathbf{x}'))\right).$$

Since $\mathcal{R}^*_{\ell_{\gamma\mathcal{H}}, \mathcal{P}} = \mathcal{R}^*_{\ell_\gamma, \mathcal{H}} = 0$, the distribution $\mathcal{P}$ is $\mathcal{H}$-realizable. Therefore $\mathbb{P}(Y = 1 | X = \mathbf{x}) = 1$ or $0$. Thus

$$\mathcal{C}_{\phi_{\mathcal{H}}, \mathcal{P}(\cdot|\mathbf{x}), \mathbf{x}}(f) = \begin{cases} \phi(f(\mathbf{x})), & \text{if } \mathbb{P}(Y = 1 | X = \mathbf{x}) = 1, \\ \phi(-f(\mathbf{x})), & \text{if } \mathbb{P}(Y = 1 | X = \mathbf{x}) = 0, \end{cases}$$

Note $\tilde{\phi}(f, \mathbf{x}, +1) = \phi(\inf_{\mathbf{x}': \|\mathbf{x} - \mathbf{x}'\| \leq \gamma} f(\mathbf{x}')) = \phi(f(m_{f, \mathbf{x}}))$, where w.l.o.g. we assume that $f$ is continuous and $m_{f, \mathbf{x}} \in \{\mathbf{x}': \|\mathbf{x} - \mathbf{x}'\| \leq \gamma\}$ is the point such that $\min_{\mathbf{x}': \|\mathbf{x} - \mathbf{x}'\| \leq \gamma} f(\mathbf{x}') = f(m_{f, \mathbf{x}})$. Similarly $\tilde{\phi}(f, \mathbf{x}, -1) = \phi(-\sup_{\mathbf{x}': \|\mathbf{x} - \mathbf{x}'\| \leq \gamma} f(\mathbf{x}')) = \phi(-f(M_{f, \mathbf{x}}))$, where w.l.o.g. we assume that $f$ is continuous and $M_{f, \mathbf{x}} \in \{\mathbf{x}': \|\mathbf{x} - \mathbf{x}'\| \leq \gamma\}$ is the point such that $\max_{\mathbf{x}': \|\mathbf{x} - \mathbf{x}'\| \leq \gamma} f(\mathbf{x}') = f(M_{f, \mathbf{x}})$. Then for $\tilde{\phi}_{\mathcal{H}}$, we have

$$\mathcal{C}_{\tilde{\phi}_{\mathcal{H}}, \mathcal{P}(\cdot|\mathbf{x}), \mathbf{x}}(f) = \begin{cases} \phi(f(m_{f, \mathbf{x}})), & \text{if } \mathbb{P}(Y = 1 | X = \mathbf{x}) = 1, \\ \phi(-f(M_{f, \mathbf{x}})), & \text{if } \mathbb{P}(Y = 1 | X = \mathbf{x})) = 0, \end{cases}$$

Since $\phi_{\mathcal{H}}$ is $\mathcal{P}$-minimizable in the almost surely sense, by the definition for all $\epsilon > 0$, there exists an $f^* \in \mathcal{H}$ such that for almost all $\mathbf{x} \in \mathcal{X}$ we have

$$\mathcal{C}_{\phi_{\mathcal{H}}, \mathcal{P}(\cdot|\mathbf{x}), \mathbf{x}}(f^*) < \mathcal{C}^*_{\phi_{\mathcal{H}}, \mathcal{P}(\cdot|\mathbf{x}), \mathbf{x}} + \epsilon.$$

When $\mathbb{P}(Y = 1 | X = \mathbf{x}) = 1$, we obtain

$$\mathcal{C}_{\tilde{\phi}_{\mathcal{H}}, \mathcal{P}(\cdot|\mathbf{x}), \mathbf{x}}(f^*) = \phi(f^*(m_{f^*, \mathbf{x}})) = \mathcal{C}_{\phi_{\mathcal{H}}, \mathcal{P}(\cdot|m_{f^*, \mathbf{x}}), m_{f^*, \mathbf{x}}}(f^*) < \mathcal{C}^*_{\phi_{\mathcal{H}}, \mathcal{P}(\cdot|m_{f^*, \mathbf{x}}), m_{f^*, \mathbf{x}}} + \epsilon \leq \mathcal{C}^*_{\tilde{\phi}_{\mathcal{H}}, \mathcal{P}(\cdot|\mathbf{x}), \mathbf{x}} + \epsilon$$

where we used the fact that $m_{f^*, \mathbf{x}}$ satisfies $\mathbb{P}(Y = 1 | X = m_{f^*, \mathbf{x}}) = 1$ by Lemma 34 and $\phi$ is non-increasing. Similarly, when $\mathbb{P}(Y = 1 | X = \mathbf{x}) = 0$, we obtain

$$\mathcal{C}_{\tilde{\phi}_{\mathcal{H}}, \mathcal{P}(\cdot|\mathbf{x}), \mathbf{x}}(f^*) = \phi(-f^*(M_{f^*, \mathbf{x}})) = \mathcal{C}_{\phi_{\mathcal{H}}, \mathcal{P}(\cdot|M_{f^*, \mathbf{x}}), M_{f^*, \mathbf{x}}}(f^*) < \mathcal{C}^*_{\phi_{\mathcal{H}}, \mathcal{P}(\cdot|M_{f^*, \mathbf{x}}), M_{f^*, \mathbf{x}}} + \epsilon \leq \mathcal{C}^*_{\tilde{\phi}_{\mathcal{H}}, \mathcal{P}(\cdot|\mathbf{x}), \mathbf{x}} + \epsilon$$

where we used the fact that $M_{f^*, \mathbf{x}}$ satisfies $\mathbb{P}(Y = 1 | X = M_{f^*, \mathbf{x}}) = 0$ by Lemma 34 and $\phi$ is non-increasing. Above all, for all $\epsilon > 0$, there exists an $f^* \in \mathcal{H}$ such that for almost all $\mathbf{x} \in \mathcal{X}$ we have

$$\mathcal{C}_{\tilde{\phi}_{\mathcal{H}}, \mathcal{P}(\cdot|\mathbf{x}), \mathbf{x}}(f^*) < \mathcal{C}^*_{\tilde{\phi}_{\mathcal{H}}, \mathcal{P}(\cdot|\mathbf{x}), \mathbf{x}} + \epsilon.$$

$\square$

We modify Theorem 2.8 of (Steinwart, 2007), whose proof is very similar.

**Theorem 36.** *Given a distribution $\mathcal{P}$ over $\mathcal{X} \times \mathcal{Y}$ and a hypothesis set $\mathcal{H}$. Let $\ell_1: \mathcal{H} \times \mathcal{X} \times \mathcal{Y} \to [0, +\infty]$, $\ell_2: \mathcal{H} \times \mathcal{X} \times \mathcal{Y} \to [0, +\infty]$ be two loss functions such that $\mathcal{R}^*_{\ell_1, \mathcal{P}} = \int_{\mathcal{X}} \mathcal{C}^*_{\ell_1, \mathcal{P}(\cdot|\mathbf{x}), \mathbf{x}} d\mathcal{P}_X(\mathbf{x}) < +\infty$ and $\int_{\mathcal{X}} \mathcal{C}^*_{\ell_2, \mathcal{P}(\cdot|\mathbf{x}), \mathbf{x}} d\mathcal{P}_X(\mathbf{x}) \leq \mathcal{R}^*_{\ell_2, \mathcal{P}} \leq \int_{\mathcal{X}} \mathcal{C}^*_{\ell_2, \mathcal{P}(\cdot|\mathbf{x}), \mathbf{x}} d\mathcal{P}_X(\mathbf{x}) + \eta < +\infty$ for $\eta \geq 0$. Furthermore, assume that there exist a function $b \in \mathcal{L}_1(\mathcal{P}_X)$ and measurable functions $\delta(\epsilon, \cdot) : X \to (0, +\infty)$, $\epsilon > 0$, such that*

$$\mathcal{C}_{\ell_1, \mathcal{P}(\cdot|\mathbf{x}), \mathbf{x}}(f) \leq \mathcal{C}^*_{\ell_1, \mathcal{P}(\cdot|\mathbf{x}), \mathbf{x}} + b(\mathbf{x})$$

*and*

$$\mathcal{C}_{\ell_2, \mathcal{P}(\cdot|\mathbf{x}), \mathbf{x}}(f) < \mathcal{C}^*_{\ell_2, \mathcal{P}(\cdot|\mathbf{x}), \mathbf{x}} + \delta(\epsilon, \mathbf{x}) \implies \mathcal{C}_{\ell_1, \mathcal{P}(\cdot|\mathbf{x}), \mathbf{x}}(f) < \mathcal{C}^*_{\ell_1, \mathcal{P}(\cdot|\mathbf{x}), \mathbf{x}} + \epsilon$$

*for all $\mathbf{x} \in \mathcal{X}$, $\epsilon > 0$ and $f \in \mathcal{H}$. Then, for all $\epsilon > 0$ there exists $\delta > 0$ such that for all $f \in \mathcal{H}$ we have*

$$\mathcal{R}_{\ell_2, \mathcal{P}}(f) + \eta < \mathcal{R}^*_{\ell_2, \mathcal{P}} + \delta \implies \mathcal{R}_{\ell_1, \mathcal{P}}(f) < \mathcal{R}^*_{\ell_1, \mathcal{P}} + \epsilon.$$

*Proof.* Define $\mathcal{C}_{1,\mathbf{x}}(f) = \mathcal{C}_{\ell_1,\mathcal{P}(\cdot|\mathbf{x}),\mathbf{x}}(f) - \mathcal{C}^*_{\ell_1,\mathcal{P}(\cdot|\mathbf{x}),\mathbf{x}}$ and $\mathcal{C}_{2,\mathbf{x}}(f) = \mathcal{C}_{\ell_2,\mathcal{P}(\cdot|\mathbf{x}),\mathbf{x}}(f) - \mathcal{C}^*_{\ell_2,\mathcal{P}(\cdot|\mathbf{x}),\mathbf{x}}$ for $\mathbf{x} \in \mathcal{X}$, $f \in \mathcal{H}$. For a fixed $\epsilon > 0$, define $h(\mathbf{x}) = \delta(\epsilon, \mathbf{x})$, $\mathbf{x} \in \mathcal{X}$. Then for all $\mathbf{x} \in \mathcal{X}$ and $f \in \mathcal{H}$ such that $\mathcal{C}_{1,\mathbf{x}}(f) \geq \epsilon$, we have $\mathcal{C}_{2,\mathbf{x}}(f) \geq h(\mathbf{x})$. Therefore,

$$\mathcal{R}_{\ell_2,\mathcal{P}}(f) - \mathcal{R}^*_{\ell_2,\mathcal{P}} + \eta \geq \mathcal{R}_{\ell_2,\mathcal{P}}(f) - \int_{\mathcal{X}} \mathcal{C}^*_{\ell_2,\mathcal{P}(\cdot|\mathbf{x}),\mathbf{x}} \, d\mathcal{P}_X(\mathbf{x})$$

$$= \int_{\mathcal{X}} \mathcal{C}_{2,\mathbf{x}}(f) \, d\mathcal{P}_X(\mathbf{x}) \geq \int_{\mathcal{C}_{1,\mathbf{x}}(f) \geq \epsilon} h(\mathbf{x}) \, d\mathcal{P}_X(\mathbf{x}),$$

for all $f \in \mathcal{H}$. Furthermore, since $h(\mathbf{x}) > 0$ for all $\mathbf{x} \in \mathcal{X}$, the measure $\nu := b\mathcal{P}_X$ is absolutely continuous with respect to $\mu := h\mathcal{P}_X$, and thus there exists $\delta > 0$ such that $\nu(A) < \epsilon$ for all measurable $A \subset X$ with $\mu(A) < \delta$. Therefore, for $f \in \mathcal{H}$ with $\mathcal{R}_{\ell_2,\mathcal{P}}(f) - \mathcal{R}^*_{\ell_2,\mathcal{P}} + \eta < \delta$ and $A := \{\mathbf{x} \in \mathcal{X}, \mathcal{C}_{1,\mathbf{x}}(f) \geq \epsilon\}$, we obtain

$$\mathcal{R}_{\ell_1,\mathcal{P}}(f) - \mathcal{R}^*_{\ell_1,\mathcal{P}} = \int_{\mathcal{C}_{1,\mathbf{x}}(f) \geq \epsilon} \mathcal{C}_{1,\mathbf{x}}(f) \, d\mathcal{P}_X(\mathbf{x}) + \int_{\mathcal{C}_{1,\mathbf{x}}(f) < \epsilon} \mathcal{C}_{1,\mathbf{x}}(f) \, d\mathcal{P}_X(\mathbf{x})$$

$$\leq \int_A b(\mathbf{x}) \, d\mathcal{P}_X(\mathbf{x}) + \epsilon < 2\epsilon.$$

$\square$

**Theorem 20.** *Let $\mathcal{P}$ be a distribution over $\mathcal{X} \times \mathcal{Y}$ and $\mathcal{H}$ a hypothesis set for which $\mathcal{R}^*_{\ell_\gamma,\mathcal{H}} = 0$. Let $\phi$ be a margin-based loss. If for $\eta \geq 0$, there exists $f^* \in \mathcal{H} \subset \mathcal{H}_{\mathrm{all}}$ such that $\mathcal{R}_\phi(f^*) \leq \mathcal{R}^*_{\phi,\mathcal{H}_{\mathrm{all}}} + \eta < +\infty$ and $\phi$ is $\mathcal{H}$-calibrated with respect to $\ell_\gamma$, then for all $\epsilon > 0$ there exists $\delta > 0$ such that for all $f \in \mathcal{H}$,*

$$\mathcal{R}_\phi(f) + \eta < \mathcal{R}^*_{\phi,\mathcal{H}} + \delta \implies \mathcal{R}_{\ell_\gamma}(f) < \mathcal{R}^*_{\ell_\gamma,\mathcal{H}} + \epsilon.$$

*Proof.* Since $\mathcal{R}^*_{\ell_{\gamma\mathcal{H}},\mathcal{P}} = \mathcal{R}^*_{\ell_\gamma,\mathcal{H}} = 0$, we obtain

$$0 \leq \int_{\mathcal{X}} \mathcal{C}^*_{\ell_{\gamma\mathcal{H}},\mathcal{P}(\cdot|\mathbf{x}),\mathbf{x}} \, d\mathcal{P}_X(\mathbf{x}) \leq \mathcal{R}^*_{\ell_{\gamma\mathcal{H}},\mathcal{P}} = 0.$$

By Lemma 33, $\phi_\mathcal{H}$ satisfies

$$\int_X \mathcal{C}^*_{\phi_\mathcal{H},\mathcal{P}(\cdot|\mathbf{x}),\mathbf{x}} \, d\mathcal{P}_X(\mathbf{x}) \leq \mathcal{R}^*_{\phi_\mathcal{H},\mathcal{P}} \leq \int_{\mathcal{X}} \mathcal{C}^*_{\phi_\mathcal{H},\mathcal{P}(\cdot|\mathbf{x}),\mathbf{x}} \, d\mathcal{P}_X(\mathbf{x}) + \eta < +\infty.$$

Since for all $\mathbf{x} \in \mathcal{X}$ and $f \in \mathcal{H}$, $\mathcal{C}_{\ell_{\gamma\mathcal{H}},\mathcal{P}(\cdot|\mathbf{x}),\mathbf{x}}(f) \leq 1$, we obtain

$$\mathcal{C}_{\ell_{\gamma\mathcal{H}},\mathcal{P}(\cdot|\mathbf{x}),\mathbf{x}}(f) \leq \mathcal{C}^*_{\ell_{\gamma\mathcal{H}},\mathcal{P}(\cdot|\mathbf{x}),\mathbf{x}} + 1.$$

Also, since $\phi$ is $\mathcal{H}$-calibrated with respect to $\ell_\gamma$, for all $x \in \mathcal{X}$, $\epsilon > 0$ and $f \in \mathcal{H}$, there exists $\delta > 0$ such that

$$\mathcal{C}_{\phi_\mathcal{H},\mathcal{P}(\cdot|\mathbf{x}),x}(f) < \mathcal{C}^*_{\phi_\mathcal{H},\mathcal{P}(\cdot|\mathbf{x}),\mathbf{x}} + \delta \implies \mathcal{C}_{\ell_{\gamma\mathcal{H}},\mathcal{P}(\cdot|\mathbf{x}),\mathbf{x}}(f) < \mathcal{C}^*_{\ell_{\gamma\mathcal{H}},\mathcal{P}(\cdot|\mathbf{x}),\mathbf{x}} + \epsilon.$$

Therefore by Theorem 36, for all $\epsilon > 0$ there exists $\delta > 0$ such that for all $f \in \mathcal{H}$ we have

$$\mathcal{R}_{\phi_\mathcal{H},\mathcal{P}}(f) + \eta < \mathcal{R}^*_{\phi_\mathcal{H},\mathcal{P}} + \delta \implies \mathcal{R}_{\ell_{\gamma\mathcal{H}},\mathcal{P}}(f) < \mathcal{R}^*_{\ell_{\gamma\mathcal{H}},\mathcal{P}} + \epsilon. \tag{45}$$

Using the notation in Section 2, we can rewrite (45) as

$$\mathcal{R}_\phi(f) + \eta < \mathcal{R}^*_{\phi,\mathcal{H}} + \delta \implies \mathcal{R}_{\ell_\gamma}(f) < \mathcal{R}^*_{\ell_\gamma,\mathcal{H}} + \epsilon.$$

$\square$

**Theorem 21.** *Let $\mathcal{P}$ be a distribution over $\mathcal{X} \times \mathcal{Y}$. Assume that there exists $g^* \in \mathcal{H}_{\mathrm{lin}}$ such that $\mathcal{R}_{\ell_\gamma}(g^*) = \mathcal{R}^*_{\ell_\gamma,\mathcal{H}_{\mathrm{all}}}$. Let $\phi$ be a margin-based loss. If for $\eta \geq 0$, there exists $f^* \in \mathcal{H}_{\mathrm{lin}} \subset \mathcal{H}_{\mathrm{all}}$ such that $\mathcal{R}_\phi(f^*) \leq \mathcal{R}^*_{\phi,\mathcal{H}_{\mathrm{all}}} + \eta < +\infty$ and $\phi$ is $\mathcal{H}_{\mathrm{lin}}$-calibrated with respect to $\ell_\gamma$, then for all $\epsilon > 0$ there exists $\delta > 0$ such that for all $f \in \mathcal{H}_{\mathrm{lin}}$ we have*

$$\mathcal{R}_\phi(f) + \eta < \mathcal{R}^*_{\phi,\mathcal{H}_{\mathrm{lin}}} + \delta \implies \mathcal{R}_{\ell_\gamma}(f) < \mathcal{R}^*_{\ell_\gamma,\mathcal{H}_{\mathrm{lin}}} + \epsilon.$$

*Proof.* As shown by Bao et al. (2020), the adversarial 0/1 loss $\ell_\gamma = \mathbb{1}_{yf(\mathbf{x}) \leq \gamma}$ is a margin-based loss when $f \in \mathcal{H}_{\text{lin}}$. By Lemma 33, $\ell_{\gamma \mathcal{H}_{\text{lin}}}$ and $\phi_{\mathcal{H}_{\text{lin}}}$ satisfy

$$\int_{\mathcal{X}} \mathcal{C}^*_{\ell_{\gamma \mathcal{H}_{\text{lin}}}, \mathcal{P}(\cdot|\mathbf{x}), \mathbf{x}} \, d\mathcal{P}_X(\mathbf{x}) = \mathcal{R}^*_{\ell_{\gamma \mathcal{H}_{\text{lin}}}, \mathcal{P}},$$

$$\int_X \mathcal{C}^*_{\phi_{\mathcal{H}_{\text{lin}}}, \mathcal{P}(\cdot|\mathbf{x}), \mathbf{x}} \, d\mathcal{P}_X(\mathbf{x}) \leq \mathcal{R}^*_{\phi_{\mathcal{H}_{\text{lin}}}, \mathcal{P}} \leq \int_{\mathcal{X}} \mathcal{C}^*_{\phi_{\mathcal{H}_{\text{lin}}}, \mathcal{P}(\cdot|\mathbf{x}), \mathbf{x}} \, d\mathcal{P}_X(\mathbf{x}) + \eta < +\infty.$$

Since for all $\mathbf{x} \in \mathcal{X}$ and $f \in \mathcal{H}_{\text{lin}}$, $\mathcal{C}_{\ell_{\gamma \mathcal{H}_{\text{lin}}}, \mathcal{P}(\cdot|\mathbf{x}), \mathbf{x}}(f) \leq 1$, we obtain

$$\mathcal{C}_{\ell_{\gamma \mathcal{H}_{\text{lin}}}, \mathcal{P}(\cdot|\mathbf{x}), \mathbf{x}}(f) \leq \mathcal{C}^*_{\ell_{\gamma \mathcal{H}_{\text{lin}}}, \mathcal{P}(\cdot|\mathbf{x}), \mathbf{x}} + 1.$$

Also, since $\phi$ is $\mathcal{H}_{\text{lin}}$-calibrated with respect to $\ell_\gamma$, for all $x \in \mathcal{X}$, $\epsilon > 0$ and $f \in \mathcal{H}_{\text{lin}}$, there exists $\delta > 0$ such that

$$\mathcal{C}_{\phi_{\mathcal{H}_{\text{lin}}}, \mathcal{P}(\cdot|\mathbf{x}), x}(f) < \mathcal{C}^*_{\phi_{\mathcal{H}_{\text{lin}}}, \mathcal{P}(\cdot|\mathbf{x}), \mathbf{x}} + \delta \implies \mathcal{C}_{\ell_{\gamma \mathcal{H}_{\text{lin}}}, \mathcal{P}(\cdot|\mathbf{x}), \mathbf{x}}(f) < \mathcal{C}^*_{\ell_{\gamma \mathcal{H}_{\text{lin}}}, \mathcal{P}(\cdot|\mathbf{x}), \mathbf{x}} + \epsilon.$$

Therefore by Theorem 36, for all $\epsilon > 0$ there exists $\delta > 0$ such that for all $f \in \mathcal{H}_{\text{lin}}$ we have

$$\mathcal{R}_{\phi_{\mathcal{H}_{\text{lin}}}, \mathcal{P}}(f) + \eta < \mathcal{R}^*_{\phi_{\mathcal{H}_{\text{lin}}}, \mathcal{P}} + \delta \implies \mathcal{R}_{\ell_{\gamma \mathcal{H}_{\text{lin}}}, \mathcal{P}}(f) < \mathcal{R}^*_{\ell_{\gamma \mathcal{H}_{\text{lin}}}, \mathcal{P}} + \epsilon. \tag{46}$$

Using the notation in Section 2, we can rewrite (46) as

$$\mathcal{R}_\phi(f) + \eta < \mathcal{R}^*_{\phi, \mathcal{H}_{\text{lin}}} + \delta \implies \mathcal{R}_{\ell_\gamma}(f) < \mathcal{R}^*_{\ell_\gamma, \mathcal{H}_{\text{lin}}} + \epsilon.$$

$\square$

**Theorem 23.** *Given a distribution $\mathcal{P}$ over $\mathcal{X} \times \mathcal{Y}$ and a hypothesis set $\mathcal{H}$ such that $\mathcal{R}^*_{\ell_\gamma, \mathcal{H}} = 0$. Let $\phi$ be a non-increasing margin-based loss. If there exists $f^* \in \mathcal{H} \subset \mathcal{H}_{\text{all}}$ such that $\mathcal{R}_\phi(f^*) = \mathcal{R}^*_{\phi, \mathcal{H}_{\text{all}}} < +\infty$ and $\tilde{\phi}(f, \mathbf{x}, y) = \sup_{\mathbf{x}': \|\mathbf{x} - \mathbf{x}'\| \leq \gamma} \phi(yf(\mathbf{x}'))$ is $\mathcal{H}$-calibrated with respect to $\ell_\gamma$, then for all $\epsilon > 0$ there exists $\delta > 0$ such that for all $f \in \mathcal{H}$ we have*

$$\mathcal{R}_{\tilde{\phi}}(f) < \mathcal{R}^*_{\tilde{\phi}, \mathcal{H}} + \delta \implies \mathcal{R}_{\ell_\gamma}(f) < \mathcal{R}^*_{\ell_\gamma, \mathcal{H}} + \epsilon.$$

*Proof.* By Lemma 32 and Lemma 35, $\tilde{\phi}_{\mathcal{H}}$ is $\mathcal{P}$-minimizable in the almost surely sense. Then for any $n \in \mathbb{N}$, there exists an $f_n^* \in \mathcal{H}$ such that for almost all $\mathbf{x} \in \mathcal{X}$ we have

$$\mathcal{C}_{\tilde{\phi}_{\mathcal{H}}, \mathcal{P}(\cdot|\mathbf{x}), \mathbf{x}}(f_n^*) < \mathcal{C}^*_{\tilde{\phi}_{\mathcal{H}}, \mathcal{P}(\cdot|\mathbf{x}), \mathbf{x}} + \frac{1}{n}.$$

Therefore

$$\mathcal{R}^*_{\tilde{\phi}_{\mathcal{H}}, \mathcal{P}} \leq \int_{\mathcal{X}} \mathcal{C}_{\tilde{\phi}_{\mathcal{H}}, \mathcal{P}(\cdot|\mathbf{x}), \mathbf{x}}(f_n^*) \, d\mathcal{P}_X(\mathbf{x}) \leq \int_{\mathcal{X}} \mathcal{C}^*_{\tilde{\phi}_{\mathcal{H}}, \mathcal{P}(\cdot|\mathbf{x}), \mathbf{x}} \, d\mathcal{P}_X(\mathbf{x}) + \frac{1}{n}$$

$$\leq \inf_{f \in \mathcal{H}} \int_{\mathcal{X}} \mathcal{C}_{\tilde{\phi}_{\mathcal{H}}, \mathcal{P}(\cdot|\mathbf{x}), \mathbf{x}}(f) \, d\mathcal{P}_X(\mathbf{x}) + \frac{1}{n} \leq \mathcal{R}^*_{\tilde{\phi}_{\mathcal{H}}, \mathcal{P}} + \frac{1}{n}.$$

By taking $n \to +\infty$, we obtain

$$\mathcal{R}^*_{\tilde{\phi}_{\mathcal{H}}, \mathcal{P}} = \int_{\mathcal{X}} \mathcal{C}^*_{\tilde{\phi}_{\mathcal{H}}, \mathcal{P}(\cdot|\mathbf{x}), \mathbf{x}} \, d\mathcal{P}_X(\mathbf{x}).$$

Since $\mathcal{R}^*_{\ell_{\gamma \mathcal{H}}, \mathcal{P}} = \mathcal{R}^*_{\ell_\gamma, \mathcal{H}} = 0$, we obtain

$$0 \leq \int_{\mathcal{X}} \mathcal{C}^*_{\ell_{\gamma \mathcal{H}}, \mathcal{P}(\cdot|\mathbf{x}), \mathbf{x}} \, d\mathcal{P}_X(\mathbf{x}) \leq \mathcal{R}^*_{\ell_{\gamma \mathcal{H}}, \mathcal{P}} = 0.$$

Since for all $x \in \mathcal{X}$ and $f \in \mathcal{H}$, $\mathcal{C}_{\ell_{\gamma \mathcal{H}}, \mathcal{P}(\cdot|\mathbf{x}), \mathbf{x}}(f) \leq 1$, we obtain

$$\mathcal{C}_{\ell_{\gamma \mathcal{H}}, \mathcal{P}(\cdot|\mathbf{x}), \mathbf{x}}(f) \leq \mathcal{C}^*_{\ell_{\gamma \mathcal{H}}, \mathcal{P}(\cdot|\mathbf{x}), \mathbf{x}} + 1.$$

Also, since $\tilde{\phi}$ is $\mathcal{H}$-calibrated with respect to $\ell_\gamma$, for all $x \in \mathcal{X}$, $\epsilon > 0$ and $f \in \mathcal{H}$, there exists $\delta > 0$ such that

$$\mathcal{C}_{\tilde{\phi}_{\mathcal{H}}, \mathcal{P}(\cdot|\mathbf{x}), \mathbf{x}}(f) < \mathcal{C}^*_{\tilde{\phi}_{\mathcal{H}}, \mathcal{P}(\cdot|\mathbf{x}), \mathbf{x}} + \delta \implies \mathcal{C}_{\ell_{\gamma \mathcal{H}}, \mathcal{P}(\cdot|\mathbf{x}), \mathbf{x}}(f) < \mathcal{C}^*_{\ell_{\gamma \mathcal{H}}, \mathcal{P}(\cdot|\mathbf{x}), \mathbf{x}} + \epsilon.$$

Therefore by Theorem 36 ($\eta = 0$ here), for all $\epsilon > 0$ there exists $\delta > 0$ such that for all $f \in \mathcal{H}$ we have

$$\mathcal{R}_{\tilde{\phi}_{\mathcal{H}}, \mathcal{P}}(f) < \mathcal{R}^*_{\tilde{\phi}_{\mathcal{H}}, \mathcal{P}} + \delta \implies \mathcal{R}_{\ell_{\gamma \mathcal{H}}, \mathcal{P}}(f) < \mathcal{R}^*_{\ell_{\gamma \mathcal{H}}, \mathcal{P}} + \epsilon. \tag{47}$$

Using the notation in Section 2, we can rewrite (47) as

$$\mathcal{R}_{\tilde{\phi}}(f) < \mathcal{R}^*_{\tilde{\phi}, \mathcal{H}} + \delta \implies \mathcal{R}_{\ell_\gamma}(f) < \mathcal{R}^*_{\ell_\gamma, \mathcal{H}} + \epsilon.$$

$\square$