# OpenReview forum: "Calibration and Consistency of Adversarial Surrogate Losses"
_NeurIPS.cc/2021/Conference — NeurIPS 2021 Spotlight_

### Official Review · Reviewer_yRf4 · 2021-07-12

**Rating:** 6
**Confidence:** 2

**Summary:**

This paper focuses on the H-calibration and H-consistency of loss functions in terms of adversarial robustness. The authors claim that the reason for the difficulty in achieving the adversarial robustness is related to the current convex surrogate losses. Theoretically, they showed the limitation of H-calibration of convex loss functions and surrogate loss functions. Furthermore, they rejected the result from previous studies which provides the relationship between H-calibration and H-consistency, and found a condition that the relationship working correctly. They proposed a non-convex loss function to satisfy H-calibration and experimentally showed the advantages of the proposed loss.

**Ethical Concerns:**

None.

**Limitations And Societal Impact:**

Refer to the main review.

**Main Review:**

Strength.
* Well-written theorems and their assumptions.
* New proposed surrogate loss.

Weakness.
* A more generalized analysis is needed: The authors use l2 norm for adversarial perturbations. How about l1 norm or l_inf norm?
* A question on the writing: Why Theorem 9 and Corollary 11 are disparted not as in Section 4.1?
* The generalization of proposed \rho-margin loss: The reasonable inspiration of proposing \rho-margin loss is not clear. How about other non-convex surrogate losses?
* Explanations on experiment details: In table 1, why the results on \phi_1 and \phi_2 are missing? Do the results are consistent with other perturbation extent \gamma? The theoretical proof of the optimal value in the unit circle experiment (\theta=0.7855) is missing.
* Lack of experiments: This paper suggests a new loss, however, the advantage of the suggested loss is not strongly supported due to the lack of experiments. Zhang et al. [1] similarly proposed a new loss and provided the proof of the calibration of the proposed loss, then included the extensive experiments on MNIST and CIFAR10. If there are more experiments on bigger data, the claims in the paper becomes more convincing.

I would be happy to raise my score if the authors could address the weakness.

[1] Zhang, H., Yu, Y., Jiao, J., Xing, E., El Ghaoui, L., & Jordan, M. (2019, May). Theoretically principled trade-off between robustness and accuracy. In International Conference on Machine Learning (pp. 7472-7482). PMLR.

**Time Spent Reviewing:**

8 hours

---

> ### Author Response · Authors · 2021-08-09
> **Response to Reviewer yRf4**
>
> Thank you for your comments. We have carefully addressed all the questions raised. Please find our responses below.
>
> **1. A more generalized analysis is needed: The authors use l2 norm for adversarial perturbations. How about l1 norm or l_inf norm?**
>
> **Response:**
> Our analysis can be extended directly to other perturbations such as the $\ell_1$ ball or $\ell_{\infty}$ ball, and in fact for any $\ell_p$ norm for $p \in [1, \infty]$. In particular, the same proof of our calibration and consistency results for general hypothesis sets (e.g. Theorem 7, Theorem 10, Theorem 16, Theorem 23, Theorem 24) works for other perturbations too. We will further elaborate on that in the final version.
>
> **2. A question on the writing: Why Theorem 9 and Corollary 11 are disparted not as in Section 4.1?**
>
> **Response:**
> In Section 4.1, all the hypothesis sets considered in Corollary 8 satisfy the conditions in Theorem 7 and thus can use the same proof. In Section 4.2, the hypothesis sets considered in Theorem 9 do not satisfy the conditions in Theorem 10, while the hypothesis sets considered in Corollary 11 satisfy the conditions in Theorem 10. Therefore, Theorem 9 and Corollary 11 are presented separately due to the different proofs.
>
> **3. The generalization of proposed \rho-margin loss: The reasonable inspiration of proposing \rho-margin loss is not clear. How about other non-convex surrogate losses?**
>
> **Response:**
> Our choice of the $\rho$-margin loss is guided by the fact that it does satisfy the quasi-concavity requirement of the corresponding inner risk (Section 4.3.1). Moreover, it has been shown recently that, e.g., in [1’],  the $\rho$-margin loss can benefit from favorable generalization margin bounds with the derived algorithmic solution.
> Additionally, other similar non-convex surrogate losses that satisfy the calibration requirement could be good candidates (although they would require a specific analysis).
>
> **4. Explanations on experiment details: In table 1, why the results on \phi_1 and \phi_2 are missing? Do the results are consistent with other perturbation extent \gamma? The theoretical proof of the optimal value in the unit circle experiment (\theta=0.7855) is missing.**
>
> **Response:**
> The goal of Table 1(a) is to demonstrate that indeed $\mathcal{H}$-calibrated surrogates may not be $\mathcal{H}$-consistent unless assumptions on the data distribution are made. We only report the results for the four surrogates, since it has already demonstrated the point. The goal of Table 1(b) is to study the necessity of the assumptions in Theorem 20. Hence we add $\phi_1$ and $\phi_2$ whose Bayes $(\phi_{\mathrm{sur}}, \mathcal{H}_{\mathrm{lin}})$-risk is 0 as special cases verifying the conditions of Theorem 20 for $\eta = 0$.
>
> The results are consistent with other $\gamma$.
>
> In the experiments, all risks are approximated by their empirical counterparts computed over $10^7$ i.i.d. samples. To find the minimizer of the robust 0-1 loss, we perform grid search over a fine grid of $\theta$.
> There also exists an exact formula. The distribution in the Unit Circle case matches the one studied in the proof of Theorem 18 in Appendix E.6 when $\sigma = \frac{\pi}{2}$. We proved there that the unique minimizer has $\theta = \frac{\sigma}{2}$ and the corresponding risk equals $1-\frac{\sigma}{\pi}$, which matches the experimental results.
>
> **5. Lack of experiments: This paper suggests a new loss, however, the advantage of the suggested loss is not strongly supported due to the lack of experiments. Zhang et al. [1] similarly proposed a new loss and provided the proof of the calibration of the proposed loss, then included the extensive experiments on MNIST and CIFAR10. If there are more experiments on bigger data, the claims in the paper becomes more convincing.**
>
> **Response:**
> Thank you for the suggestions. We would like to point out that the work of Zhang et al. does not provide guarantees for consistency of the adversarial 0-1 loss. The authors in [1] show that for any classification calibrated surrogate loss (including convex losses), the difference of the robust accuracy and the natural accuracy of the classifier obtained by optimizing the surrogate can be upper bounded by a term that captures the vulnerability of the surrogate near the boundary. However, this does not provide any theoretical guarantees on when the adversarial risk itself can be bounded. This is for a good reason, since as we show, no convex surrogate can be consistent. Hence ours is the first work to formally establish calibration and consistency properties that directly apply to the adversarial loss.
>
> We have also initiated an experimental study based on our theoretical results. We have already obtained some very promising preliminary results. Here is one example. For the adversarial binary classification (with perturbation size 0.3, measured in $\ell_{\infty}$ norm) on two hard digits 1 vs 7 from MNIST, using the same neural network for MNIST as in Madry et al. (2017), our experiment shows sup-$\rho$ margin loss can achieve 98.75% adversarial accuracy, outperforming the benchmark sup-logistics loss (default loss used in practice). For this example, our proposed conditions of $\mathcal{H}$-consistency (Theorem 24) can also be empirically verified. It further empirically demonstrates the advantage of the proposed loss over the commonly used loss functions.
>
>
> [1'] Relative Deviation Margin Bounds. Corinna Cortes, Mehryar Mohri, Ananda Theertha Suresh. ICML 2021

---

> > ### Comment · Reviewer_yRf4 · 2021-08-12
> > **Rating Changed**
> >
> > The authors have cleared my questions. I'm raising my score.

---

> > > ### Author Response · Authors · 2021-08-17
> > > **Thank You**
> > >
> > > We are glad to hear we could clear all questions raised. Please let us know if there is any other concern we can address for the reviewer to consider further increasing his overall score.

---

### Official Review · Reviewer_vHnZ · 2021-07-14

**Rating:** 9
**Confidence:** 3

**Summary:**

This paper provides an extensive analysis of the satisfiability of H-calibration and H-consistency for surrogate losses in binary classification, where the evaluation metric is not the standard 0-1 loss but the adversarial 0-1 loss. The adversarial 0-1 loss is a pointwise loss defined on an input x by calculating the supremum of the 0-1 loss value of a point that can perturb around x within a ball of a pre-specified radius. This paper reveals a negative result showing that many surrogate losses are not H-consistent, especially the convex losses.  This paper also reveals a positive result of a certain class of non-convex losses that can be H-consistent under some assumptions. Synthetic experiments were also provided to justify the relevance of theoretical findings.

**Ethical Concerns:**

None.

**Limitations And Societal Impact:**

The discussion was adequate.

**Main Review:**

Although this paper looks theoretically heavy and dense at glance, I found that reading this paper was a very enjoyable experience. The paper is very well-written. The authors did a very good job to provide motivation and intuitive explanations for most theoretical results presented in this paper. I appreciate the fact that both margin-based losses and their supremum version are both studied. Also, the hypothesis class considered in this paper goes beyond a simple linear model. Furthermore, it resolves some issues of recent work. Overall, I found the theoretical results of this paper are highly significant and definitely help to improve our understanding of a surrogate loss behavior in adversarially robust classification. Thus, I vote for a strong acceptance for this paper.

Weaknesses:
1. Perhaps more illustration at the beginning of the paper can improve the readability for readers who are not familiar with adversarial surrogate losses, margin losses, a regularity condition for $\mathcal{H}$. Due to space limitations, I understand that doing all this can be quite challenging but it may still be worth considering.
2. One may criticize that the result is limited to the binary classification case, but I think the result only on the binary classification is already sufficient.
3. Some parts of the paper may not be self-contained enough since many contents are deferred to Appendix (deferring the proof is perfectly acceptable in my opinion). But for example, in experiments, my impression is that it is necessary to write down a loss function form (provided in Appendix C.1).

Questions:
1. It is very insightful that H-calibration does not imply H-consistency. Nevertheless, in the setting of adversarial classification considered here, can we say that H-calibration is a "necessary" condition for H-consistency? I believe so but it was not stated clearly in the paper in my understanding or perhaps I missed it. If so, then the notion of H-calibration is still very useful to filter out some surrogate losses without checking the H-consistency condition.
2. Is there any existing results of Fisher consistency in the context of adversarial classification before this paper? If not, do I understand correctly that when we use $\mathcal{H}_{\textrm{all}}$, we are considering a notion of standard Fisher consistency (i.e., infinite sample consistency) and this paper is the first to prove such results?
3. I am a bit confused around Lines 156--160. it is said that studying regular $mathcal{H}$ is enough because Theorem 6, which is specific for symmetric hypothesis set, states that "if $\mathcal{H}$ is NOT regular, then any surrogate loss is $\mathcal{H}$-calibrated. My understanding is that for non-regular set, it causes the definition of $\mathcal{H}$-calibration to be useless, however, since $\mathcal{H}$-calibration is not equivalent to $\mathcal{H}$-consistency, wouldn't this mean they might be worth considering?
4. How strong the requirement that $\mathcal{H}$ is symmetric is? Is this the case for most hypothesis classes we can think of?
5. Is there other settings that H-calibration does not imply H-consistency other than adversarially robust classification? If I understood correctly, I guess this setting is not the first work that has this phenomenon. Thus, it could be useful to discuss this phenomenon a little bit more to suggest that it is not special only for adversarially robust classification (if space allows). To my knowledge, if I understand correctly, I am aware that in AUC-optimization, calibration also does not imply consistency in the all measurable function cases [1][2].
6. Does a relaxed assumption to obtain H-calibration that does not require the Bayes $(\ell_0, \mathcal{H})$-risk to be zero in Theorem 21--23  equivalent to saying that the model misspecification does not occur (i.e., it is well-specified)?
7. Do the presented numbers in Table 1 sufficient to tell that surrogate risks of all candidate losses were successfully minimized? If not, is there a way to do that (maybe it is difficult...)?
8. Is it straightforward to extend this framework to the multiclass case?

[1] Uematsu and Lee, On Theoretically Optimal Ranking Functions in Bipartite Ranking, JASA, 2016

[2] Gao and Zhou, On the consistency of AUC pairwise optimization, IJCAI, 2015 (Lemma 3).

Minor comments:
1. The use of \citep and \citet are mixed in a strange way. I am not sure if this was intentional. For example, I think Line 72, many parts in Appendix B, Line 79, 122, 132, 133, 162, 237, 342, 345, 364, 365 should be \citet in my understanding.

Update: I have read other reviews and authors' feedback. I believe this paper did a great job and therefore my score remains unchanged (strong accept).

**Time Spent Reviewing:**

10 hours

---

> ### Author Response · Authors · 2021-08-09
> **Response to Reviewer vHnZ**
>
> Thank you for your encouraging review. We will take your suggestions into account when preparing the final version. Below please find responses to specific questions.
>
> **Response to Weaknesses:**
> Thank you for the appreciation of our work and the detailed suggestions. We will follow your suggestions to improve our presentation for readers unfamiliar with some of these notions.
> Additionally, we have already initiated the study of the multi-class setting and hope to present that as a follow-up. While our extension benefits from several of the results in this paper, it also comes with new challenges and proofs needed to tackle the additional complexity arising in the multi-class setting, already present in the non-adversarial analysis of consistency.
>
> **Response to Questions:**
>
> **1. It is very insightful that H-calibration does not imply H-consistency. Nevertheless, in the setting of adversarial classification considered here, can we say that H-calibration is a "necessary" condition for H-consistency? I believe so but it was not stated clearly in the paper in my understanding or perhaps I missed it. If so, then the notion of H-calibration is still very useful to filter out some surrogate losses without checking the H-consistency condition.**
>
> **Response:**
> That’s a good question! As with the current references, it’s common sense to first consider $\mathcal{H}$-calibration when considering $\mathcal{H}$-consistency. Theoretically, we can establish that $\mathcal{H}$-calibration (Definition 2) is a necessary condition for $\mathcal{H}$-consistency (Definition 1) by considering a distribution supported on a single point (calibration is a pointwise property). Empirically, our experiments also show that the non-calibrated surrogates are not consistent. We will state this more clearly in the paper.
>
> **2. Is there any existing results of Fisher consistency in the context of adversarial classification before this paper? If not, do I understand correctly that when we use $\mathcal{H}_{\mathrm{all}}$, we are considering a notion of standard Fisher consistency (i.e., infinite sample consistency) and this paper is the first to prove such results?**
>
> **Response:**
> That’s a good point! When considering $\mathcal{H}_{\mathrm{all}}$, $\mathcal{H}$-consistency exactly coincides with the standard Fisher consistency. To the best of our knowledge, our work is the first to prove such results in the context of adversarial classification. Note that the results of Bao et al. (even after correction) apply only to the linear case.
>
>
> **3. I am a bit confused around Lines 156--160. it is said that studying regular $\mathcal{H}$ is enough because Theorem 6, which is specific for symmetric hypothesis set, states that "if $\mathcal{H}$ is NOT regular, then any surrogate loss is $\mathcal{H}$-calibrated. My understanding is that for non-regular set, it causes the definition of $\mathcal{H}$-calibration to be useless, however, since $\mathcal{H}$-calibration is not equivalent to $\mathcal{H}$-consistency, wouldn't this mean they might be worth considering?**
>
> **Response:**
> Sorry for the confusion. The “enough” here is only meant for the calibration part in Section 4. Of course it is still worth considering them for the consistency part in Section 5. Since any surrogate loss is $\mathcal{H}$-calibrated for such hypothesis sets, the only requirement for the surrogates to be consistent is to satisfy the consistency condition in Section 5 without specifically looking into the calibration. Thank you for pointing this out. We will make it more clear in the final version.
>
> **4. How strong the requirement that $\mathcal{H}$ is symmetric is? Is this the case for most hypothesis classes we can think of?**
>
> **Response:**
> We believe that symmetry holds for most hypothesis sets one can think of, especially for commonly used hypothesis sets such as the linear hypothesis set, neural networks with a fully connected layer at the top, and all measurable functions. Additionally, we can extend our results beyond the symmetric case.
>
> **5. Is there other settings that H-calibration does not imply H-consistency other than adversarially robust classification? If I understood correctly, I guess this setting is not the first work that has this phenomenon. Thus, it could be useful to discuss this phenomenon a little bit more to suggest that it is not special only for adversarially robust classification (if space allows). To my knowledge, if I understand correctly, I am aware that in AUC-optimization, calibration also does not imply consistency in the all measurable function cases [1][2].**
>
> **Response:**
> Thanks for pointing out these references. We will include a discussion in the final version. Broadly speaking these works show that classification calibrated losses may not be consistent for other tasks such as ranking/AUC-optimization. Hence here calibration not implying consistency stems from the mismatch between using a calibrated surrogate for a different loss (classification loss) and applying it for a different purpose.
> However, in our case the situation is more subtle. Even a calibrated surrogate (with respect to the adversarial 0-1 loss) may not be consistent in general.
>
> **6. Does a relaxed assumption to obtain H-calibration that does not require the Bayes ($\ell_0$, $\mathcal{H}$)-risk to be zero in Theorem 21--23 equivalent to saying that the model misspecification does not occur (i.e., it is well-specified)?**
>
> **Response:**
> We can say that the relaxed assumption in Theorem 21 means the model is well-specified in the sense that the Bayes classifier in the all measurable functions class is contained in the considered linear hypothesis set. The condition Bayes risk to be zero is still required In Theorem 22 and 23.
>
> **7. Do the presented numbers in Table 1 sufficient to tell that surrogate risks of all candidate losses were successfully minimized? If not, is there a way to do that (maybe it is difficult...)?**
>
> **Response:**
> In the experiments, all risks are approximated by their empirical counterparts computed over $10^7$ i.i.d. samples. To find the minimizer of each loss, we perform grid search over a fine grid of $\theta$. We can say that surrogate risks of all surrogate losses are successfully minimized since the error is negligible compared to the difference between any reported surrogate loss and the adversarial 0/1 loss. In terms of the adversarial 0/1 loss, there exists an exact formula for the minimizer. The distribution in the Unit Circle case matches the one studied in the proof of Theorem 18 in Appendix E.6 when $\sigma = \frac{\pi}{2}$. We proved there that the unique minimizer corresponds to $\theta = \frac{\sigma}{2}$ and the corresponding risk equals $1-\frac{\sigma}{\pi}$, which matches the experimental results.
>
> **8. Is it straightforward to extend this framework to the multiclass case?**
>
> **Response:**
> For a first study of calibration and consistency in the adversarial robust classification for general hypothesis sets, binary classification is a natural starting point that already poses several new challenges compared to the standard classification, which we addressed in a comprehensive way. The extension to multi-class is not straightforward since, as in the standard non-adversarial case the multi-class setting comes with some complications. Nevertheless, we have already initiated that extension and can already report that the extension of several results can be proven by extending the proofs for the binary classification and that some other results require addressing some extra questions.
>
> **Response to Minor comments:**
> We will double check the use of \citet vs. \citep and make sure that it is consistent.

---

> > ### Comment · Reviewer_vHnZ · 2021-08-25
> > **Thank you for your detailed answers.**
> >
> > I really appreciate that the authors answered every single question that I asked. I learned a lot from reading this paper and also the authors' feedback. It is nice to know that the result of this paper also covers the standard Fisher consistency case (maybe saying that a bit as well in the paper would be nice). My score remains unchanged (strong accept). I am quite convinced that this paper did an amazing job to advance the field and thus it should be accepted.

---

> > > ### Author Response · Authors · 2021-08-25
> > > **Thank you**
> > >
> > > We thank the reviewer for the strong support of our work! We enjoy thinking through all the reviewer's great questions, and really appreciate the insightful comments and constructive suggestions.

---

### Official Review · Reviewer_mcZd · 2021-07-15

**Rating:** 6
**Confidence:** 3

**Summary:**

This paper provides a detailed analysis on the H-calibration and H-consistency of adversarial surrogate losses.

It shows that some common used losses for adversarial training are not H-calibrated.

Some theoretical results on the characterization of H-calibration are established.  The paper also proves that some surrogate losses are indeed H-calibrated.

Some additional theoretical results on H-consistency are established too.



**Limitations And Societal Impact:**

Yes

**Main Review:**

1. This paper contains too many theorems, definitions, propositions, and corollaries. It is very difficult to track the main ideas, especially for those who are not familiar with this area.

2. There is a key reference Bao et al. (2020). It seems that this paper is questioning that many results in Bao et al. (2020) are wrong results. Is it possible to make a summary what results in Bao et al. (2020) are not correct and how to correct them?

3. There is a gap between the theory and the practice. Can you provide some numerical analysis on some benchmark datasets by using the proposed H-consistency losses? It would be interesting to see the benefit over the commonly uses loss functions.

4. The perturbation ball is fixed to be the $\ell_2$ ball. Can the analysis be extended to other perturbation such as the $\ell_\infty$ ball?

5. For the proofs of these theorems, is there any novel techniques beyond the current references?

6. Is it straightforward to extent the current analysis from binary classification to multiple-class classification?

**Time Spent Reviewing:**

4

---

> ### Author Response · Authors · 2021-08-10
> **Response to Reviewer mcZd**
>
> Thank you for your comments. We have carefully addressed all the questions raised. Please find our responses below.
>
> **1. This paper contains too many theorems, definitions, propositions, and corollaries. It is very difficult to track the main ideas, especially for those who are not familiar with this area.**
>
> **Response:**
> Thank you for the comments. We totally understand that the paper looks theoretically dense. This is because we present an extensive set of results tackling both $\mathcal{H}$-calibrations and $\mathcal{H}$-consistency. We have sought to discuss the strong motivation for this work and give intuitive explanations for the theoretical results presented in the paper and as the Reviewer vHnZ commented, such as before or after all our main theorems. We will continue working on improving them in the final version and make it more accessible. Specifically, we will add a short overview to the beginning of each main section.
>
> **2. There is a key reference Bao et al. (2020). It seems that this paper is questioning that many results in Bao et al. (2020) are wrong results. Is it possible to make a summary what results in Bao et al. (2020) are not correct and how to correct them?**
>
> **Response:**
> We commented on (Bao et al., 2020) in Appendix B and will make a more clear summary of that in the final version as suggested. For clarity, the main points are as follows.
>
> (1) Incorrect step in (Bao et al., 2020): the definition of calibration adopted by the authors does not coincide with the correct definition (Steinwart, 2007) in the case of the linear models they study: the minimal inner risk in the definition should be defined for a fixed $x$ and the infimum should be over $f$, instead of an infimum over both $f$ and $x$.
>
> How to correct it: we adopt the correct definition (Definition 2) in the paper. We fix previous calibration results presented for the family of linear models in (Bao et al., 2020) and significantly generalize the results to the nonlinear hypothesis sets.
>
> (2) Incorrect step in (Bao et al., 2020): the authors mixed up the calibration and consistency without considering $P$-minimizability. They incorrectly concluded that $\mathcal{H}$-calibration of proposed surrogate losses implies their $\mathcal{H}$-consistency.
>
> How to correct it: We point out that $P$-minimizability does not hold, in general, for adversarially robust classification and a specific hypothesis set $\mathcal{H}$, although it holds for standard binary classification and the family of all measurable functions (Steinwart, 2007, Theorem 3.2).
> We prove that, in the absence of distributional assumptions, many surrogate losses shown to be $\mathcal{H}$-calibrated are in fact not $\mathcal{H}$-consistent (Theorem 18). This counterexample, in turn falsifies a claim presented in (Bao et al., 2020) which incorrectly concluded that $\mathcal{H}$-calibration of proposed surrogate losses implies their $\mathcal{H}$-consistency. In contrast, we provide natural $\mathcal{H}$-consistency guarantees under realizability assumptions that have been studied in prior works (Long and Servedio (2013) and Zhang and Agarwal (2020)).
>
> (3) Incorrect step in (Bao et al., 2020): the experiments in (Bao et al., 2020) are problematic.  Equation (12) in Appendix D.1., which they used to compute the Bayes risks, is wrong since in general $R_{\ell}(f^*) = R_{\ell,\mathcal{H}}^*$ cannot imply $C_{\ell}(f^*,x,\eta)=C_{\ell,\mathcal{H}}^*(x,\eta)$ when $\mathcal{H}=\mathcal{H}_{\mathrm{lin}}$.
>
> How to correct it: With the correct approximation of Bayes risks, our experiments further empirically demonstrate that indeed the $\mathcal{H}$-calibrated losses proposed in (Bao et al., 2020) are not $\mathcal{H}$-consistent and justify our proposed conditions for $\mathcal{H}$-consistency.
>
> **3. There is a gap between the theory and the practice. Can you provide some numerical analysis on some benchmark datasets by using the proposed H-consistency losses? It would be interesting to see the benefit over the commonly uses loss functions.**
>
> **Response:**
> Thank you for the suggestions. We have initiated an experimental study based on our theoretical results. We have already obtained some very promising preliminary results. Here is one example. For the adversarial binary classification (with perturbation size 0.3, measured in $\ell_{\infty}$ norm) on two hard digits 1 vs 7 from MNIST, using the same neural network for MNIST as in Madry et al. (2017), our experiment shows sup-$\rho$ margin loss can achieve 98.75% adversarial accuracy, outperforming the benchmark sup-logistics loss (default loss used in practice). For this example, our proposed conditions of $\mathcal{H}$-consistency (Theorem 24) can also be empirically verified. It further empirically demonstrates the advantage of the proposed loss over the commonly used loss functions.
>
> **4. The perturbation ball is fixed to be the $\ell_2$ ball. Can the analysis be extended to other perturbation such as the $\ell_{\infty}$  ball?**
>
> **Response:**
> Our analysis can be extended directly to other perturbations such as the $\ell_1$ ball or $\ell_{\infty}$ ball, and in fact for any $\ell_p$ norm for $p \in [1, \infty]$. In particular, the same proof of our calibration and consistency results for general hypothesis sets (e.g. Theorem 7, Theorem 10, Theorem 16, Theorem 23, Theorem 24) works for other perturbations too. We will further elaborate on that in the final version.
>
> **5. For the proofs of these theorems, is there any novel techniques beyond the current references?**
>
> **Response:**
> To the best of our knowledge, this is the first work to study the calibration and consistency in the adversarial robust classification for general hypothesis sets. As reviewer ks5M commented, calibration and consistency to the standard 0-1 loss is by now a classical topic with over a decade of serious research in the area. However, the adversarial loss presents new challenges and requires carefully distinguishing among different notions to avoid drawing false conclusions. Additionally, $\mathcal{H}$-calibration and $\mathcal{H}$-consistency, that is calibration and consistency for a specific hypothesis set H are still research topics even in the standard non-adversarial case. As an example, the recent COLT 2020 paper of Bao et al. (2020), which presents a study of $\mathcal{H}$-calibration for the adversarial loss in the special case where $\mathcal{H}$ is the class of linear functions, concludes that the $\mathcal{H}$-calibrated surrogates they propose are $\mathcal{H}$-consistent. This is falsified as a by-product of our results, which further suggests that the adversarial setting is more complex and requires a more delicate analysis. In (Bao et al., 2020), calibration analysis on the nonlinear hypothesis sets is actually an open problem. In contrast with that prior work, instead of studying the $\gamma$-margin loss and the specific family of linear models, we directly study the adversarial 0/1 loss and general hypothesis sets by using the novel theoretical analysis. In particular, Theorem 24 in our paper provided consistent surrogates for general hypothesis sets, including the complex multi-layer neural networks. Our proofs are novel and do not readily follow those of previous work referenced.
>
> **6. Is it straightforward to extent the current analysis from binary classification to multiple-class classification?**
>
> **Response:**
> For a first study of calibration and consistency in the adversarial robust classification for general hypothesis sets, binary classification is a natural starting point that already poses several new challenges compared to the standard classification, which we addressed in a comprehensive way. The extension to multi-class is not straightforward since, as in the standard non-adversarial case the multi-class setting comes with some complications. Nevertheless, we have already initiated that extension and can already report that the extension of several results can be proven by extending the proofs for the binary classification and that some other results require addressing some extra questions.

---

> > ### Comment · Reviewer_mcZd · 2021-08-13
> > **Response is clear**
> >
> > The response has cleared most of my concerns. I am raising my score to 6.

---

> > > ### Author Response · Authors · 2021-08-17
> > > **Thank You**
> > >
> > > We are glad to hear we could clear concerns. Please let us know if there is any other question we can answer for the reviewer to consider further increasing his overall score.

---

### Official Review · Reviewer_ks5M · 2021-07-18

**Rating:** 8
**Confidence:** 3

**Summary:**

The authors provide an extensive characterization of calibration and consistency for surrogates to the l2 robust 0-1 loss, along the lines of what Bartlett et al. and Steinwart provided for the standard 0-1 loss.  They demonstrate that convex losses and supremum-based convex losses are not calibrated, and further that calibration is in general not sufficient to guarantee consistency.  This corrects a previous incorrect assertion made by Bao et al. in a COLT 2020 paper.  They provide experiments which corroborate their results.

**Limitations And Societal Impact:**

Adequately addressed

**Main Review:**

The results in this paper are, to my knowledge, significant.  Consistency and calibration of convex surrogates to the standard 0-1 loss is by now a classical topic with over a decade of serious research in the area.  Given the numerous questions about the existence of adversarially robust classifiers in complicated hypothesis classes, the study of calibration and consistency in this setting is important.  Although Bao et al. (2020) analyzed consistency and calibration of surrogates to the l2 robust 0-1 loss, the authors demonstrate that they made serious errors throughout their work, thus this work helps address a significant problem with previous work.  They also provide a number of significant extensions of the (incorrect) work of Bao et al, providing analysis of how the calibration and consistency results differ when using linear/relu hypothesis classes as well as the classes of neural networks and all measurable functions.

I found the exposition clean and friendly to read in the main section, which was impressive given the large number of results presented in the paper.  (I did not have time to read the proof details in the appendix.)  I don't have any significant issues with the paper, but below are a few minor points that I think could help improve the paper.

(1) Figure 1 is unreadable.  Even zooming to maximum on computer, it wasn't possible to see what was happening.  Please re-make the figure and give a more descriptive caption.

(2) A figure for the Segments experiment, showing what the distribution looks like, would be helpful.

(3) For experiments, it wasn't clear to me how the authors calculated the minimizer of the robust 0-1 loss for different settings; is there an exact formula that is possible from using the constructed distribution?  If not, how?

(4) Further discussion on the computational tradeoffs of adversarial consistency would improve the paper.  The results suggest that in order for consistency to hold in this setting, one must use non-convex surrogates, leaving open the question of how to minimize such surrogates, even in the linear hypothesis setting.  Recent work has shown that in the non-robust linear hypothesis setting, minimizers of convex surrogates have sub-optimal performance for learning linear classifiers in the non-realizable setting, while minimizing nonconvex surrogates with gradient descent leads to nearly-optimal guarantees [1'].  Likewise, the current state-of-the-art algorithms for learning robust linear classifiers in the non-realizable setting are based on a supremum-based *non*-convex surrogate in the language of the paper, i.e. adversarial training on a non-convex sigmoidal loss, with convex surrogates having less strong guarantees (although with no lower bounds proving a separation) [2'].  It would be worth commenting on these two works, e.g. around lines 189-191.


[1'] Non-Convex SGD Learns Halfspaces with Adversarial Label Noise. Ilias Diakonikolas, Vasilis Kontonis, Christos Tzamos, Nikos Zarifis. NeurIPS 2020

[2'] Provable Robustness of Adversarial Training for Learning Halfspaces with Noise. Difan Zou, Spencer Frei, Quanquan Gu.  ICML 2021

** Post rebuttal **

I'm happy with the author response and am sticking with my strong recommendation for acceptance.

**Time Spent Reviewing:**

6

---

> ### Author Response · Authors · 2021-08-09
> **Response to Reviewer ks5M**
>
> Thank you for your appreciation of our work and suggestions on improving the readability. We will take them all into account when preparing the final version. Below please find responses to specific questions.
>
> **(1) Figure 1 is unreadable. Even zooming to maximum on computer, it wasn't possible to see what was happening. Please re-make the figure and give a more descriptive caption.**
>
> **Response:** Thank you for the suggestions. We will make Figure 1 more readable and provide a better caption in the final version.
>
> **(2) A figure for the Segments experiment, showing what the distribution looks like, would be helpful.**
>
> **Response:** Thank you for your suggestion. We will add the figure in the final version.
>
> **(3) For experiments, it wasn't clear to me how the authors calculated the minimizer of the robust 0-1 loss for different settings; is there an exact formula that is possible from using the constructed distribution? If not, how?**
>
> **Response:** In the experiments, all risks are approximated by their empirical counterparts computed over $10^7$ i.i.d. samples. To find the minimizer of the robust 0-1 loss, we perform grid search over a fine grid of $\theta$.
> There also exists an exact formula. The distribution in the Unit Circle case matches the one studied in the proof of Theorem 18 in Appendix E.6 when $\sigma = \frac{\pi}{2}$. We proved there that the unique minimizer has $\theta = \frac{\sigma}{2}$ and the corresponding risk equals $1-\frac{\sigma}{\pi}$, which matches the experimental results. For the Segments case, since $\hat{\gamma}$ is slightly larger than $\gamma$, we can directly observe that $w^* = (1,0
> )$ achieves zero Bayes risk.
>
> **(4) Further discussion on the computational tradeoffs ... around lines 189-191.**
>
> **Response:** Thank you for bringing those works to our attention. These works are of course relevant and we will comment on them in the final version. Broadly speaking, the work of [1'] considers agnostic learning (as mentioned by the reviewer, a setting different from adversarial learning) and shows that under certain distributional assumptions on the marginal of x, SGD on the sigmoid loss achieves a constant factor approximation to the error of the best halfspace. The work of [2'] establishes a similar (with worse approximation guarantees) for competing with the adversarial error of the best halfspace.
>
> Our work on the other hand is concerned with studying the notion of calibration and consistency for arbitrary hypothesis classes. Hence, our consistency results suggest that under certain assumptions, the use of the supremum-based $\rho$-margin loss can compete with the robust error of the best function in a class $\mathcal{H}$ (say a class of neural networks). We will certainly discuss these works more specifically in the final version.

---

### Public Comment · ~Han_Bao2 · 2021-12-05
**Remark from the authors of “Calibrated surrogate losses for adversarially robust classification”**

We are the authors of the COLT 2020 paper “Calibrated surrogate losses for adversarially robust classification” (Bao et al., 2020). We agree with the reviewers that the present paper makes important contributions to the study of H-calibration and H-consistency in the context of adversarially robust classification, and believe these results are worthy of inclusion in the NeurIPS 2021 program.

We would like to offer some reflections on the camera-ready version of the paper in relation to our paper.

This paper identifies and addresses two basic issues in our paper: (a) we made an erroneous remark that H-calibration always implies H-classification; (b) we did not use the correct definition of calibration as studied by Steinwart (2007). The latter error propagated throughout the paper. We acknowledge both of these errors.

On May 13, 2021, our corrigendum appeared on arXiv: https://arxiv.org/pdf/2005.13748v2.pdf, and we emailed the link to the authors of the present paper. These corrections coincide with those of the present paper. These corrections were arrived at independently, with one exception noted below. We thank the authors for contacting us in Dec. 2020, pointing out an issue with our experiments which led us to discover (b), and for later sharing two technical reports. Remark 1 of our corrigendum contains a concise summary of all updates to our original COLT paper.

Our corrigendum points out that issue (a) was a remark used to motivate the study of calibration, but was not a formal result and did not factor into the rest of the paper. Some reviewers have the impression that the erroneous remark is one of our formal results. See, for example, the comments of reviewer mcZd and the authors’ response.

Regarding (b), our corrigendum clearly states that when the correct definition of calibration is used, all formal results in our COLT 2020 paper are easily salvaged. The theorem statements, theorem proofs, and example loss functions require only occasional and minor modifications to constants. Several statements and examples require no modification, and inconsistencies with our numerical examples disappear.

The primary technical contribution in our COLT paper is the identification of a sufficient condition on a surrogate loss to ensure H-calibration when H is the set of linear classifiers.  The present paper uses the same condition in its reconsideration of one of our results, and further analysis of $\rho$-margin losses. This condition pertains to the quasiconcavity of the conditional risk (Assumption A in our corrigendum). We note that this formal assumption did not appear in an arXiv version of the present paper published on May 6, 2021 (https://arxiv.org/pdf/2105.01550v2.pdf). Instead the authors relied on a different condition that we had used in our COLT paper that turned out to have some deficiencies (see Remark 17 of our corrigendum for details). Therefore it appears to us that the authors have benefited from our corrigendum, without proper acknowledgement, in the formulation of some of their results.

Our corrigendum is mentioned once in the camera ready version of the paper, in an appendix that responds to a reviewer request (it is also incorrectly given a 2020 date; it appeared in 2021). We believe that citing the corrigendum in the main text at appropriate places (as discussed above) would have led readers to a better appreciation of the contributions of both our paper and the present paper. We request that future references to our paper cite and refer to the corrigendum.

Respectfully,

Han Bao, Clayton Scott, Masashi Sugiyama

---

### Decision · Program_Chairs · 2021-09-27

**Decision:**

Accept (Spotlight)

**Comment:**

I agree with the reviewers that this paper makes substantive contributions to understanding the role of surrogate losses in adversarial learning. To make the paper more accessible, I encourage the authors to make the exposition (at least the first few sections) more friendly to readers who don’t have an extensive background in learning theory.